# THE POWER OF CONTRAST FOR FEATURE LEARNING: A THEORETICAL ANALYSIS

## ABSTRACT

Contrastive learning has achieved state-of-the-art performance in various self-supervised learning tasks and even outperforms its supervised counterpart. Despite its empirical success, theoretical understanding of why contrastive learning works is still limited. In this paper, under linear representation settings, (i) we provably show that contrastive learning outperforms autoencoder, a classical unsupervised learning method, for both feature recovery and downstream tasks; (ii) we also illustrate the role of labeled data in supervised contrastive learning. This provides theoretical support for recent findings that contrastive learning with labels improves the performance of learned representations in the in-domain downstream task, but it can harm the performance in transfer learning. We verify our theory with numerical experiments.

## 1 INTRODUCTION

Deep supervised learning has achieved great success in various applications, including computer vision (Krizhevsky et al., 2012), natural language processing (Devlin et al., 2018), and scientific computing (Han et al., 2018). However, its dependence on manually assigned labels, which is usually difficult and costly, has motivated research into alternative approaches to exploit unlabeled data. Self-supervised learning is a promising approach that leverages the unlabeled data itself as supervision and learns representations that are beneficial to potential downstream tasks.

At a high level, there are two common approaches for feature extraction in self-supervised learning: generative and contrastive (Liu et al., 2021). Both approaches aim to learn latent representations of the original data, while the difference is that the generative approach focused on minimizing the reconstruction error from latent representations, and the contrastive approach targets to decrease the similarity between the representations of contrastive pairs. Recent works have shown the benefits of contrastive learning in practice (Chen et al., 2020a; He et al., 2020; Chen et al., 2020b;c). However, why the contrastive approach outperforms the generative approach remains mysterious.

Additionally, recent works aim to further improve contrastive learning by introducing the label information. Specifically, Khosla et al. (2020) proposed the *supervised contrastive learning*, where the contrasting procedures are performed across different classes rather than different instances. With the help of label information, their proposed method outperforms self-supervised contrastive learning and classical cross entropy based supervised learning. However, despite this improvement on in-domain downstream tasks, Islam et al. (2021) found that such improvement in transfer learning is limited and even negative for such supervised contrastive learning. This phenomenon motivates us to rethink the role of labeled data in the contrastive learning framework.

In this paper, we first compare contrastive learning with a representative method in the generative approach – the autoencoders. Specifically, we initialize the investigation in the linear representation setting, which has been widely adopted in theory to shed light upon complex machine learning phenomena such as in Du et al. (2020); Tripuraneni et al. (2021). We provide a theoretical analysis of their feature learning performances on the spiked covariance model (Bai & Yao, 2012; Yao et al., 2015; Zhang et al., 2018) and theoretically justify why contrastive learning outperforms autoencoders—contrastive learning is able to remove more noises by constructing contrastive samples. Then we investigate the role of label information in the contrastive learning framework and provide a theoretical justification of why labeled data help to gain accuracy in same-domain classification while can hurt multi-task transfer learning.

**Related works** The idea of contrastive learning was first proposed in Hadsell et al. (2006) as an effective method to perform dimension reduction. Following this line of research, Dosovitskiy et al. (2014) proposed to perform instance discrimination by creating surrogate classes for each instance and Wu et al. (2018) further proposed to preserve a memory bank as a dictionary of negative samples. Other extensions based on this memory bank approach include He et al. (2020); Misra & Maaten (2020); Tian et al. (2020); Chen et al. (2020c). Rather than keeping a costly memory bank, another line of works exploits the benefit of mini-batch training where different samples are treated as negative to each other Ye et al. (2019); Chen et al. (2020a). Moreover, Khosla et al. (2020) explores the supervised version of contrastive learning where pairs are generated based on label information.

Despite its success in practice, theoretical understanding of contrastive learning is still limited. Previous works provide provable guarantees for contrastive learning under conditional independence assumption (or its variants) (Arora et al., 2019; Lee et al., 2020; Tosh et al., 2021; Tsai et al., 2020). Specifically, they assume the two contrastive views are independent conditioned on the label and show that contrastive learning can provably learn representations beneficial for downstream tasks. In addition to this line of research, Wang & Isola (2020); Graf et al. (2021) investigated the representation geometry of supervised contrastive loss, and HaoChen et al. (2021) provided analysis via a novel concept of augmentation graph with a new loss function that performs spectral decomposition on such graph. Moreover, Wen & Li (2021) considered the representation learning under the sparse coding model and studied the optimization properties on shallow ReLU neural networks. Different from all previous works, which aim to show that contrastive learning can learn useful representation, our paper aim to explain why contrastive learning outperforms other representation learning methods and also shed light on the role of labeled data in contrastive learning framework, which is under-explored in prior works.

## 2 PRELIMINARIES

**Notations** In this paper, we use $O, \Omega, \Theta$ to hide universal constants and we write $a_k \lesssim b_k$ for two sequences of positive numbers $\{a_k\}$ and $\{b_k\}$ if and only if there exists an universal constant $C > 0$ such that $a_k < Cb_k$ for any $k$. We use $\|\cdot\|, \|\cdot\|_2, \|\cdot\|_F$ to represent the $\ell_2$ norm of vectors, spectral norm of matrices and Frobenius norm of matrices respectively. Let $\mathbb{O}_{d,r}$ be a set of $d \times r$ orthogonal matrices. i.e., $\mathbb{O}_{d,r} \triangleq \{U \in \mathbb{R}^{d \times r} : U^\top U = I_r\}$. We use $|A|$ to denote the cardinality of a set $A$. For any $n \in \mathbb{N}^+$, let $[n] = \{1, 2, \cdots, n\}$. We use $\|\sin\Theta(U_1, U_2)\|_F$ to refer to the sine distance between two orthogonal matrices $U_1, U_2 \in \mathbb{O}_{d,r}$, which is defined by: $\|\sin\Theta(U_1, U_2)\|_F \triangleq \|U_{1\perp}^\top U_2\|_F$. More properties of sine distance can be found in Section A.1. We use $\{e_i\}_{i=1}^d$ to denote the canonical basis in $d$-dimensional Euclidean space $\mathbb{R}^d$, that is, $e_i$ is the vector whose $i$-th coordinate is 1 and all the other coordinates are 0. Let $\mathbb{I}\{A\}$ be an indicator function that takes 1 when $A$ is true, otherwise takes 0. We write $a \vee b$ and $a \wedge b$ to denote $\max(a, b)$ and $\min(a, b)$, respectively.

### 2.1 SETUP

Given an input $x \in \mathbb{R}^d$, contrastive learning aims to learn a low dimensional representation $h = f(x; \theta) \in \mathbb{R}^r$ by contrasting different samples, i.e., maximizing the agreement between positive pairs, and minimizing the agreement between negative pairs. Suppose we have $n$ data points $X = [x_1, x_2, \cdots, x_n] \in \mathbb{R}^{d \times n}$ from the population distribution $\mathcal{D}$. The contrastive learning task can be formulated to be an optimization problem:

$$\min_\theta \mathcal{L}(\theta) = \min_\theta \frac{1}{n} \sum_{i=1}^n \ell(x_i, \mathcal{B}_i^{Pos}, \mathcal{B}_i^{Neg}; f(\cdot, \theta)) + \lambda R(\theta), \tag{2.1}$$

where $\ell(\cdot)$ is a contrastive loss and $\lambda R(\theta)$ is a regularization term; $\mathcal{B}_i^{Pos}, \mathcal{B}_i^{Neg}$ are the sets of positive samples and negative samples corresponding to $x_i$, which we will describe in detail below.

**Losses and Models.** We then present the model setup considered in this paper.

*(a). Linear representation and regularization term.* We consider the linear representation function $f(x, W) = Wx$, where the parameter $\theta$ is a matrix $W \in \mathbb{R}^{r \times d}$. Since regularization techniques

have been widely adapted in contrastive learning practice (Chen et al., 2020a; He et al., 2020; Grill et al., 2020), we further consider penalizing the representation by a regularization term $R(W) = \|WW^\top\|_F^2/2$ to encourage the orthogonality of $W$ and therefore promote the diversity of $w_i$ to learn different representations.

*(b). Triplet Contrastive loss.* The contrastive loss is set to be the average similarity between positive pairs minus that between negative pairs:

$$\ell(x, \mathcal{B}^{Pos}, \mathcal{B}^{Neg}, f(\cdot, \theta)) = -\sum_{x^{Pos} \in \mathcal{B}^{Pos}} \frac{\langle f(x, \theta), f(x^{Pos}, \theta) \rangle}{|\mathcal{B}^{Pos}|} + \sum_{x^{Neg} \in \mathcal{B}^{Neg}} \frac{\langle f(x, \theta), f(x^{Neg}, \theta) \rangle}{|\mathcal{B}^{Neg}|},$$
(2.2)

where $\mathcal{B}^{Pos}, \mathcal{B}^{neg}$ are sets of positive samples and negative samples corresponding to $x$. This loss has been commonly used in constrastive learning (Hadsell et al., 2006) and metric learning (Schroff et al., 2015; He et al., 2018). In Khosla et al. (2020), the authors show that it is an approximation of the NT-Xent contrastive loss, which has been highlighted in recent contrastive learning practice (Sohn, 2016; Wu et al., 2018; Oord et al., 2018; Chen et al., 2020a).

*(c). Generation of positive and negative pairs.* There are two common approaches to generate such pairs, depending on whether or not label information is available. When the label information is not available, the typical strategy is to generate different views of the original data via augmentation (Hadsell et al., 2006; Chen et al., 2020a). Two views of the same data point serve as positive pair for each other while those of different data serve as negative pairs.

**Definition 2.1** (Augmented pairs generation). Given two augmentation functions $g_1, g_2 : \mathbb{R}^d \to \mathbb{R}^d$ and $n$ training samples $\mathcal{B} = \{x_i\}_{i \in [n]}$, the augmented views are given by: $\{(g_1(x_i), g_2(x_i))\}_{i \in [n]}$. Then for each view $g_v(x_i)$, $v = 1, 2$, the corresponding positive samples and negative samples are defined by: $\mathcal{B}_{i,v}^{Pos} = \{g_s(x_i) : s \in [2] \setminus \{v\}\}$ and $\mathcal{B}_{i,v}^{Neg} = \{g_s(x_j) : s \in [2], j \in [n] \setminus \{i\}\}$.

The loss function of self-supervised contrastive learning problem can be written as:

$$\mathcal{L}_{\text{SelfCon}}(W) = -\frac{1}{2n} \sum_{i=1}^n \sum_{v=1}^2 \left[ \langle Wg_v(x_i), Wg_{[2]\setminus\{v\}}(x_i) \rangle - \sum_{j \neq i} \sum_{s=1}^2 \frac{\langle Wg_v(x_i), Wg_s(x_j) \rangle}{2n-2} \right] + \frac{\lambda}{2} \|WW^\top\|_F^2.$$
(2.3)

In particular, we adopt the following augmentation in our analysis.

**Definition 2.2** (Random masking augmentation). The two views of the original data are generated by randomly dividing its dimensions to two sets, that is, $g_1(x_i) = Ax_i$, and $g_2(x_i) = (I - A)x_i$, where $A = \text{diag}(a_1, \cdots, a_d) \in \mathbb{R}^{d \times d}$ is the diagonal masking matrix with $\{a_i\}_{i=1}^d$ being *i.i.d.* random variables sampled from a Bernoulli distribution with mean $1/2$.

A similar augmentation was considered in Wen & Li (2021). However, our primary interest lies in comparing the performance of contrastive learning against autoencoders and analyzing the role of labeled data, while their work focuses on understanding the training process of neural networks in contrastive learning.

When the label information is available, Khosla et al. (2020) proposed the following approach to generate pairs.

**Definition 2.3** (Supervised pairs generation). In a $K$-class classification problem, given $n_k$ samples for each class $k \in [K]$: $\{x_i^k : i \in [n_k]\}_{k=1}^K$ and let $n = \sum_{k=1}^K n_k$, the corresponding positive samples and negative samples for $x_i^k$ are defined by $\mathcal{B}_{i,k}^{Pos} = \{x_j^k : j \in [n_k] \setminus i\}$ and $\mathcal{B}_{i,k}^{Neg} = \{x_j^s : s \in [K] \setminus k, j \in [n_s]\}$. That is, the positive samples are the remaining ones in the same class with $x_i^k$ and the negative samples are the samples from different classes.

Correspondingly, the loss function of the supervised contrastive learning problem can be written as:

$$\mathcal{L}_{\text{SupCon}}(W) = -\frac{1}{nK} \sum_{k=1}^K \sum_{i=1}^n \left[ \sum_{j \neq i} \frac{\langle Wx_i^k, Wx_j^k \rangle}{n-1} - \sum_{j=1}^n \sum_{s \neq k} \frac{\langle Wx_i^k, Wx_j^s \rangle}{n(K-1)} \right] + \frac{\lambda}{2} \|WW^\top\|_F^2. \quad (2.4)$$

*(d). Spiked Covariance Model.* We consider the following spiked covariance model (Bai & Yao, 2012; Yao et al., 2015; Zhang et al., 2018) to study the power of contrastive learning:

$$x = U^\star z + \xi, \quad \text{Cov}(z) = \nu^2 I_r, \quad \text{Cov}(\xi) = \Sigma, \quad (2.5)$$

where $z \in \mathbb{R}^r$ and $\xi \in \mathbb{R}^d$ are both zero mean sub-Gaussian random variables. In particular, $U^\star \in \mathbb{O}_{d,r}$ and $\Sigma = \mathrm{diag}(\sigma_1^2, \cdots, \sigma_d^2)$. The first term $U^\star z$ represents the signal of interest residing in a low-dimensional subspace spanned by the columns of $U^\star$. The second term $\xi$ is the dense noise with heteroskedastic noise. Given that, the ideal low-dimensional representation is to compress the observed $x$ into a low-dimensional representation spanned by the columns of $U^\star$.

In this paper, ***we aim to learn a good projection*** $W \in \mathbb{R}^{r \times d}$ ***onto a lower-dimensional subspace from observation*** $x$. Since the information of $W$ is invariant with the transformation $W \leftarrow OW$ for any invertible matrix $O \in \mathbb{R}_{r,r}$, the essential information of $W$ is contained in the right eigenvector of $W$. Thus we quantify the goodness of the representation $W$ by the sine distance $\|\sin\Theta(U, U^\star)\|_F$, where $U$ is the top-$r$ right eigenspace of $W$.

## 3 SELF-SUPERVISED CONTRASTIVE LEARNING VERSUS AUTOENCODER

Autoencoder and contrastive learning are two popular approaches for self-supervised learning. Recent experiments have highlighted the improved performance of contrastive learning compared with autoencoders. In this section, we rigorously demonstrate the advantage of contrastive learning over autoencoder by investigating the linear representation settings under the spiked covariance model Eq.(2.5). The investigation is conducted for both feature recovery and downstream tasks.

### 3.1 RECOVER FEATURES FROM NOISY DATA

Here we focus on the analysis of feature recovery to understand the benefit of contrastive learning over autoencoders. As mentioned above, our target is to recover the subspace spanned by the columns of $U^\star$, which can further help us obtain information on the unobserved $z$ that is important for downstream tasks. However, the observed data has covariance matrix of $\nu^2 U^\star U^{\star\top} + \Sigma$ rather than the desired $\nu^2 U^\star U^{\star\top}$, which brings difficulty to representation learning. We demonstrate that contrastive learning can better exploit the structures of core features and obtain better estimation than an autoencoder in this setting.

We start with autoencoders. Formally, an autoencoder consists of an encoder $f^{AE} : \mathbb{R}^d \to \mathbb{R}^r$ and a decoder $g^{DE} : \mathbb{R}^r \to \mathbb{R}^d$. While the encoder compresses the original data into low dimensional features, and the decoder recovers the original data from those features. It can be formalized as the following optimization problem for samples $\{x_i\}_{i=1}^n$ (Ballard, 1987; Fan et al., 2019):

$$\min_{f^{AE}, g^{AE}} \frac{1}{n} \sum_{i=1}^n \|x_i - g^{DE}(f^{AE}(x_i))\|_2^2.$$

In the linear representation setting, where $f^{AE}(x) = W_{AE} x + b_{AE}$ and $g^{DE}(y) = W_{DE} y + b_{DE}$, previous works (Bourlard & Kamp, 1988; Plaut, 2018) have shown that the autoencoder can be reduced to principal component analysis (PCA). That is, the optimal $W_{AE}$ is given by:

$$W_{AE} = (U_{AE} \Sigma_{AE} V_{AE}^\top)^\top, \tag{3.1}$$

where $U_{AE}$ is the top-$r$ eigenvectors of matrix $M := X(I - 1_n 1_n^\top/n)X^\top$, $\Sigma_{AE}$ is a diagonal matrix consists of eigenvalues of $M$ and $V_{AE} = [v_1, \cdots, v_n] \in \mathbb{R}^{r \times r}$ can be any orthonormal matrix. In the noiseless case, the covariance matrix is $\nu^2 U^\star U^{\star\top}$ and autoencoder can ***perfectly*** recover the core features. However, in noisy cases, the random noises sometimes perturb the core features, which make autoencoders fail to learn core features. Such noisy cases are very common in real applications such as measurement errors and backgrounds in images such as grasses and sky. Interestingly, we will later show that contrastive learning can better recover $U^\star$ despite the presence of large noise.

To provide rigorous analysis, we first introduce the incoherent constant (Candès & Recht, 2009).

**Definition 3.1** (Incoherent constant). We define the incoherence constant of $U \in \mathbb{O}_{d,r}$ as

$$I(U) = \max_{i \in [d]} \|e_i^\top U\|^2. \tag{3.2}$$

Intuitively, the incoherent constant measures the degree of the incoherence of the distribution of entries among different coordinates, or loosely speaking, the similarity between $U$ and canonical

basis $\{e_i\}_{i=1}^d$. For uncorrelated random noise, the covariance matrix is diagonal and its eigenspace is exactly spanned by the canonical basis $\{e_i\}_{i=1}^d$ (if the diagonal entries in $\Sigma$ are all different), which attains the maximum value of incoherent constant. On the contrary, core features usually exhibit certain correlation structures and the corresponding eigenspace of the covariance matrix is expected to have a lower incoherent constant.

We then introduce a few assumptions where our theoretical results are built on. Recall that in the spiked covariance model Eq.(2.5), $x = U^\star z + \xi$, $\mathrm{Cov}(z) = \nu^2 I_r$ and $\mathrm{Cov}(\xi) = \mathrm{diag}(\sigma_1^2, \cdots, \sigma_d^2)$.

*Assumption* 3.1 (Regular covariance condition). The condition number of covariance matrix $\Sigma = \mathrm{diag}(\sigma_1^2, \cdots, \sigma_d^2)$ satisfies $\kappa := \sigma_{(1)}^2 / \sigma_{(d)}^2 < C$, where $\sigma_{(j)}^2$ represents the $j$-th largest number among $\sigma_1^2, \cdots, \sigma_d^2$ and $C > 0$ is a universal constant.

*Assumption* 3.2 (Signal to noise ratio condition). Define the signal to noise ratio $\rho := v/\sigma_{(1)}$, we assume $\rho = \Theta(1)$, implying that the covariance of noise is of the same order as that of core features.

*Assumption* 3.3 (Incoherent condition). The incoherent constant of the core feature matrix $U^\star \in \mathbb{O}_{d,r}$ satisfies $I(U^\star) = O(r \log d/d)$. [1]

*Remark* 3.1. Assumption 3.1 implies that the variances of all dimensions are of the same order. For Assumption 3.2, we focus on a large-noise regime where the noise may hurt the estimation significantly. Here we assume the ratio lies in a constant range, but our theory can easily adapt to the case where $\rho$ has a decreasing order. Assumption 3.3 implies a stronger correlation among coordinates of core features, which is the essential property to distinguish them from random noise.

Now we are ready to present our first result, showing that the autoencoder is unable to recover the core features in the large-noise regime.

**Theorem 3.1** (Recovery ability of autoencoder, lower bound). *Consider the spiked covariance model Eq.(2.5), under Assumption 3.1-3.3 and $n > d \gg r$, let $W_{AE}$ be the learned representation of autoencoder with singular value decomposition $W_{AE} = (U_{AE}\Sigma_{AE}V_{AE}^\top)^\top$ (as in Eq.(3.1)). If we further assume $\{\sigma_i^2\}_{i=1}^d$ are different from each other and $\sigma_{(1)}^2/(\sigma_{(r)}^2 - \sigma_{(r+1)}^2) < C_\sigma$ for some universal constant $C_\sigma$. Then there exist two universal constants $C_\rho > 0, c \in (0,1)$, such that when $\rho < C_\rho$, we have*

$$\mathbb{E} \|\sin\Theta(U^\star, U_{AE})\|_F \geq c\sqrt{r}. \tag{3.3}$$

*Remark* 3.2. The additional assumptions $\{\sigma_i^2\}_{i=1}^d$ are different from each other and $\sigma_{(1)}^2/(\sigma_{(r)}^2 - \sigma_{(r+1)}^2) < C_\sigma$ for some universal constant $C_\sigma$ are for technical consideration. We need these conditions to guarantee the uniqueness of $U_{AE}$. As an extreme example, the top-$r$ eigenspace of identity matrix can be any $r$-dimensional subspace and thus not unique. To avoid discussing such arbitrariness of output, we make these assumptions to guarantee the separability of eigenspace.

Then we investigate the feature recovery ability of the self-supervised contrastive learning approach.

**Theorem 3.2** (Recovery ability of contrastive learning, upper bound). *Under the spiked covariance model Eq.(2.5), random masking augmentation in Definition 2.2, Assumption 3.1-3.3 and $n > d \gg r$, let $W_{CL}$ be any solution that minimizes Eq.(2.3), and denote its singular value decomposition as $W_{CL} = (U_{CL}\Sigma_{CL}V_{CL}^\top)^\top$, then we have*

$$\mathbb{E} \|\sin\Theta(U^\star, U_{CL})\|_F \lesssim \frac{r^{3/2}}{d}\log d + \sqrt{\frac{dr}{n}}. \tag{3.4}$$

*Remark* 3.3. In Eq.(3.4), the first term is due to the shift between the distributions of the augmented data and the original data. Specifically, the random masking augmentation generates two views with disjoint non-zero coordinates and thus can mitigate the influence of random noise on the diagonal entries in the covariance matrix. However, such augmentation slightly hurts the estimation of core features. This bias, appearing as the first term in Eq.(3.4), is measured by the incoherent constant Eq.(3.2). The second term corresponds to the estimation error of the population covariance matrix.

Theorem 3.1 and 3.2 characterize the difference of feature recovery ability between autoencoder and contrastive learning. The autoencoder fails to recover most of the core features in the large-noise

---

[1] The order of $I(U^\star)$ can be chosen to be any function that decreasing to 0 when $d \to \infty$ and one can easily adapt the later results to this setting. Here we set it to $O(r \log d/d)$, the order when $U$ is drawn from a uniform distribution on $\mathbb{O}_{d,r}$ (see the proof in Lemma B.1) for simplicity to obtain an exact order in later results.

regime, since $\|\sin\Theta(U, U^\star)\|_F$ has a trivial upper bound $\sqrt{r}$. In contrast, with the help of data augmentation, the contrastive learning approach mitigates the corruption of random noise while preserving core features. As $n$ and $d$ increase, it yields a consistent estimator of core features and further leads to better performance in the downstream tasks, as shown in the next section.

## 3.2 PERFORMANCE ON THE DOWNSTREAM TASK

In the previous section, we have seen that contrastive learning can recover the core feature effectively. In practice, we are interested in using the learned features to downstream tasks. He et al. (2020) experimentally showed the overwhelming performance of linear classifiers trained on representations learned with contrastive learning against several supervised learning methods in downstream tasks.

Following the recent empirical studies, here we evaluate the downstream performance of simple predictors, which take a linear transformation of representation as an input. Specifically, we consider the regression setting with a class of predictors $\delta_{W,w}(x) = w^\top W x$ constructed upon the learned representations $W = W_{CL}$ and $W_{AE}$ respectively. Given the observation $\check{x} = U^\star \check{z} + \check{\xi}$ independent of unsupervised data $X$, our prediction target is $\check{y}$ generated from a linear model $\check{y} = \langle \check{z}, w^\star \rangle / \nu + \check{\epsilon}$, where $\check{z} \sim N(0, \nu^2 I_r)$ is the low-dimensional core feature, $\nu$ is the scale of $\check{z}$ and $\check{\epsilon}$ is the error term independent of $\check{z}$ with zero mean and finite variance. $w^\star \in \mathbb{R}^r$ is a unit vector of coefficients. We can interpret this model as a principal component regression model (PCR) (Jolliffe, 1982) under standard error-in-variables settings[2], where we assume that coefficients lie in a low-dimensional subspace spanned by column vectors of $U^\star$. We either estimate or predict the signal based on observed samples contaminated by the measurement error $\check{\xi}$. For details of PCR in error-in-variables settings, see, for example, Ćevid et al. (2020); Agarwal et al. (2020); Bing et al. (2021).

Now we state our result on the downstream performance in prediction task. For any linear representation $W \in \mathbb{R}^{r \times d}$, let $\mathcal{R}(W) := \mathbb{E}_X[\inf_{w \in \mathbb{R}^r} \mathbb{E}_{\check{y}, \check{x}}[\ell(\delta_{W,w})]]$ be the risk of the best linear predictor, where $\ell(\delta) = (\check{y} - \delta(\check{x}))^2$.

**Theorem 3.3** (Upper Bound for Downstream Excess Risk of $W_{CL}$). *Suppose conditions in Theorem 3.2 are satisfied. Then we have*

$$\mathcal{R}(W_{CL}) - \mathcal{R}(U^{\star\top}) \lesssim \frac{r^{3/2}}{d} \log d + \sqrt{\frac{dr}{n}}.$$

This result shows that the price of estimating $U^\star$ by contrastive learning on a downstream prediction task can be made small in a case where the core feature lies in a relatively low-dimensional subspace, and the number of samples is relatively large compared to the ostensible dimension of data.

However, the downstream performance of autoencoders is not as good as contrastive learning. We obtain the following lower bound for the downstream prediction risk with the autoencoder.

**Theorem 3.4** (Lower Bound for Downstream Excess Risk of $W_{AE}$). *Suppose the conditions in Theorem 3.1 hold. Assume $r \leq r_c$ holds for some constant $r_c > 0$. Additionally assume that $\rho = \Theta(1)$ is sufficiently small and $n \gg d \gg r$. Then,*

$$\mathcal{R}(W_{AE}) - \mathcal{R}(U^{\star\top}) \geq c',$$

*where $c' > 0$ is a constant independent of $n$ and $d$.*

Comparing theorems 3.3 and 3.4, we find that even when $r/d$ and $\sqrt{d/n}$ are small, the downstream performance of autoencoders is not satisfying and has a much larger downstream task error rate than that of contrastive learning. We note that the constant lower bound of Theorem 3.4 can be obtained without the assumption $r \leq r_c$ by assuming a slightly stronger conditions $\rho = O(1/\log d)$ and $n \gg dr$. We also illustrate this phenomenon via numerical simulation in Fig.1. As predicted by Theorem 3.1 and 3.2, the downstream task risk of contrastive learning decreases as $d$ increases (Fig. 1: **Left**) and as $n$ increases (Fig. 1: **Center**) while that of autoencoder remains large when $n$ and $d$ increase.

---

[2]In error-in-variables settings, the bias term from the measurement error appears in prediction and estimation risk. Since our focus lies in proving a better performance of contrastive learning against autoencoders, we ignore the unavoidable bias term here by considering the excess risk.

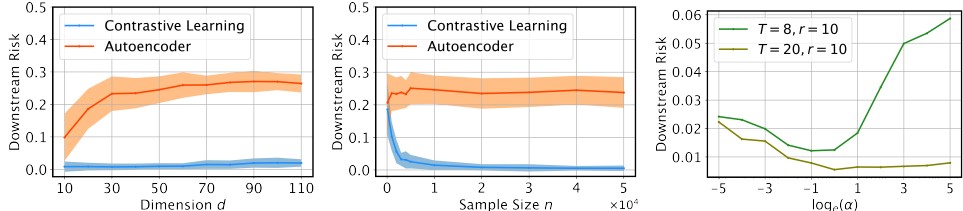

Figure 1: The vertical axes indicate the prediction risk. **Left:** Comparison of downstreak task performance between contrastive learning and autoencoders the dimension $d$. The sample size $n$ is set as $n = 20000$. **Center:** Comparison of downstreak task performance between contrastive learning and autoencoders the dimension $n$. The dimension $d$ is set as $d = 40$. **Right:** Downstream task performance in transfer learning against penalty parameter $\alpha$ in log scale. $T$ is the number of source tasks and $r$ is the dimension of the representation function. We set the number of labeled data and unlabeled data as $m = 1000$ and $n = 1000$ respectively.

*Remark* 3.4. A similar results of upper and lower bound hold for linear predictors $\delta(x) = \mathbb{I}\{F(w^\top W x) > 1/2\}$ under binary classification settings, where label is generated from a binary response model $\check{y}|\check{z} = \text{Ber}(F(\langle\check{z}, w^\star\rangle/\nu))$ with a known function $F : \mathbb{R} \to [0, 1]$. Our results imply that the learned representations by contrastive learning are also useful in downstream classification tasks, compared to an autoencoder, thus supporting the empirical success of contrastive learning. A detailed results and proofs are deferred to Appendix B.2.

## 4 THE IMPACT OF LABELED DATA IN SUPERVISED CONTRASTIVE LEARNING

Recent works have explored adding label information to improve contrastive learning (Khosla et al., 2020). Empirical results show that label information can significantly improve the accuracy of the in-domain downstream tasks. However, when domain shift (multiple sources) is considered, the label information hardly improves and even hurts transferability (Islam et al., 2021). Motivated by those empirical observations, in this section, we aim to investigate the role of labeled data in contrastive learning and provide a theoretical foundation for such phenomena.

### 4.1 FEATURE MINING IN MULTI-CLASS CLASSIFICATION

We first demonstrate the role of label in the single-sourced case in contrastive learning. Suppose our samples are drawn from $r + 1$ different classes with probability $p_k$ for class $k \in [r + 1]$, and $\sum_{k=1}^{r+1} p_k = 1$. For each class, samples are generated from a class-specific Gaussian distribution:

$$x^k = \mu^k + \xi^k, \quad \xi^k \sim \mathcal{N}(0, \Sigma^k), \quad \forall k = 1, 2, \cdots, r + 1. \tag{4.1}$$

To be consistent with the spiked covariance model Eq.(2.5), we assume $\|\mu^k\| = \sqrt{r}\nu, \forall k \in [r+1]$. We further assume $\Sigma^k = \text{diag}(\sigma_{1,k}^2, \cdots, \sigma_{d,k}^2)$, denote $\sigma_{(1)}^2 = \max_{1 \le i \le d, 1 \le j \le r+1} \sigma_{i,j}^2$ and assume $\sum_{k=1}^{r+1} p_k \mu_k = 0$, where the last assumption is added to ensure identifiable since the classification problem (4.1) is invariant under translation. Denote $\Lambda = \sum_{k=1}^{r+1} p_k \mu_k \mu_k^\top$, we assume $\text{rank}(\Lambda) = r$ and $C_1 \nu^2 < \lambda_{(r)}(\Lambda) < \lambda_{(1)}(\Lambda) < C_2 \nu^2$ for two universal constants $C_1$ and $C_2$. We remark that this model is a labeled version of the spiked covariance model Eq.(2.5) since the core features and random noise are both sub-Gaussian. We use $r + 1$ classes to ensure that $\mu_k$'s span an $r$-dimensional space, and denote its orthonormal basis as $U^\star$. Recall that our target is to recover $U^\star$.

As introduced in Definition 2.3, Khosla et al. (2020) proposed a novel approach named supervised contrastive learning, which allows us to discriminate instances across classes. When we have both labeled data and unlabeled data, we can perform contrastive learning based on pairs that are generated separately for the two types of data.

***Data Generation Process.*** Formally, let us consider the case in which we draw $n$ samples as unlabeled data $X = [x_1, \cdots, x_n] \in \mathbb{R}^{d \times n}$ from the Gaussian mixture model Eq.(4.1) with $p_1 = p_2 = \cdots = p_{r+1}$. For the labeled data, we draw $(r + 1)m$ samples, i.e., $m$ samples for each of the $r + 1$ classes in the Gaussian mixture model, and denote them as $\hat{X} = [\hat{x}_1, \cdots, \hat{x}_{(r+1)m}] \in \mathbb{R}^{d \times (r+1)m}$. We discuss the above case for simplicity. More general versions are considered in Theorem C.2 (in

the appendix). We study the following hybrid loss to illustrate how label information helps promote the performance over the self-supervised contrastive learning:

$$\min_{W \in \mathbb{R}^{r \times d}} \mathcal{L}(W) := \min_{W \in \mathbb{R}^{r \times d}} \mathcal{L}_{\text{SelfCon}}(W) + \alpha \mathcal{L}_{\text{SupCon}}(W), \tag{4.2}$$

where $\alpha > 0$ is the ratio between supervised loss and self-supervised contrastive loss.

We first provide a high-level explanation of why label information help learn core features. When the label information is unavailable, no matter how much (unlabeled) data we have, we can only take themselves (and their augmented views) as positive samples. In such a scenario, performing augmentation leads to an unavoidable trade-off between estimation bias and accuracy. However, if we have additional class information, we can contrast between data in the same class to extract more beneficial features that help distinguish a particular class from others and therefore reduce the bias.

**Theorem 4.1.** *Suppose the labeled and unlabeled samples are generated as the process mentioned above. If Assumption 3.1-3.3 hold, $n > d \gg r$ and let $W_{CL}$ be any solution that minimizes the supervised contrastive learning problem in Eq.(4.2), and denote its singular value decomposition as $W_{CL} = (U_{CL} \Sigma_{CL} V_{CL}^\top)^\top$, then we have*

$$\mathbb{E} \| \sin \Theta(U_{CL}, U^\star) \|_F \lesssim \frac{1}{1+\alpha} \Big( \frac{r^{3/2}}{d} \log d + \sqrt{\frac{dr}{n}} \Big) + \frac{\alpha}{1+\alpha} \sqrt{\frac{dr}{m}}.$$

*Moreover, if we have $m$ labeled data for each class and no unlabeled data, then*

$$\mathbb{E} \| \sin \Theta(U_{CL}, U) \|_F \lesssim \sqrt{\frac{dr}{m}}.$$

The first bound in Theorem 4.1 demonstrates how the effect of labeled data changes against the ratio $\alpha$ in the hybrid loss in Eq.(4.2). In addition, compared with Theorem 3.2, when we only have labeled data ($\alpha \to \infty$), the second bound in Theorem 4.1 indicates that with labeled data being available, the supervised contrastive learning can yield consistent estimation as $m \to \infty$ while the self-supervised contrastive learning consists of an irreducible bias term $O(r^{3/2} \log d/d)$. At a high level, label information help gain accuracy by creating more positive samples for a single anchor and therefore extract more decisive features. One should notice a caveat that when labeled data is extremely rare compared with unlabeled data, the estimation of supervised contrastive learning suffers from high variance. In comparison, self-supervised contrastive learning, which can exploit a much larger number of samples, may outperform it.

## 4.2 INFORMATION FILTERING IN MULTI-TASK TRANSFER LEARNING

Label information can tell us the beneficial information for the downstream task, and learning with labeled data will filter out useless information and preserve the decisive parts of core features. However, in transfer learning, the label information is sometimes found to hurt the performance of contrastive learning (Khosla et al., 2020; Islam et al., 2021). In this section, we consider two regimes of transfer learning – tasks are insufficient/abundant. In both regimes, we provide theories to support the empirical observations and further demonstrate how to wisely combine the supervised and self-supervised contrastive learning to avoid those harms and reach better performance. Specifically, we consider a transfer learning problem with regression settings. Suppose we have $T$ source tasks which share a common data generating model Eq.(2.5). In order to study the case of transfer learning, the labels are generated in a different way, that is, for the $t$-th task, the labels are generated by $y^t = \langle w_t, z \rangle / \nu$, where $w_t \in \mathbb{R}^r$ is a unit vector and different across tasks.

To incorporate label information, we maximize the Hilbert-Schmidt Independence Criteria (HSIC) (Gretton et al., 2005; Barshan et al., 2011), which has been widely used in literature (Song et al., 2007a;b;c; Barshan et al., 2011). HSIC is defined as $\text{HSIC}(X, y; W) = X^\top W^\top W X H y y^\top H / (n - 1)^2$, where $W \in \mathbb{R}^{r \times d}$ is linear representation to be learned and $H = I_n - (1/n) 1_n 1_n^\top$ is the centering matrix. A detailed discussion about its background, motivation and its connection to the mean squared loss is presented in Appendix A.3. Suppose we have $n$ unlabeled data $X = [x_1, \cdots, x_n] \in \mathbb{R}^{d \times n}$ and $m$ labeled data for each source task $\hat{X}^t = [\hat{x}_1^t, \cdots, \hat{x}_m^t], y^t = [y_1^t, \cdots, y_m^t], \forall t = 1, \ldots, T$ where $x_i$'s and $\hat{x}_j^t$'s are independently drawn from spiked covariance model Eq. (2.5), we learn the linear representation via the joint optimization:

$$\min_{W \in \mathbb{R}^{r \times d}} \mathcal{L}(W) := \min_{W \in \mathbb{R}^{r \times d}} \mathcal{L}_{\text{SelfCon}}(W) - \alpha \sum_{t=1}^{T} \text{HSIC}(\hat{X}^t, y^t; W), \tag{4.3}$$

where $\alpha > 0$ is a pre-specified ratio between the self-supervised contrastive loss and HSIC. (A more general setting is considered in the appendix, see Section C.2 for details.) We now present a theorem showing the recoverability of $W$ by minimizing the hybrid loss function (4.3).

**Theorem 4.2.** *Suppose Assumption 3.1-3.3 hold for spiked covariance model Eq.(2.5) and $n > d \gg r$, if we further assume that $\alpha > C$ for some constant $C$, $T < r$ and $w_t$'s are orthogonal to each other, and let $W^{CL}$ be any solution that optimizes the problem in Eq.(4.3), and denote its singular value decomposition as $W_{CL} = (U_{CL}\Sigma_{CL}V_{CL}^\top)^\top$, then we have:*

$$\mathbb{E}\|\sin\Theta(U_{CL}, U^\star)\|_F \lesssim \sqrt{r-T}\Big(\frac{r\log d}{d} + \sqrt{\frac{d}{n}} + \alpha T\sqrt{\frac{d}{m}} \wedge 1\Big) + \sqrt{T}\Big(\frac{r\log d}{\alpha d} + \frac{1}{\alpha}\sqrt{\frac{d}{n}} + T\sqrt{\frac{d}{m}}\Big). \tag{4.4}$$

In Theorem 4.2, as $\alpha$ goes to infinity (corresponding to the case where we only use the supervised loss), Eq.(4.4) is reduced to $\sqrt{r-T} + T^{3/2}\sqrt{d/m}$, which is worse than the $r^{3/2}\log d/d$ rate obtained by self-supervised contrastive learning (Theorem 3.2). This implies that when the model focuses mainly on the supervised loss, the algorithm will extract the information only beneficial for the source tasks and fail to estimate other parts of core features. As a result, when the target task has a very different distribution, labeled data will bring extra bias and therefore hurt the transferability. Additionally, one can minimize the right-hand side of Eq.(4.4) to obtain a sharper rate. Specifically, we can choose an appropriate $\alpha$ such that the upper bound becomes $\sqrt{r^2(r-T)}\log d/d$ (when $n, m \to \infty$), obtaining a smaller rate than that of the self-supervised contrastive learning. These facts provide theoretical foundations for the recent empirical observations that smartly combining supervised and self-supervised contrastive learning achieves significant improvement on transferability compared with performing each of them individually (Islam et al., 2021).

When the tasks are abundant enough then estimation via labeled data can recover core features completely. Similar to Theorem 4.2, we have the following result.

**Theorem 4.3.** *Suppose Assumptions 3.1-3.3 hold for spiked covariance model Eq.(2.5) and $n > d \gg r$, if we further assume that $T > r$ and $\lambda_{(r)}(\sum_{i=1}^T w_i w_i^\top) > c$ for some constant $c > 0$, suppose $W^{CL}$ is the optimal solution of optimization problem eq.(4.3), and denote its singular value decomposition as $W_{CL} = (U_{CL}\Sigma_{CL}V_{CL}^\top)^\top$, then we have:*

$$\mathbb{E}\|\sin\Theta(U_{CL}, U^\star)\|_F \lesssim \frac{\sqrt{r}}{\alpha+1}\Big(\frac{r}{d}\log d + \sqrt{\frac{d}{n}}\Big) + T\sqrt{\frac{dr}{m}}. \tag{4.5}$$

Similar to Theorem 4.2, Theorem 4.3 shows that in the case where tasks are abundant, as $\alpha$ goes to infinity (corresponding to the case where we use the supervised loss only), Eq.(4.5) is reduced to $T\sqrt{rd/m}$. This rate can be worse than the $\sqrt{r^3}\log d/d + \sqrt{rd/n}$ rate obtained by self-supervised contrastive learning when $m$ is small. Recall that when the number of tasks is small, labeled data introduce extra bias term $\sqrt{r-T}$ (Theorem 4.2). We note that when the tasks are abundant enough, the harm of labeled data is mainly due to the variance brought by the labeled data. When $m$ is sufficiently large, supervised learning on source tasks can yield consistent estimation of core features, whereas self-supervised contrastive learning can not. We also illustrate the different behaviors of the two regimes via numerical simulations in Fig. 1 right panel. Consistent with our theory, it is observed that when tasks are not abundant, the transfer performance exhibit a $U$-shaped curve, and the best result is achieved by choosing an appropriate $\alpha$. When tasks are abundant and labeled data are sufficient, the error remains small when we take large $\alpha$.

## 5   CONCLUSION

In this work, we theoretically prove that contrastive learning, compared with autoencoders, can obtain a better low-rank representation under the spiked covariance model, which further leads to better performance in downstream tasks. We also highlight the role of labeled data in supervised contrastive learning and multi-task transfer learning: labeled data can reduce the domain shift bias in contrastive learning, but it harms the learned representation in transfer learning. To our knowledge, our result is the first theoretical result to guarantee the success of contrastive learning by comparing it with existing representation learning methods. However, in order to get tractable analysis, like many other theoretical works in representation learning (Du et al., 2020; Lee et al., 2020; Tripuraneni et al., 2021), our work starts with linear representations, which still provides important insights. Extending the results to more complex models is an interesting direction of future work.

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

# A BACKGROUND

## A.1 DISTANCE BETWEEN SUBSPACES

In this section, we will provide some basic properties of $\sin\Theta$ distance between subspaces. Recall that the definition is:

$$\|\sin\Theta(U_1, U_2)\|_F \triangleq \|U_{1\perp}^\top U_2\|_F = \|U_{2\perp}^\top U_1\|_F. \tag{A.1}$$

where $U_1, U_2 \in \mathbb{O}_{d,r}$ are two orthogonal matrices. Similarly, we can also define:

$$\|\sin\Theta(U_1, U_2)\|_2 \triangleq \|U_{1\perp}^\top U_2\|_2 = \|U_{2\perp}^\top U_1\|_2.$$

We first give two equivalent definition of this distance:

**Proposition A.1.**

$$\|\sin\Theta(U_1, U_2)\|_F^2 = r - \|U_1^\top U_2\|_F^2$$

*Proof.* Write $U = [U_1, U_{1\perp}] \in \mathbb{O}_{d,d}$ we have :

$$r = \|U_2\|_F^2 = \|U^\top U_2\|_F^2 = \|U_{1\perp}^\top U_2\|_F^2 + \|U_1^\top U_2\|_F^2,$$

then by definition of $\sin\Theta$ distance we can obtain the desired equation. $\square$

**Proposition A.2.**

$$\|\sin\Theta(U_1, U_2)\|_F^2 = \frac{1}{2}\|U_1 U_1^\top - U_2 U_2^\top\|_F^2$$

*Proof.* Expand the right hand and use Proposition A.1 we have:

$$\begin{aligned}
\frac{1}{2}\|U_1 U_1^\top - U_2 U_2^\top\|_F^2 &= \frac{1}{2}(\|U_1 U_1^\top\|_F^2 + \|U_2 U_2^\top\|_F^2 - 2\operatorname{tr}(U_1 U_1^\top U_2 U_2^\top)) \\
&= \frac{1}{2}(r + r - 2\operatorname{tr}(U_1^\top U_2 U_2^\top U_1)) \\
&= r - \|U_1^\top U_2\|_F^2 = \|\sin\Theta(U_1, U_2)\|_F^2.
\end{aligned}$$

$\square$

With Proposition A.1 and A.2, it's easy to verify its properties to be a distance function. Obviously, we have $0 \le \|\sin\Theta(U_1, U_2)\|_F \le \sqrt{r}$ and $\|\sin\Theta(U_1, U_2)\|_F = \|\sin\Theta(U_2, U_1)\|_F$ by definition. Moreover, we have the following results:

**Lemma A.1** (Lemma 1 in Cai & Zhang (2018)). *For any $U, V \in \mathbb{O}_{d,r}$,*

$$\|\sin\Theta(U, V)\|_2 \le \inf_{O \in \mathbb{O}_{r,r}} \|UO - V\|_2 \le \sqrt{2}\|\sin\Theta(U, V)\|_2, \tag{A.2}$$

*and*

$$\|\sin\Theta(U, V)\|_F \le \inf_{O \in \mathbb{O}_{r,r}} \|UO - V\|_F \le \sqrt{2}\|\sin\Theta(U, V)\|_F. \tag{A.3}$$

**Proposition A.3** (Identity of indiscernibles).

$$\|\sin\Theta(U_1, U_2)\|_F = 0 \Leftrightarrow \exists O \in \mathbb{O}^{r \times r}, \text{ s.t. } U_1 O = U_2$$

*Proof.* It's a straightforward corollary by definition:

$$\begin{aligned}
\|\sin\Theta(U_1, U_2)\|_F = 0 &\Leftrightarrow \|U_{1\perp}^\top U_2\|_F = 0 \Leftrightarrow U_{2\perp} \perp U_1 \\
&\Leftrightarrow \exists O \in \mathbb{O}^{r \times r}, \text{ s.t. } U_1 O = U_2.
\end{aligned}$$

$\square$

**Proposition A.4** (Triangular inequality).

$$\|\sin\Theta(U_1, U_2)\|_F \le \|\sin\Theta(U_1, U_3)\|_F + \|\sin\Theta(U_2, U_3)\|_F$$

*Proof.* By the triangular inequality for Frobenius norm we have:

$$\|U_1 U_1^\top - U_2 U_2^\top\|_F \le \|U_1 U_1^\top - U_3 U_3^\top\|_F + \|U_2 U_2^\top - U_3 U_3^\top\|_F,$$

then apply Proposition A.2 to replace the Frobenius norm with $\sin\Theta$ distance we can finish the proof. $\square$

## A.2 PRINCIPAL COMPONENT ANALYSIS AND AUTOENCODERS

Autoencoders are popular unsupervised learning methods to perform dimension reduction. Its basic idea is to learn low dimensional representations for the original data while largely preserving its salient features. To achieve this goal, autoencoders learn two functions: encoder $f : \mathbb{R}^d \to \mathbb{R}^r$ and decoder $g : \mathbb{R}^r \to \mathbb{R}^d$. While the encoder $f$ compresses the high dimensional data to the low dimensional space, we try to recover the original data based on this low dimensional representation via the decoder $g$. Formally, it can be formulated to be the following optimization problem:

$$\min_{f,g} \mathbb{E}_x \mathcal{L}(x, g(f(x))). \tag{A.4}$$

Intuitively speaking, optimization problem (A.4) is trying to preserve the most essential features to recover the original data in the low dimensional representation. In practice, a commonly used form is to optimize the empirical loss function and choose the loss function to be mean squared error, that is:

$$\min_{f^{AE}, g^{DE}} \frac{1}{n} \|x_i - g^{DE}(f^{AE}(x_i))\|_2^2, \quad \forall i \in [n], \tag{A.5}$$

where $f^{AE}, g^{DE}$ are usually two neural networks. Now if we take $f^{AE}, g^{DE}$ to be both linear transformations, i.e.,

$$f^{AE}(x) = W_{AE}x + b_{AE}, g^{DE}(x) = W_{DE}x + b_{DE},$$

where $W_{AE} \in \mathbb{R}^{r \times d}, b_{AE} \in \mathbb{R}^r, W_{DE} \in \mathbb{R}^{d \times r}, , b_{AE} \in \mathbb{R}^d$, and denote $X = [x_1, \cdots, x_n] \in \mathbb{R}^{d \times n}$, the optimization problem (A.5) can be reduced to be:

$$\min_{W_{AE}, b_{AE}, W_{DE}, b_{DE}} \frac{1}{n} \|X - (W_{DE}(W_{AE}X + b_{AE}1_n^\top) + b_{DE}1_n^\top)\|_F^2.$$

It's shown in Bourlard & Kamp (2004) that the bias term can be reduced by centeralize the data matrix $X$, that is, denote $\bar{x} = \frac{1}{n}\sum_{i=1}^n x_i$ as the sample mean and $X_0 = X - \bar{x}1_n^\top$ as the centered sample matrix, then the optimization problem correspond to $W_{AE}$ and $W_{DE}$ can be transfromed to be:

$$\min_{W_{AE}, W_{DE}} \frac{1}{n} \|X_0 - W_{DE}W_{AE}X_0\|_F^2.$$

By Theorem 2.4.8 in (Golub & Loan, 1996), the optimal solution is given by the singular value decomposition of matrix $X_0$, i.e., the eigenspace of $X(I_n - \frac{1}{n}1_n1_n^\top)X^\top$, which is actually what PCA does. This fact indicates that PCA is actually a linear case of autoencoders, which is often known as *undercomplete linear autoencoders*. (Bourlard & Kamp, 1988; Plaut, 2018; Fan et al., 2019)

## A.3 HILBERT-SCHMIDT INDEPENDENT CRITERIA

In Gretton et al. (2005), the Hilbert Schmidt Independent Criteria (HSIC) is proposed to measure the dependence between two random variables by computing the Hilbert-Schmidt norm of the cross-covariance operator associated with their Reproducing Kernel Hilbert Spaces (RKHSs). Such measurement has been widely used as supervised loss function in feature selection(Song et al., 2007c), feature extraction(Song et al., 2007a), clustering(Song et al., 2007b) and supervised PCA (Barshan et al., 2011).

The basic idea behind HSIC is that two random variables, named $\mathcal{X}$ and $\mathcal{Y}$, are independent if and only if any bounded continuous function of the two random variables are uncorrelated. Let $\mathcal{F}$ be a separable RKHS containing all continuous bounded real-valued functions of $x$ from $\mathcal{X}$ to $\mathbb{R}$ and $\mathcal{G}$ be that of $\mathcal{Y}$, likewise. To each point $x \in \mathcal{X}$, there corresponds an element $\phi(x) \in \mathcal{F}$ such that $\langle \phi(x), \phi(x') \rangle_{\mathcal{F}} = k(x, x')$, where $k : \mathcal{X} \times \mathcal{X} \to \mathbb{R}$ is a unique positive definite kernel. Likewise, define the kernel $l(\cdot, \cdot)$ and feature map $\psi$ for $\mathcal{G}$. Denote the joint measure of $p_{x,y}$, then the empirical HSIC is defined to be:

**Definition A.1** (Empirical HSIC (Gretton et al., 2005))**.** Let $Z := \{(x_1, y_1), \ldots, (x_m, y_m)\} \subseteq \mathcal{X} \times \mathcal{Y}$ be a series of $m$ independent observations drawn from $p_{xy}$. An estimator of HSIC, written $\mathrm{HSIC}(Z, \mathcal{F}, \mathcal{G})$, is given by

$$\mathrm{HSIC}(Z, \mathcal{F}, \mathcal{G}) := (m-1)^{-2} \mathrm{tr}(KHLH),$$

where $H, K, L \in \mathbb{R}^{m \times m}$, $K_{ij} := k\left(x_i, x_j\right)$, $L_{ij} := l\left(y_i, y_j\right)$ and $H = I_m - \frac{1}{m} 1_m 1_m^\top$.

In our problems, we hope to maximize the dependency between learned features $WX \in \mathbb{R}^{r \times n}$ and label $y \in \mathbb{R}^n$ via HSIC, equivalently maximize $\mathrm{tr}(KHLH)$ where $K$ is a kernel of $WX$ (e.g. $X^\top W^\top WX$) and $L$ is a kernel of $y$ (e.g. $yy^\top$). Then we obtain our supervised loss corresponding to parameter $W$:

$$\mathrm{HSIC}(X, y; W) = \frac{1}{(n-1)^2} \mathrm{tr}\left(X^\top W^\top WXH yy^\top H\right). \tag{A.6}$$

### A.3.1 CONNECTION WITH MEAN SQUARED ERROR

In regression tasks, a more commonly used loss function is mean squared error:

$$\mathcal{L}_{\mathrm{MSE}}(\theta) = \frac{1}{n} \sum_{i=1}^{n} \|f(x_i, \theta) - y_i\|_F^2, \tag{A.7}$$

where $\theta$ is model parameter. In our contrastive learning framework, where we first learn the representation via a linear transformation and then perform linear regression to learn a classifier $w \in \mathbb{R}^r$ with learned representation. Assuming that both of $X$ and $y$ have been centered and we ignore the bias term, the model can be viewed as a two layer linear network:

$$f(x, \theta) = w^\top Wx.$$

Plug this formula back into mean squared error, we have:

$$\mathcal{L}_{\mathrm{MSE}}(\theta) = \frac{1}{n} \|w^\top WX - y^\top\|_F^2.$$

Since the label is a scalar, the optimal $W$ will be singular since we perform a 1-dimensional projection via $w$. That is, the feature filtering discussed in the main body and if we jointly optimize this loss with unsupervised contrastive loss, the optimal $W$ should be full rank in general.

On the other hand, the classifier $w$ only involves in the supervised loss function and does not affect the contrastive loss, thus we can find the optimal solution for $w$ and $W$ sequentially. For any fixed $W$ such that $\mathrm{rank}(WX) = d$, which can be achieved via joint optimization, the optimal solution of $w$ is:

$$w^\star = (WXX^\top W^\top)^{-1} WXy.$$

And the optimal error is:

$$\begin{aligned}\mathcal{L}(W) =& \frac{1}{n} \|X^\top W^\top (WXX^\top W^\top)^{-1} WXy - y\|_F^2 \\ =& \frac{1}{n}(y^\top y - \mathrm{tr}(y^\top X^\top W^\top (WXX^\top W^\top)^{-1} WXy)).\end{aligned}$$

Ignoring the constant term $y^\top y$ and scalar, we can find that the only difference between this loss function and HSIC (A.6) is the inverse matrix $(WXX^\top W^\top)^{-1}$ which can be viewed as normalization of $W$. In our contrastive learning framework, this normalization can be achieved by the regularization term $\|WW^\top\|_F^2$. Thus we can use the HSIC to replace the standard regression error which helps us to avoid dealing with singularity and additional parameters in optimization.

## B OMITTED PROOFS FOR SECTION 3

### B.1 PROOFS FOR SECTION 3.1

In this section, we will prove Theorem 3.1 and 3.2 in Section 3.1. The restatement and proof of them can be found in B.1 and B.3. Before starting the proof, we first provide a lemma to justify the order of $I(U^\star)$ in the Assumption 3.3.

**Lemma B.1** (Expectation of incoherent constant over a uniform distribution).

$$\mathbb{E}_{U \sim Uniform(\mathbb{O}_{d,r})} I(U^\star) = O\left(\frac{r}{d} \log d\right). \tag{B.1}$$

Before starting the proof, we give two technical lemmas to help the proof.

**Lemma B.2** (Uniform distribution on the unit shpere (Marsaglia, 1972))**.** *If $x_1, x_2, \cdots, x_n$ i.i.d.* $\sim \mathcal{N}(0, 1)$, *then* $(x_1/\sqrt{\sum_{i=1}^n x_i^2}, \cdots, x_n/\sqrt{\sum_{i=1}^n x_i^2})$ *is uniformly distributed on the unit sphere* $\mathbb{S}^d = \{(x_1, \cdots, x_n) \in \mathbb{R}^n : \sum_{i=1}^n x_i^2 = 1\}$.

**Lemma B.3.** *If $x_1, x_2, \cdots, x_n$ i.i.d. $\sim \mathcal{N}(0, 1)$, then:*

$$\mathbb{E} \max_{1 \leq i \leq n} x_i^2 \leq 2 \log(n).$$

*Proof.* Denote $Y = \max_{1 \leq i \leq n} x_i^2$, then we have:

$$\exp(t\mathbb{E}Y) \leq \mathbb{E} \exp(tY) \leq \mathbb{E} \sum_{i=1}^n \exp(tx_i^2) = n\mathbb{E} \exp(tx_i^2).$$

Note that the moment-generating function of chi square distribution with $v$ degrees of freedom is:

$$M_X(t) = (1 - 2t)^{-v/2}.$$

Then combine this fact with equation B.1 we have:

$$\exp(t\mathbb{E}Y) \leq n(1 - 2t)^{-\frac{1}{2}},$$

which implies:

$$\mathbb{E}Y \leq \frac{\log(n)}{t} - \frac{1 - 2t}{2t}, \quad \forall t < \frac{1}{2}.$$

In particular, take $t \to \frac{1}{2}$ yields:

$$\mathbb{E}Y \leq 2 \log(n)$$

as desired. $\square$

*Proof of Lemma B.1.* Denote the columns of $U$ as $U = [u_1, \cdots, u_r] \in \mathbb{O}_{d,r}$, we have:

$$\mathbb{E}_{U \sim \text{Uniform}(\mathbb{O}_{d,r})} I(U) = \mathbb{E}_{U \sim \text{Uniform}(\mathbb{O}_{d,r})} \max_{i \in [d]} \sum_{j=1}^r |e_i^\top u_j|^2$$

$$\leq \mathbb{E}_{U \sim \text{Uniform}(\mathbb{O}_{d,r})} \sum_{j=1}^r \max_{i \in [d]} |e_i^\top u_j|^2$$

$$= r \mathbb{E}_{u \sim \text{Uniform}(\mathbb{S}^d)} \max_{i \in [d]} |e_i^\top u|^2.$$

By Lemma B.2 we can transform this expectation on the uniform sphere distribution into normalized multivariate Gaussian variables:

$$\mathbb{E}_{U \sim \text{Uniform}(\mathbb{O}_{d,r})} I(U) = r \mathbb{E}_{x_1, \cdots, x_d} \frac{\max_{i \in [d]} x_i^2}{\sum_{j=1}^d x_j^2}. \tag{B.2}$$

where $x_1, x_2, \cdots, x_d$ are i.i.d. standard normal random variables. Apply Chebyshev's inequality we know that:

$$\mathbb{P}\left(|\frac{1}{d} \sum_{i=1}^d x_j^2 - 1| > \epsilon\right) \leq \frac{2}{d\epsilon^2}.$$

In particular, take $\epsilon = 1$ we have:

$$\mathbb{P}\left(\sum_{i=1}^d x_j^2 < \frac{d}{2}\right) \leq \frac{8}{d}.$$

Then take it back into equation B.2 and apply Lemma B.3 we obtain:

$$
\begin{aligned}
\mathbb{E}_{U \sim \text{Uniform}(\mathbb{O}_{d,r})} I(U) =& r\mathbb{E}_{x_1,\cdots,x_d} \frac{\max_{i\in[d]} x_i^2}{\sum_{j=1}^d x_j^2} \mathbb{I}\{\sum_{i=1}^d x_j^2 < \frac{d}{2}\} \\
&+ r\mathbb{E}_{x_1,\cdots,x_d} \frac{\max_{i\in[d]} x_i^2}{\sum_{j=1}^d x_j^2} \mathbb{I}\{\sum_{i=1}^d x_j^2 \geq \frac{d}{2}\} \\
\leq& r\mathbb{P}\left(\sum_{i=1}^d x_j^2 < \frac{d}{2}\right) + \frac{2r}{d}\mathbb{E}_{x_1,\cdots,x_d} \max_{i\in[d]} x_i^2 \\
\leq& \frac{8r}{d} + \frac{4r\log d}{d}
\end{aligned}
$$

as desired. □

Lemma B.1 demonstrates the expectation of incoherent constant over a uniform distribution on $\mathbb{O}_{d,r}$ takes the order of $O(\frac{r}{d}\log d)$, thus in the main body we take the order to be the same as it. Again we need to mention that the order can be chosen to be other functions that decrease to 0 when $d \to \infty$, and one can easily prove our results under more general assumptions.

Now, let's start proving our main results. In the mainbody and section A.2, we have shown that in the linear representation setting, the autoencoder can be deduced to PCA. Here we briefly review the results again, for the autoencoder with an encoder $f^{AE} : \mathbb{R}^d \to \mathbb{R}^r$ and a decoder $g^{DE} : \mathbb{R}^r \to \mathbb{R}^d$, it can be formalized as solving the following optimization problem for samples $\{x_i\}_{i=1}^n$

$$
\min_{f^{AE},g^{AE}} \frac{1}{n}\sum_{i=1}^n \|x_i - g^{DE}(f^{AE}(x_i))\|_2^2.
$$

In the linear representation setting, where $f^{AE}(x) = W_{AE}x + b_{AE}$ and $g^{DE}(y) = W_{DE}y + b_{DE}$, it has been shown that the optimal $W_{AE}$ is given by:

$$
W_{AE} = \left(U_{AE}\Sigma_{AE}V_{AE}^\top\right)^\top, \tag{B.3}
$$

where $U_{AE}$ is top-r eigenspace of matrix $M := X(I - 1_n 1_n^\top/n)X^\top$, $\Sigma_{AE}$ is a diagonal matrix consists of eigenvalues of $M$ and $V_{AE} = [v_1,\cdots,v_n] \in \mathbb{R}^{r\times r}$ can be any orthonormal matrix. Note that $U_{AE}$ is the top-$r$ left eigenspace of the observed covariance matrix and $U^\star$ is that of core feature covariance matrix, and by Assumption 3.2 the observed covariance matrix is dominated by the covariance of random noise. The Davis-Kahan theorem provides a technique to estimate the eigenspace distance via estimate the difference between target matrices. We will adopt this technique to prove the lower bound of feature recovery ability of autoencoder in Theorem 3.1.

**Theorem B.1** (Restatement of Theorem 3.1). *Consider the spiked covariance model Eq.(2.5), under Assumption 3.1-3.3 and $n > d \gg r$, let $W_{AE}$ be the learned representation of autoencoder with singular value decomposition $W_{AE} = (U_{AE}\Sigma_{AE}V_{AE}^\top)^\top$ (as in Eq.(3.1)). If we further assume $\{\sigma_i^2\}_{i=1}^d$ are different from each other and $\sigma_{(1)}^2/(\sigma_{(r)}^2 - \sigma_{(r+1)}^2) < C_\sigma$ for some universal constant $C_\sigma$. Then there exist two universal constants $C_\rho > 0, c \in (0,1)$, such that when $\rho < C_\rho$, we have*

$$
\mathbb{E}\|\sin\Theta(U^\star, U_{AE})\|_F \geq c\sqrt{r}. \tag{B.4}
$$

*Proof.* Denote $M = \nu^2 U^\star U^{\star\top}$ to be the target matrix, $x_i = U^\star z_i + \xi_i, \quad i = 1,2,\cdots n$ to be the samples generated from model 2.5 and let $X = [x_1,\cdots,x_n] \in \mathbb{R}^{d\times n}, Z = [z_1,\cdots,z_n] \in \mathbb{R}^{r\times n}, E = [\xi_1,\cdots,\xi_n] \in \mathbb{R}^{d\times n}$ to be the corresponding matrices. In addition, we write the column mean matrix $\bar{X} \in \mathbb{R}^{n\times d}$ of a matrix $X \in \mathbb{R}^{n\times d}$ to be $\bar{X} = \frac{1}{n}X1_n1_n^\top$, that is, each column of $\bar{X}$ is the column mean of $X$. We denote the sum of variance $\sigma_i^2$ as $\sigma_{\text{sum}}^2 = \sum_{i=1}^d \sigma_i^2$. As shown in B.3, autoencoder finds the top-r eigenspace of the following matrix:

$$
\hat{M}_1 = \frac{1}{n}X(I_n - \frac{1}{n}1_n1_n^\top)X^\top = \frac{1}{n}(U^\star Z + E)(U^\star Z + E)^\top - \frac{1}{n}(U^\star\bar{Z} + \bar{E})(U^\star\bar{Z} + \bar{E})^\top.
$$

The rest of the proof is divided into three steps for the sake of presentation.

Step 1, bound the difference between $\hat{M}_1$ and $\Sigma$. In this step, we aim to show that the data recovery of autoencoder is dominated by the random noise term. Note that $\Sigma = \mathrm{Cov}(\xi) = \mathbb{E}\xi\xi^\top$, we just need to bound the norm of the following matrix:

$$\hat{M}_1 - \Sigma = \frac{1}{n}U^\star ZZ^\top U^{\star\top} + \frac{1}{n}(U^\star ZE^\top + EZ^\top U^{\star\top}) + (\frac{1}{n}EE^\top - \Sigma) - \frac{1}{n}(U^\star\bar{Z} + \bar{E})(U^\star\bar{Z} + \bar{E})^\top,$$
(B.5)

and we will deal with these four terms separately.

1. For the first term, note that $\mathbb{E}zz^\top = \nu^2 I_r$, the first term can then be divided into two terms

$$\frac{1}{n}U^\star ZZ^\top U^{\star\top} = M + U^\star(\frac{1}{n}ZZ^\top - \mathbb{E}zz^\top)U^{\star\top}.$$
(B.6)

Then apply the concentration inequality of Wishart-type matrices (Lemma D.3) we have:

$$\mathbb{E}\|\frac{1}{n}ZZ^\top - \mathbb{E}zz^\top\|_2 \le (\sqrt{\frac{r}{n}} + \frac{r}{n})\nu^2.$$

Plug it back into (B.6) we obtain the bound for the first term:

$$\|\frac{1}{n}UZZ^\top U^\top\|_2 \le \|M\|_2 + \|U\|_2\|\frac{1}{n}ZZ^\top - \mathbb{E}zz^\top\|_2\|U\|_2 \le \left(1 + \sqrt{\frac{r}{n}} + \frac{r}{n}\right)\nu^2.$$
(B.7)

2. For the second term, since $Z$ and $E$ are independent, we must have $\mathbb{E}U^\star ZE^\top = 0$, so apply Lemma D.2 twice we have:

$$\begin{aligned}
\frac{1}{n}\mathbb{E}\|EZ^\top U^\star\|_2 &= \frac{1}{n}\mathbb{E}_Z[\mathbb{E}_E[\|EZ^\top U^\star\|_2|Z]] \\
&\lesssim \frac{1}{n}\mathbb{E}_Z[\|Z\|_2(\sigma_{\mathrm{sum}} + r^{1/4}\sqrt{\sigma_{\mathrm{sum}}\sigma_{(1)}} + \sqrt{r}\sigma_{(1)})] \\
&\lesssim \frac{1}{n}\mathbb{E}_Z[\|Z\|_2]\sqrt{d}\sigma_{(1)} \\
&\lesssim \frac{1}{n}\sqrt{d}\sigma_{(1)}(r^{1/2}\nu + (nr)^{1/4}\nu + n^{1/2}\nu) \\
&\lesssim \frac{\sqrt{d}}{\sqrt{n}}\sigma_{(1)}\nu.
\end{aligned}$$
(B.8)

3. For the third term, apply Lemma D.3 again yields:

$$\mathbb{E}\|\frac{1}{n}EE^\top - \Sigma\|_2 \le \left(\sqrt{\frac{d}{n}} + \frac{d}{n}\right)\sigma_{(1)}^2.$$
(B.9)

4. For the last term, note that each columns of $\bar{Z}$ and $\bar{E}$ are the same, so we can rewrite is as:

$$\frac{1}{n}(U^\star\bar{Z} + \bar{E})(U^\star\bar{Z} + \bar{E})^\top = (U^\star\bar{z} + \bar{\xi})(U^\star\bar{z} + \bar{\xi})^\top,$$

where $\bar{z} = \frac{1}{n}\sum_{i=1}^n z_i$ and $\bar{\xi} = \frac{1}{n}\sum_{i=1}^n \xi_i$. Since $z$ and $\xi$ are independent zero mean sub-Gaussian random variables and $\mathrm{Cov}(z) = \nu^2 I_r, \mathrm{Cov}(\xi) = \Sigma$, we can conclude that:

$$\mathbb{E}\|\frac{1}{n}(U^\star\bar{Z} + \bar{E})(U^\star\bar{Z} + \bar{E})^\top\|_2 \le \mathbb{E}\|\bar{z}\bar{z}^\top\|_2 + 2\mathbb{E}\|\bar{z}\bar{\xi}^\top\|_2 + \mathbb{E}\|\bar{\xi}\bar{\xi}^\top\|_2$$
$$\lesssim \frac{r\nu^2}{n} + \frac{\sqrt{d}}{\sqrt{n}}\sigma_{(1)}\nu + \frac{d\sigma_{(1)}^2}{n}.$$

To sum up, combine equations (B.7)(B.8)(B.9)(4) together we obtain the upper bound for the 2 norm expectation of matrix $\hat{M} - \Sigma$:

$$\mathbb{E}\|\hat{M}_1 - \Sigma\|_2 \lesssim \nu^2\left(1 + \sqrt{\frac{r}{n}} + \frac{r}{n}\right) + \sigma_{(1)}^2\left(\sqrt{\frac{d}{n}} + \frac{d}{n}\right) + \sqrt{\frac{d}{n}}\sigma_{(1)}\nu. \qquad \text{(B.10)}$$

Step 2, bound the $\sin\Theta$ distance between eigenspaces. As we have shown in step 1, the target matrix of autoencoder is close to the covariance matrix of random noise, i.e., $\Sigma$. Note that $\Sigma$ is assumed to be diagonal matrix with different elements, hence its eigenspace only consists of canonical basis $e_i$. Denote $U_\Sigma$ to be the top-r eigenspace of $\Sigma$ and $\{e_i\}_{i\in C}$ to be its corresponding basis vectors, apply the Davis-Kahan Theorem D.1 we can conclude that:

$$\mathbb{E}\|\sin\Theta(U_{AE}, U_\Sigma)\|_F \le \frac{2\sqrt{r}\mathbb{E}\|\hat{M}_1 - \Sigma\|_2}{\sigma_{(r)}^2 - \sigma_{(r+1)}^2}$$

$$\lesssim \sqrt{r}\frac{1}{\sigma_{(1)}^2}\left(\nu^2\left(1 + \sqrt{\frac{r}{n}} + \frac{r}{n}\right) + \sigma_{(1)}^2\left(\sqrt{\frac{d}{n}} + \frac{d}{n}\right) + \sqrt{\frac{d}{n}}\sigma_{(1)}\nu\right)$$

$$\lesssim \sqrt{r}\left(\rho^2 + \sqrt{\frac{d}{n}} + \rho\sqrt{\frac{d}{n}}\right).$$

Step 3, obtain the final result by triangular inequality. By Assumption 3.3 we know that the distance between canonical basis and the eigenspace of core features can be large:

$$\|\sin\Theta(U^\star, U_\Sigma)\|_F^2 = \|U_{\Sigma\perp}^\top U^\star\|_F^2 = \sum_{i\in[d]/C}\|e_i^\top U^\star\|_F^2 = \|U^\star\|_F^2 - \sum_{i\in C}\|e_i^\top U^\star\|_F^2$$

$$\ge r - rI(U^\star) = r - O\left(\frac{r^2}{d}\log d\right).$$

Then apply the triangular inequality of $\sin\Theta$ distance (Proposition A.4) we can obtain the lower bound of autoencoder.

$$\mathbb{E}\|\sin\Theta(U_{AE}, U^\star)\|_F \ge \mathbb{E}\|\sin\Theta(U^\star, U_\Sigma)\|_F - \mathbb{E}\|\sin\Theta(U_{AE}, U_\Sigma)\|_F$$

$$\ge \sqrt{r} - O\left(\frac{r}{\sqrt{d}}\sqrt{\log d}\right) - O\left(\sqrt{r}\left(\rho^2 + \sqrt{\frac{d}{n}} + \rho\sqrt{\frac{d}{n}}\right)\right).$$

By Assumption 3.2, it implies that when n and d is sufficient large and $\rho$ is sufficient small (smaller than a given constant $C_\rho > 0$), there exist a universal constant $c \in (0, 1)$ such that:

$$\mathbb{E}\|\sin\Theta(U_{AE}, U^\star)\|_F \ge c\sqrt{r}.$$

$\square$

On the other hand, recall that the optimization problem for self-supervised contrastive learning has been formulated to be:

$$\min_{W\in\mathbb{R}^{d\times r}} -\frac{1}{n}\sum_{i=1}^n\left[\sum_{x_i^{Pos}\in\mathcal{B}_i^{Pos}}\frac{\langle f(x_i, W), f(x_i^{Pos}, W)\rangle}{|\mathcal{B}_i^{Pos}|} - \sum_{x_i^{Neg}\in\mathcal{B}_i^{Neg}}\frac{\langle f(x_i, \theta), f(x_i^{Neg}, W)\rangle}{|\mathcal{B}_i^{Neg}|}\right] + \lambda R(W),$$

(B.11)

where $f(x, W) = Wx$, $R(W) = \|WW^\top\|_F^2/2$. To compare contrastive learning with autoencoder, we now derive the optimal solution of the optimization problem B.11. Let's start with the general result for self-supervised contrastive learning with augmented pairs generation (Definition 2.1), and then turn to the special case for random masking augmentation (Definition 2.2).

**Theorem B.2.** *For two fixed augmentation function $g_1, g_2 : \mathbb{R}^d \to \mathbb{R}^d$, denote the augmented data matrices as $X_1 = [g_1(x_1), \cdots, g_1(x_n)] \in \mathbb{R}^{d\times n}$ and $X_2 = [g_2(x_1), \cdots, g_2(x_n)] \in \mathbb{R}^{d\times n}$, when the augmented pairs are generated as in Definition 2.1, the optimal solution of contrastive learning problem (B.11) is given by:*

$$W_{CL} = C\left(\sum_{i=1}^r u_i\sigma_iv_i^\top\right)^\top,$$

*where $C > 0$ is a positive constant, $\sigma_i$ is the $i$-th largest eigenvalue of the following matrix:*

$$X_1 X_2^\top + X_2 X_1^\top - \frac{1}{2(n-1)}(X_1 + X_2)(1_r 1_r^\top - I_r)(X_1 + X_2)^\top, \tag{B.12}$$

*$u_i$ is the corresponding eigenvector and $V = [v_1, \cdots, v_n] \in \mathbb{R}^{r \times r}$ can be any orthonormal matrix.*

*Proof of Theorem B.2.* When augmented pairs generation 2.1 is applied, the contrastive loss can be written as:

$$
\begin{aligned}
\mathcal{L}(W) =& \frac{\lambda}{2}\|WW^\top\|_F^2 - \frac{1}{n}\sum_{i=1}^n [\langle W t_1(x_i), W t_2(x_i)\rangle \\
& - \frac{1}{4(n-1)}\sum_{j \neq i}\langle W t_1(x_i) + W t_2(x_i), W t_1(x_j) + W t_2(x_i)\rangle] \\
=& \frac{\lambda}{2}\|WW^\top\|_F^2 - \frac{1}{n}\sum_{i=1}^n \langle W t_1(x_i), W t_2(x_i)\rangle \\
& + \frac{1}{4n(n-1)}\sum_{i=1}^n\sum_{j \neq i}\langle W t_1(x_i) + W t_2(x_i), W t_1(x_j) + W t_2(x_i)\rangle \\
=& \frac{\lambda}{2}\|WW^\top\|_F^2 - \frac{1}{2n}\operatorname{tr}\big(X_1^\top W^\top W X_2 + X_2^\top W^\top W X_1\big) \\
& + \frac{1}{4n(n-1)}\operatorname{tr}\big((1_n 1_n^\top - I_n)(X_1 + X_2)^\top W^\top W (X_1 + X_2)\big) \\
=& \frac{\lambda}{2}\|WW^\top\|_F^2 \\
& - \frac{1}{2n}\operatorname{tr}\left((X_2 X_1^\top + X_1 X_2^\top - \frac{1}{2(n-1)}(X_1 + X_2)(1_n 1_n^\top - I_n)(X_1 + X_2)^\top)W^\top W\right) \\
=& \frac{1}{2}\left\|\lambda W^\top W - \frac{1}{2n\lambda}\left(X_2 X_1^\top + X_1 X_2^\top - \frac{1}{2(n-1)}(X_1 + X_2)(1_n 1_n^\top - I_n)(X_1 + X_2)^\top\right)\right\|_F^2 \\
& - \left\|\frac{1}{2n\lambda}\left(X_2 X_1^\top + X_1 X_2^\top - \frac{1}{2(n-1)}(X_1 + X_2)(1_n 1_n^\top - I_n)(X_1 + X_2)^\top\right)\right\|_F^2.
\end{aligned}
$$

Note that the last term only depends on $X$, and the first term implies that when $W_{CL}$ is the optimal solution, $\lambda W_{CL} W_{CL}^\top$ is the best rank-$r$ approximation of $\frac{1}{(n-1)\lambda} X H X^\top$. Then apply Lemma D.4 we can conclude that $W_{CL}$ satisfy the desired conditions. $\qquad\square$

Theorem B.2 shows a general result for augmented pairs generation with any augmentation. Specifically, if we apply the random masking augmentation 2.2, we can obtain a more precise result to characterize the optimal solution. To formally state the result, we need additional notations: for any square matrix $A \in \mathbb{R}^{d \times d}$, we denote $D(A)$ to be $A$ with all off-diagonal entries set to be zero and $\Delta(A) = A - D(A)$ to be $A$ with all diagonal entries set to be zero. Then we have the following corollary for random masking augmentation.

**Corollary B.1.** *Under the same conditions as in Theorem B.2, if we use random masking (Definition 2.2) as our augmentation function, then in expectation, the optimal solution of contrastive learning problem (B.11) is given by:*

$$W_{CL} = C\Big(\sum_{i=1}^r u_i \sigma_i v_i^\top\Big)^\top,$$

*where $C > 0$ is a positive constant, $\sigma_i$ is the $i$-th largest eigenvalue of the following matrix:*

$$\Delta(XX^\top) - \frac{1}{n-1}X(1_n 1_n^\top - I_n)X^\top, \tag{B.13}$$

*$u_i$ is the corresponding eigenvector and $V = [v_1, \cdots, v_n] \in \mathbb{R}^{r \times r}$ can be any orthonormal matrix.*

*Proof of Corollary B.1.* Following the proof of Theorem B.2, now we only need to compute the expectation over the augmentation distribution defined in 2.2:

$$
\begin{aligned}
\mathcal{L}(W) = &\frac{\lambda}{2}\|WW^\top\|_F^2 - \mathbb{E}_{(t_1,t_2)\sim\mathcal{T}}[\frac{1}{n}\sum_{i=1}^n[\langle Wt_1(x_i), Wt_2(x_i)\rangle \\
&- \frac{1}{4(n-1)}\sum_{j\neq i}\langle Wt_1(x_i) + Wt_2(x_i), Wt_1(x_j) + Wt_2(x_i)\rangle]] \\
= &\frac{\lambda}{2}\|WW^\top\|_F^2 - \mathbb{E}_{(t_1,t_2)\sim\mathcal{T}}[\frac{1}{2n}\mathrm{tr}((X_2 X_1^\top + X_1 X_2^\top \\
&- \frac{1}{2(n-1)}(X_1 + X_2)(1_n 1_n^\top - I_n)(X_1 + X_2)^\top)W^\top W)].
\end{aligned}
\tag{B.14}
$$

Note that by the definition of random masking augmentation, we have $X_1 = AX, X_2 = (I-A)X$, which implies $X_1 + X_2 = X$. On the other hand, $X_1$ and $X_2$ has disjoint nonzero dimensions, hence the matrix $X_1 X_2^\top + X_2 X_1^\top$ only consists of off-diagonal entries and each of the off-diagonal entry, let's say $x_{ij}$, appears if and only if $a_i + a_j = 1$. Moreover, once it appears, we must have $x_{ij}$ equals to the $(i,j)$ element of $XX^\top$. With this result, we can then compute the expectation in equation (B.14):

$$
\begin{aligned}
\mathcal{L}(W) = &\frac{\lambda}{2}\|WW^\top\|_F^2 - \mathbb{E}_{(t_1,t_2)\sim\mathcal{T}}[\frac{1}{2n}\mathrm{tr}((X_2 X_1^\top + X_1 X_2^\top \\
&- \frac{1}{2(n-1)}(X_1 + X_2)(1_n 1_n^\top - I_n)(X_1 + X_2)^\top)W^\top W)] \\
= &\frac{\lambda}{2}\|WW^\top\|_F^2 - \frac{1}{2n}\mathrm{tr}\left((\frac{1}{2}\Delta(XX^\top) - \frac{1}{2(n-1)}X(1_n 1_n^\top - I_n)X^\top)W^\top W\right) \\
= &\frac{1}{2}\|\lambda W^\top W - \frac{1}{4n\lambda}(\Delta(XX^\top) - \frac{1}{n-1}X(1_n 1_n^\top - I_n)X^\top\|_F^2 \\
&- \|\frac{1}{4n\lambda}(\Delta(XX^\top) - \frac{1}{n-1}X(1_n 1_n^\top - I_n)X^\top\|_F^2.
\end{aligned}
$$

By similar argument as in the proof of Theorem B.2, we can conclude that $W_{CL}$ satisfy the desired conditions. $\square$

*Remark* B.1. Note that the two views generated by random masking augmentation have disjoint non-zero dimensions, hence contrasting such positive pairs yields correlation between different dimensions only. That's why the first term in equation (B.13) appears to be $\Delta(XX^\top)$ where the diagonal entries are eliminated.

With Theorem B.2 and Corollary B.1 established, we can find that the self-supervised contrastive learning equipped with augmented pairs generation and random masking augmentation can eliminate the effect of random noise on the diagonal entries of the observed covariance matrix. Since $\mathrm{Cov}(\xi) = \Sigma$ is a diagonal matrix, and by Assumption 3.3 we know that the diagonal entries $\mathrm{Cov}(U^\star z) = \nu^2 U^\star U^{\star\top}$ only take a small proportion of the total Frobenius norm. Thus contrasting augmented pairs will preserve the core features while eliminating most of the random noise, and give a more accurate estimation of core features. To start the proof, we introduce a technical lemma first.

**Lemma B.4** (Lemma 4 in Zhang et al. (2018)). *If $M \in \mathbb{R}^{p\times p}$ is any square matrix and $\Delta(M)$ is the matrix $M$ with diagonal entries set to 0 , then*

$$
\|\Delta(M)\|_2 \leq 2\|M\|_2.
$$

*Here, the factor " 2 " in the statement above cannot be improved.*

Then we turn to prove Theorem 3.2.

**Theorem B.3** (Restatement of Theorem 3.2). *Under the spiked covariance model Eq.(2.5), random masking augmentation in Definition 2.2, Assumption 3.1-3.3 and $n > d \gg r$, let $W_{CL}$ be any solution that minimizes Eq.(2.3), and denote its singular value decomposition as $W_{CL} = (U_{CL}\Sigma_{CL}V_{CL}^\top)^\top$, then we have*

$$
\mathbb{E}\|\sin\Theta(U^\star, U_{CL})\|_F \lesssim \frac{r^{3/2}}{d}\log d + \sqrt{\frac{dr}{n}}.
\tag{B.15}
$$

*Proof.* The proof strategy is quite similar to that of Theorem 3.2 and we follow the notation defined in the first paragraph of that proof. As we have shown in Corollary B.1, under our linear representation setting, the contrastive learning algorithm finds the top-r eigenspace of the following matrix:

$$
\begin{aligned}
\hat{M}_2 =& \frac{1}{n}\left(\Delta(XX^\top) - \frac{1}{n-1}X(1_n 1_n^\top - I_n)X^\top\right) \\
=& \frac{1}{n}\Delta((U^\star Z + E)(U^\star Z + E)^\top) - \frac{1}{n-1}(U^\star \bar{Z} + \bar{E})(U^\star \bar{Z} + \bar{E})^\top \\
&+ \frac{1}{n(n-1)}(U^\star Z + E)(U^\star Z + E)^\top.
\end{aligned}
$$

To prove the theorem, first we need to bound the difference between $\hat{M}_2$ and $M$. We aim to show that the contrastive learning algorithm is dominated by the core feature term. Note that $\Sigma = \mathbb{E}Uzz^\top U^\top$, we just need to bound the norm of the following matrix:

$$
\begin{aligned}
\hat{M}_2 - M =& (\frac{1}{n}\Delta(U^\star ZZ^\top U^{\star\top}) - M) + \frac{1}{n}\Delta(U^\star ZE^\top + EZ^\top U^{\star\top}) + \frac{1}{n}\Delta(EE^\top) \\
&- \frac{1}{n-1}(U^\star \bar{Z} + \bar{E})(U^\star \bar{Z} + \bar{E})^\top + \frac{1}{n(n-1)}(U^\star Z + E)(U^\star Z + E)^\top.
\end{aligned} \tag{B.16}
$$

and we will also deal with these five terms separately.

1. For the first term, we can divide it into two parts:

$$
\frac{1}{n}\Delta(U^\star ZZ^\top U^{\star\top}) - M = \Delta(\frac{1}{n}U^\star ZZ^\top U^{\star T} - M) + \Delta(M) - M. \tag{B.17}
$$

Then apply Lemma B.4 and Lemma D.3 we have:

$$
\mathbb{E}\|\Delta(\frac{1}{n}U^\star ZZ^\top U^{\star\top} - M)\|_2 \le 2\mathbb{E}\|\frac{1}{n}U^\star ZZ^\top U^{\star\top} - M\|_2 \le 2(\sqrt{\frac{r}{n}} + \frac{r}{n})\nu^2.
$$

Using the incoherent condition $I(U) = O(\frac{r}{d}\log d)$, we know that:

$$
\|M - \Delta(M)\|_2 \le \nu^2 \max_{i\in[d]}\|e_i^\top U^\star\|_2^2 = \nu^2 I(U^\star) \lesssim \frac{r}{d}\log d\nu^2.
$$

Combine the two equations above together we obtain the bound for the first term:

$$
\mathbb{E}\|\frac{1}{n}\Delta(U^\star ZZ^\top U^{\star\top}) - M\|_2 \le \mathbb{E}\|\Delta(\frac{1}{n}U^\star ZZ^\top U^{\star\top} - M)\|_2 + \|M - \Delta(M)\|_2
$$
$$\tag{B.18}$$

$$
\lesssim \nu^2(\frac{r}{d}\log d + \frac{r}{n} + \sqrt{\frac{r}{n}}). \tag{B.19}
$$

2. For the second term, apply equation (B.8) yields:

$$
\frac{1}{n}\mathbb{E}\|\Delta(U^\star ZE^\top + EZ^\top U^{\star\top})\|_2 \le \frac{4}{n}\mathbb{E}\|EZ^\top U^{\star\top}\|_2 \lesssim \frac{\sqrt{d}}{\sqrt{n}}\sigma_{(1)}\nu. \tag{B.20}
$$

3. For the third term, apply equation (B.9) yields:

$$
\mathbb{E}\|\frac{1}{n}\Delta(EE^\top)\|_2 = \mathbb{E}\|\Delta(\frac{1}{n}EE^\top - \Sigma)\|_2 \le 2\|\frac{1}{n}EE^\top - \Sigma\|_2 \lesssim (\sqrt{\frac{d}{n}} + \frac{d}{n})\sigma_{(1)}^2. \tag{B.21}
$$

4. For the fourth term, apply equation (4) yields:

$$
\mathbb{E}\|\frac{1}{n-1}(U^\star \bar{Z} + \bar{E})(U\bar{Z} + \bar{E})^\top\|_2 \lesssim \mathbb{E}\|\frac{1}{n}(U\bar{Z} + \bar{E})(U\bar{Z} + \bar{E})^\top\|_2
$$
$$\tag{B.22}$$

$$
\lesssim \frac{r\nu^2}{n} + \frac{\sqrt{d}}{\sqrt{n}}\sigma_{(1)}\nu + \frac{d\sigma_{(1)}^2}{n}.
$$

5. For the last term, by equations (B.7)(B.8)(B.9) we know:

$$\mathbb{E}\|\frac{1}{n}(U^\star Z + E)(U^\star Z + E)^\top\|_2$$

$$\lesssim \|\Sigma\|_2 + \left(1 + \sqrt{\frac{r}{n}} + \frac{r}{n}\right)\nu^2 + \sqrt{\frac{d}{n}}\sigma_{(1)}\nu + \left(\sqrt{\frac{d}{n}} + \frac{d}{n}\right)\sigma_{(1)}^2.$$

Thus we can conclude that:

$$\mathbb{E}\|\frac{1}{n(n-1)}(U^\star Z + E)(U^\star Z + E)^\top\|_2 \lesssim \frac{d}{n}\sigma_{(1)}^2 + \frac{r}{n}\nu^2. \tag{B.23}$$

To sum up, combine equations (B.18)(B.20)(B.21)(B.22)(B.23) together we obtain the upper bound for the 2 norm expectation of matrix $\hat{M}_2 - M$:

$$\mathbb{E}\|\hat{M}_2 - M\|_2 \lesssim \nu^2\left(\frac{r}{d}\log d + \sqrt{\frac{r}{n}} + \frac{r}{n}\right) + \sigma_{(1)}^2\left(\sqrt{\frac{d}{n}} + \frac{d}{n}\right) + \sigma_{(1)}\nu\sqrt{\frac{d}{n}}. \tag{B.24}$$

With the upper bound for $\|\hat{M}_2 - M\|_2$, simply apply Lemma D.1 we can obtain the desired bound for $\sin\Theta$ distance:

$$\mathbb{E}\|\sin\Theta(U_{CL}, U^\star)\|_F \leq \frac{2\sqrt{r}\mathbb{E}\|\hat{M}_2 - M\|_2}{\nu^2}$$

$$\lesssim \sqrt{r}\frac{1}{\nu^2}\left(\nu^2\left(\frac{r}{d}\log d + \sqrt{\frac{r}{n}} + \frac{r}{n}\right) + \sigma_{(1)}^2\left(\sqrt{\frac{d}{n}} + \frac{d}{n}\right) + \sigma_{(1)}\nu\sqrt{\frac{d}{n}}\right)$$

$$= \sqrt{r}\left(\left(\frac{r}{d}\log d + \sqrt{\frac{r}{n}} + \frac{r}{n}\right) + \rho^{-2}\left(\sqrt{\frac{d}{n}} + \frac{d}{n}\right) + \rho^{-1}\sqrt{\frac{d}{n}}\right)$$

$$\lesssim \frac{r^{3/2}}{d}\log d + \sqrt{\frac{dr}{n}}.$$

Moreover, there exists an orthogonal matrix $\hat{O} \in \mathbb{O}^{r\times r}$ depending on $U_{CL}$ such that:

$$\mathbb{E}\|U^\top U_{CL}\hat{O} - I_r\|_F = \mathbb{E}\|U_{CL}\hat{O} - U\|_F \leq \frac{2\sqrt{r}\mathbb{E}\|\hat{M}_2 - M\|_2}{\nu^2} \lesssim \frac{r^{3/2}}{d}\log d + \sqrt{\frac{dr}{n}}.$$

which finishes the proof. $\qquad\square$

In the following, we will show that our results do not change if we applied the same augmentation (2.2) for autoencoders, which indicates that our comparison is fair. As discussed in Section A.2, we can ignore the bias term in autoencoders for simplicity, which only serves as centralization of the data matrix. In that case, we applied random augmentation $t_1(x) = Ax$ and $t_2(x) = (I - A)x$ to the original data $\{x_i\}_{i=1}^n$, and the optimization problem can be formulated as follows:

$$\min_{W_{AE}, W_{DE}} \frac{1}{2n}\mathbb{E}_A[\|AX - W_{DE}W_{AE}AX\|_F^2 + \|(I - A)X - W_{DE}W_{AE}(I - A)X\|_F^2]. \tag{B.25}$$

Then, similar to Theorem B.2 for contrastive learning, we can also obtain an explicit solution for this optimization problem.

**Theorem B.4.** *The optimal solution of autoencoders with random masking augmentation (B.25) is given by:*

$$W_{AE} = W_{DE}^\top = C\left(\sum_{i=1}^r u_i\sigma_i v_i^\top\right)^\top,$$

*where $C > 0$ is a positive constant, $\sigma_i$ is the $i$-th largest eigenvalue of the following matrix:*

$$\frac{1}{2}\Delta(XX^\top) + D(XX^\top), \tag{B.26}$$

*$u_i$ is the corresponding eigenvector and $V = [v_1, \cdots, v_n] \in \mathbb{R}^{r\times r}$ can be any orthonormal matrix.*

*Proof.* We first derive the equivalent form for this objective function:

$$
\begin{aligned}
&\frac{1}{2n}\mathbb{E}_A[\|AX - W_{DE}W_{AE}AX\|_F^2 + \|(I-A)X - W_{DE}W_{AE}(I-A)X\|_F^2] \\
=&\frac{1}{2n}\mathbb{E}_A[\operatorname{tr}\big(X^\top A^\top AX\big) + \operatorname{tr}\big(X^\top A^\top W_{DE}W_{AE}AX\big) + \operatorname{tr}\big(X^\top A^\top W_{AE}^\top W_{DE}^\top W_{DE}W_{AE}AX\big) \\
&+ \operatorname{tr}\big(X^\top(I-A)^\top(I-A)X\big) + \operatorname{tr}\big(X^\top(I-A)^\top W_{DE}W_{AE}(I-A)X\big) \\
&+ \operatorname{tr}\big(X^\top(I-A)^\top W_{AE}^\top W_{DE}^\top W_{DE}W_{AE}(I-A)X\big)] \\
=&\frac{1}{2n}\mathbb{E}_A[\operatorname{tr}\big(X^\top AX\big) + \operatorname{tr}\big(AXX^\top A^\top W_{DE}W_{AE}\big) + \operatorname{tr}\big(AXX^\top A^\top W_{AE}^\top W_{DE}^\top W_{DE}W_{AE}\big) \\
&+ \operatorname{tr}\big(X^\top(I-A)X\big) + \operatorname{tr}\big((I-A)XX^\top(I-A)^\top W_{DE}W_{AE}\big) \\
&+ \operatorname{tr}\big((I-A)XX^\top(I-A)^\top W_{AE}^\top W_{DE}^\top W_{DE}W_{AE}\big)] \\
=&\frac{1}{2n}\mathbb{E}_A[\operatorname{tr}\big(X^\top X\big) + \operatorname{tr}(MW_{DE}W_{AE}) + \operatorname{tr}\big(MW_{AE}^\top W_{DE}^\top W_{DE}W_{AE}\big)],
\end{aligned}
\tag{B.27}
$$

where $M := AXX^\top A^\top + (I-A)XX^\top(I-A)^\top$. Note that by Definition 2.2 we have $A = \operatorname{diag}(a_1, \cdots, a_d)$ and $a_i$ follows the Bernoulli distribution, so we have:

$$
\mathbb{E}_A M = \frac{1}{2}\Delta(XX^\top) + D(XX^\top)
\tag{B.28}
$$

Again, by Theorem 2.4.8 in (Golub & Loan, 1996), the optimal solution of B.25 is given by the eigenvalue decomposition of $\mathbb{E}_A M = \frac{1}{2}\Delta(XX^\top) + D(XX^T)$, up to an orthogonal transformation, which finishes the proof. $\qquad\square$

With Theorem B.4 established, we can now derive the space distance for autoencoders with random masking augmentation.

**Theorem B.5.** *Consider the spiked covariance model Eq.(2.5), under Assumption 3.1-3.3 and $n > d \gg r$, let $W_{AE}$ be the learned representation of augmented autoencoder with singular value decomposition $W_{AE} = (U_{AE}\Sigma_{AE}V_{AE}^\top)^\top$ (i.e., the optimal solution of optimization problem B.25). If we further assume $\{\sigma_i^2\}_{i=1}^d$ are different from each other and $\sigma_{(1)}^2/(\sigma_{(r)}^2 - \sigma_{(r+1)}^2) < C_\sigma$ for some universal constant $C_\sigma$. Then there exist two universal constants $C_\rho > 0, c \in (0,1)$, such that when $\rho < C_\rho$, we have*

$$
\mathbb{E}\left\|\sin\Theta\left(U^\star, U_{AE}\right)\right\|_F \geq c\sqrt{r}.
\tag{B.29}
$$

*Proof.* Step1, similar to the proof of Theorem B.1, we first bound the difference between $\hat{M} := \Delta(XX^\top) + 2D(XX^\top)$ and $\Sigma := \operatorname{Cov}(\xi\xi^\top)$. Note that:

$$
\|\hat{M} - \Sigma\|_2 = \|XX^\top - \Sigma - \frac{1}{2}\Delta(XX^\top)\|_2 \leq \|XX^\top - \Sigma\|_2 + \frac{1}{2}\|\Delta(XX^\top - \Sigma)\|_2 + \frac{1}{2}\|\Delta(\Sigma)\|_2
\tag{B.30}
$$

Since $\Sigma$ is a diagonal matrix, then by Lemma B.4 we have:

$$
\|\hat{M} - \Sigma\|_2 \leq 2\|XX^\top - \Sigma\|_2
\tag{B.31}
$$

Now, directly apply equation (B.7)(B.8)(B.9) we can obtain that:

$$
\mathbb{E}\|\hat{M} - \Sigma\|_2 \lesssim \nu^2\left(1 + \sqrt{\frac{r}{n}} + \frac{r}{n}\right) + \sigma_{(1)}^2\left(\sqrt{\frac{d}{n}} + \frac{d}{n}\right) + \sqrt{\frac{d}{n}}\sigma_{(1)}\nu.
\tag{B.32}
$$

Step 2, bound the $\sin\Theta$ distance between eigenspaces. As we have shown in step 1, the target matrix of autoencoder is close to the covariance matrix of random noise, i.e., $\Sigma$. Note that $\Sigma$ is assumed to be diagonal matrix with different elements, hence its eigenspace only consists of canonical basis $e_i$. Denote $U_\Sigma$ to be the top-r eigenspace of $\Sigma$ and $\{e_i\}_{i\in C}$ to be its corresponding basis vectors, apply

the Davis-Kahan Theorem D.1 we can conclude that:

$$\mathbb{E}\| \sin\Theta(U_{AE}, U_\Sigma)\|_F \leq \frac{2\sqrt{r}\mathbb{E}\|\hat{M} - \Sigma\|_2}{\sigma_{(r)}^2 - \sigma_{(r+1)}^2}$$

$$\lesssim \sqrt{r}\frac{1}{\sigma_{(1)}^2}\left(\nu^2\left(1 + \sqrt{\frac{r}{n}} + \frac{r}{n}\right) + \sigma_{(1)}^2\left(\sqrt{\frac{d}{n}} + \frac{d}{n}\right) + \sqrt{\frac{d}{n}}\sigma_{(1)}\nu\right)$$

$$\lesssim \sqrt{r}\left(\rho^2 + \sqrt{\frac{d}{n}} + \rho\sqrt{\frac{d}{n}}\right).$$

Step 3, obtain the final result by triangular inequality. By Assumption 3.3 we know that the distance between canonical basis and the eigenspace of core features can be large:

$$\|\sin\Theta(U^\star, U_\Sigma)\|_F^2 = \|U_{\Sigma\perp}^\top U^\star\|_F^2 = \sum_{i\in[d]/C}\|e_i^\top U^\star\|_F^2 = \|U^\star\|_F^2 - \sum_{i\in C}\|e_i^\top U^\star\|_F^2$$

$$\geq r - rI(U^\star) = r - O\left(\frac{r^2}{d}\log d\right).$$

Then apply the triangular inequality of $\sin\Theta$ distance (Proposition A.4) we can obtain the lower bound of autoencoder.

$$\mathbb{E}\|\sin\Theta(U_{AE}, U^\star)\|_F \geq \mathbb{E}\|\sin\Theta(U^\star, U_\Sigma)\|_F - \mathbb{E}\|\sin\Theta(U_{AE}, U_\Sigma)\|_F$$

$$\geq \sqrt{r} - O\left(\frac{r}{\sqrt{d}}\sqrt{\log d}\right) - O\left(\sqrt{r}\left(\rho^2 + \sqrt{\frac{d}{n}} + \rho\sqrt{\frac{d}{n}}\right)\right).$$

By Assumption 3.2, it implies that when n and d is sufficient large and $\rho$ is sufficient small (smaller than a given constant $C_\rho > 0$), there exist a universal constant $c \in (0, 1)$ such that:

$$\mathbb{E}\|\sin\Theta(U_{AE}, U^\star)\|_F \geq c\sqrt{r}.$$

$\square$

Compared with Theorem 3.1, we can find that random masking augmentation makes no difference to autoencoders, which justifies the fairness of our comparison between contrastive learning and autoencoders.

## B.2 PROOFS FOR SECTION 3.2

In this section, we will provide the proof of Theorem 3.3 and 3.4 with both regression and classification settings. The statement and detailed proof can be found in Theorem B.6 and B.7.

Before going onto the proof, we clarify our models and assumptions. Let $W_{CL}$ and $W_{AE}$ be the learned representations based on train data $X \in \mathbb{R}^{n\times d}$. We observe a new signal $\check{x} = U^\star\check{z} + \check{\xi}$ independent of $X$ following the spiked covariance model 2.5. For simplicity, assume $\check{z} \sim N(0, \nu^2 I_r)$ and $\check{\xi} \perp \check{z}$. We consider the two major types of downstream tasks: classification and regression. For binary classification task, we observe a new supervised sample $\check{y}$ following the binary response model:

$$\check{y}|\check{z} = \text{Ber}(F(\langle\check{z}, w^\star\rangle/\nu)), \tag{B.33}$$

where $F : \mathbb{R} \to [0, 1]$ is a known monotone increasing function satisfying $1 - F(u) = F(-u)$ for any $u \in \mathbb{R}$. Notice that our model B.33 includes logistic models (when $F(u) = 1/(1 + e^{-u})$) and probit models (when $F(u) = \Phi(u)$, where $\Phi$ is the cumulative distribution function of standard normal distribution.) We can also interpret model B.33 as a shallow neural network model with width $r$ for binary classification. For regression task, we observe a new supervised sample $\check{y}$ following the linear regression model:

$$\check{y} = \langle\check{z}, w^\star\rangle/\nu + \check{\epsilon}, \tag{B.34}$$

where $\check{\epsilon} \sim (0, \sigma_\epsilon^2)$ is independent of $\check{z}$.

In classification setting, we specify 0-1 loss, i.e., $\ell_c(\delta) \triangleq \mathbb{I}\{\check{y} \neq \delta(\check{x})\}$ for some predictor $\delta$ taking values in $\{0, 1\}$. For regression task, we employ squared error loss $\ell_r(\delta) \triangleq (\check{y} - \delta(\check{x}))^2$. Based on some learned representation $W$, we consider a class of linear predictors, i.e., $\delta_{W,w}(\check{x}) \triangleq \mathbb{I}\{F(w^\top W \check{x}) \geq 1/2\}$ for classification task and $\delta_{W,w}(\check{x}) \triangleq w^\top W \check{x}$ for regression task, where $w \in \mathbb{R}^r$ is a weight vector $w \in \mathbb{R}^r$. Note that the learned representation depends only on unsupervised samples $X$. Let $\mathbb{E}_\mathcal{D}[\cdot]$ and $\mathbb{E}_\mathcal{E}[\cdot]$ the expectations with respect to $(X, Z)$ and $(\check{y}, \check{x}, \check{z})$, respectively.

For notational simplicity, define the prediction risk of predictor $\delta$ for classification and regression tasks as $\mathcal{R}_c(\delta) := \mathbb{E}_\mathcal{D}[\ell_c(\delta)]$ and $\mathcal{R}_r(\delta) := \mathbb{E}_\mathcal{D}[\ell_r(\delta)]$, respectively. Define $\Sigma_x := \nu^2 U^\star U^{\star\top} + \Sigma$. Since any representation $W$ can be decomposed into $W = V\Sigma_W U^\top$ by singular value decomposition, where $U \in \mathbb{O}_{d,r}$, $\{\delta_{W,w} : w \in \mathbb{R}^r\} = \{\delta_{U^\top,w} : w \in \mathbb{R}^r\}$. Thus we write $\delta_{U,w}$ for $\delta_{W,w}$ with a slight abuse of notation. Our goal as stated above is to bound the prediction risk of predictors $\{\delta_{W,w} : w \in \mathbb{R}^r\}$ constructed upon the learned representations $W_{CL}$ and $W_{AE}$, i.e., the quantity $\inf_{w \in \mathbb{R}^r} \mathbb{E}_\mathcal{E}[\ell(\delta_{W_{CL},w})]$ and $\inf_{w \in \mathbb{R}^r} \mathbb{E}_\mathcal{E}[\ell(\delta_{W_{AE},w})]$.

Now we state our result on the downstream task.

**Theorem B.6** (Excess Risk for Downstream Task: Upper Bound). *Suppose the conditions in Theorem 3.2 hold. Then, for classification task, we have*

$$\mathbb{E}_\mathcal{D}\big[\inf_{w \in \mathbb{R}^r} \mathbb{E}_\mathcal{E}[\ell_c(\delta_{W_{CL},w})] - \inf_{w \in \mathbb{R}^r} \mathbb{E}_\mathcal{E}[\ell_c(\delta_{U^\star,w})]\big] = O\left(\frac{r^{3/2}}{d}\log d + \sqrt{\frac{dr}{n}}\right) \wedge 1,$$

*and for regression task,*

$$\mathbb{E}_\mathcal{D}\big[\inf_{w \in \mathbb{R}^r} \mathbb{E}_\mathcal{E}[\ell_r(\delta_{W_{CL},w})] - \inf_{w \in \mathbb{R}^r} \mathbb{E}_\mathcal{E}[\ell_r(\delta_{U^\star,w})]\big] = O\left(\frac{r^{3/2}}{d}\log d + \sqrt{\frac{dr}{n}}\right).$$

We obtain the lower bound for the downstream predction risk with autoencoders.

**Theorem B.7.** *Suppose the conditions in Theorem 3.1 hold. Suppose $r \leq r_c$ for some constant $r_c > 0$. Additionally assume that $\rho = \Theta(1)$ is sufficiently small and $n \gg d \gg r$. For classification task, assume $F$ is differentiable at 0 and $F'(0) > 0$. Then,*

$$\mathbb{E}_\mathcal{D}\big[\inf_{w \in \mathbb{R}^r} \mathbb{E}_\mathcal{E}[\ell_c(\delta_{U_{AE},w})] - \inf_{w \in \mathbb{R}^r} \mathbb{E}_\mathcal{E}[\ell_c(\delta_{U^\star,w})]\big] \gtrsim 1.$$

*For regression task,*

$$\mathbb{E}_\mathcal{D}\big[\inf_{w \in \mathbb{R}^r} \mathbb{E}_\mathcal{E}[\ell_r(\delta_{U_{AE},w})] - \inf_{w \in \mathbb{R}^r} \mathbb{E}_\mathcal{E}[\ell_r(\delta_{U^\star,w})]\big] \gtrsim 1.$$

For two matrices $A$ and $B$ of the same order, we define $A \succeq B$ when $A - B$ is positive semi-definite.

The proofs of Theorem B.6 and B.7 relies on Lemma B.8, B.9, B.10, B.11 and B.12 which are proved later in this section.

*Proof of Theorem B.6: Classification Task Part.* Lemma B.10 gives for any $U \in \mathbb{O}_{d,r}$,

$$\mathbb{E}_\mathcal{D}\big[\inf_{w \in \mathbb{R}^r} \mathcal{R}_c(\delta_{U,w}) - \inf_{w \in \mathbb{R}^r} \mathcal{R}_c(\delta_{U^\star,w})\big]$$
$$\leq ((\kappa(1 + \rho^2))^3 + \kappa\rho^2(1 + \rho^{-2})^2 + (\kappa\rho^2 \vee 1)^{-1})\mathbb{E}_\mathcal{D}[\|\sin\Theta(U, U^\star)\|_2].$$

Substituting $U \leftarrow U_{AE}$ combined with Assumption 3.2 and $\kappa = O(1)$ concludes the proof. $\qquad\square$

*Proof of Theorem B.6: Regression Part.* Note that under Assumption 3.2 and $\kappa = O(1)$, $(1 + \rho^{-2})/(1 + \kappa^{-1}\rho^{-2})^2 = O(1)$. Lemma B.12 gives for any $U \in \mathbb{O}_{d,r}$,

$$\mathbb{E}_\mathcal{D}\big[\inf_{w \in \mathbb{R}^r} \mathcal{R}_r(\delta_{U,w}) - \inf_{w \in \mathbb{R}^r} \mathcal{R}_r(\delta_{U^\star,w})\big] = O\big((1 + \rho^{-2})\mathbb{E}_\mathcal{D}[\|\sin\Theta(U, U^\star)\|_2]\|w^\star\|^2\big).$$

Theorem 3.2 with substitution $U \leftarrow U_{AE}$ gives the desired result. $\qquad\square$

*Proof of Theorem B.7: Classification Part.* Lemma B.9 gives that for $c_1 := 1 - 1/(2\kappa r_c) \in (0, 1)$, we can take $n \gg d \gg r$ and sufficiently small $\rho > 0$ so that $\mathbb{E}_{\mathcal{D}}[\|\sin\Theta(U_{AE}, U^\star)\|_F^2] \geq c_1 r$ holds. By Lemma B.11,

$$\mathbb{E}_{\mathcal{D}}\big[\inf_{w\in\mathbb{R}^r} \mathcal{R}_c(\delta_{U_{AE},w}) - \inf_{w\in\mathbb{R}^r} \mathcal{R}_c(\delta_{U^\star,w})\big]$$

$$\gtrsim \frac{(1+\rho^2)^{3/2}}{(1+\kappa\rho^2)^{3/2}}\rho^2\left(\frac{1}{1+\rho^2} - \kappa(r - \|\sin\Theta(U_{AE}, U^\star)\|_F^2)\right) \tag{B.35}$$

$$\geq \frac{(1+\rho^2)^{3/2}}{(1+\kappa\rho^2)^{3/2}}\rho^2\left(\frac{1}{1+\rho^2} - \kappa(1-c_1)r\right) \tag{B.36}$$

$$\geq \frac{(1+\rho^2)^{3/2}}{(1+\kappa\rho^2)^{3/2}}\rho^2\left(\frac{1}{1+\rho^2} - \frac{1}{2}\right), \tag{B.37}$$

where the last inequality follows since $r \leq r_c$. If we further take $\rho = \Theta(1) < 1/2$, the right hand becomes a positive constant. This concludes the proof. $\qquad\square$

*Proof of Theorem B.7: Regression Part.* From proposition B.1, we have

$$\inf_{w\in\mathbb{R}^r} \mathcal{R}_r(\delta_{U_{AE},w}) - \inf_{w\in\mathbb{R}^r} \mathcal{R}_r(\delta_{U^\star,w})$$

$$= w^{\star\top}((I + (1/\nu^2)U^{\star\top}\Sigma U^\star)^{-1}$$

$$- U^{\star\top}U_{AE}(U_{AE}^\top U^\star U^{\star\top}U_{AE} + (1/\nu^2)U_{AE}^\top\Sigma U_{AE})^{-1}U_{AE}^\top U^\star)w^\star.$$

Thus from Lemma B.8,

$$\inf_{w\in\mathbb{R}^r} \mathcal{R}_r(\delta_{U_{AE},w}) - \inf_{w\in\mathbb{R}^r} \mathcal{R}_r(\delta_{U^\star,w})$$

$$\geq \left(\frac{1}{1+\rho^{-2}} + \rho^2\kappa\big(\|\sin\Theta(U_{AE}, U^\star)\|_F^2 - r\big)\right)\|w^\star\|^2.$$

Using Lemma B.9 and by the same argument in the proof of Theorem B.7: Classification Part, we conclude the proof. $\qquad\square$

**Lemma B.5.** *For any* $U \in \mathbb{O}_{d,r}$,

$$\lambda_{\min}(\nu^2 U^{\star\top}U(U^\top\Sigma_x U)^{-1}U^\top U^\star) \geq \frac{\nu^2}{\nu^2 + \sigma_{(1)}^2}(1 - \|\sin\Theta(U, U^\star)\|_2^2).$$

*Proof.* Since $\lambda_{\min}(AC) \geq \lambda_{\min}(A)\lambda_{\min}(C)$ for symmetric positive semi-definite matrices $A$ and $C$,

$$\lambda_{\min}(\nu^2 U^{\star\top}U(U^\top\Sigma_x U)^{-1}U^\top U^\star)$$

$$\geq \lambda_{\min}(U^\top U^\star U^{\star\top}U)\lambda_{\min}(\nu^2(U^\top\Sigma_x U)^{-1})$$

$$\geq \lambda_{\min}(I - (I - U^\top U^\star U^{\star\top}U))\frac{\nu^2}{\lambda_{\max}(\nu^2 U^\top U^\star U^{\star\top}U + U^\top\Sigma U)}$$

$$\geq \frac{\nu^2}{\nu^2 + \sigma_{(1)}^2}(1 - \|\sin\Theta(U, U^\star)\|_2^2),$$

where we used Weyl's inequality $\lambda_{\min}(A + C) \geq \lambda_{\min}(A) - \|C\|_2$ in the second inequality. $\qquad\square$

**Lemma B.6.** *For any* $U \in \mathbb{O}_{d,r}$,

$$\lambda_{\max}(\nu^2 U^{\star\top}U(U^\top\Sigma_x U)^{-1}U^\top U^\star) \leq \frac{\nu^2}{\nu^2(1 - \|\sin\Theta(U, U^\star)\|_2) + \sigma_{(d)}^2}.$$

*Proof.* Since $\|AC\|_2 \leq \|A\|_2\|C\|_2$,

$$\lambda_{\max}(\nu^2 U^{\star\top}U(U^\top\Sigma_x U)^{-1}U^\top U^\star) \leq \lambda_{\max}(\nu^2(U^\top\Sigma_x U)^{-1})$$

$$\leq \frac{\nu^2}{\lambda_{\min}(\nu^2 U^\top U^\star U^{\star\top}U + U^\top\Sigma U)}$$

$$\leq \frac{\nu^2}{\lambda_{\min}(\nu^2 I - \nu^2(I - U^\top U^\star U^{\star\top}U) + U^\top\Sigma U)}$$

$$\leq \frac{\nu^2}{\nu^2(1 - \|\sin\Theta(U, U^\star)\|_2) + \sigma_{(d)}^2},$$

where we used Weyl's inequality $\lambda_{\min}(A + C) \geq \lambda_{\min}(A) - \|C\|_2$ and $\lambda_{\min}(\nu^2 I + U^\top\Sigma U) \geq \nu^2 + \sigma_{(d)}^2$. $\square$

**Lemma B.7.** *For any $U \in \mathbb{O}_{d,r}$,*

$$\|\nu^2(U^{\star\top}\Sigma_x U^\star)^{-1} - \nu^2 U^{\star\top}U(U^\top\Sigma_x U)^{-1}U^\top U^\star\|_2$$

$$= O\left(\frac{1}{1 - \|\sin\Theta(U, U^\star)\|_2^2 + \kappa^{-1}\rho^{-2}}\frac{1 + \rho^{-2}}{1 + \kappa^{-1}\rho^{-2}}\|\sin\Theta(U, U^\star)\|_2\right).$$

*Proof.* Observe that

$$\|(U^{\star\top}\Sigma_x U^\star)^{-1} - U^{\star\top}U(U^\top\Sigma_x U)^{-1}U^\top U^\star\|_2$$

$$\leq \|(U^{\star\top}\Sigma_x U^\star)^{-1} - (U^\top\Sigma_x U)^{-1}\|_2 + \|(U^\top\Sigma_x U)^{-1} - U^{\star\top}U(U^\top\Sigma_x U)^{-1}U^\top U^\star\|_2$$

$$:= (T1) + (T2).$$

For the term $(T1)$,

$$(T1) = \|(U^\top\Sigma_x U)^{-1}(U^\top\Sigma_x U)(U^{\star\top}\Sigma_x U^\star)^{-1} - (U^\top\Sigma_x U)^{-1}(U^{\star\top}\Sigma_x U^\star)(U^{\star\top}\Sigma_x U^\star)^{-1}\|_2$$

$$\leq \|(U^\top\Sigma_x U)^{-1}\|_2\|U^\top\Sigma_x U - U^{\star\top}\Sigma_x U^\star\|_2\|(U^{\star\top}\Sigma_x U^\star)^{-1}\|_2.$$

Note

$$\|U^\top\Sigma_x U - U^{\star\top}\Sigma_x U^\star\|_2 = \|\nu^2 U^\top U^\star U^{\star\top}U - \nu^2 I + U^\top\Sigma U - U^{\star\top}\Sigma U^\star\|_2$$

$$\leq \nu^2\|\sin\Theta(U, U^\star)\|_2^2 + \|U^\top\Sigma(U - U^\star) + (U - U^\star)^\top\Sigma U^\star\|_2$$

$$\leq \nu^2\|\sin\Theta(U, U^\star)\|_2^2 + 2\sigma_{(1)}^2\|U - U^\star\|_2.$$

Also we have $\lambda_{\min}(U^\top\Sigma_x U) \geq \nu^2(1 - \|\sin\Theta(U, U^\star)\|_2^2) + \sigma_{(d)}^2$ from the proof of Lemma B.6 and $\lambda_{\min}(U^{\star\top}\Sigma_x U^\star) \geq \nu^2 + \sigma_{(d)}^2$. Therefore

$$(T1) \leq \frac{1}{(\nu^2 + \sigma_{(d)}^2)(\nu^2(1 - \|\sin\Theta(U, U^\star)\|_2^2) + \sigma_{(d)}^2)}(\nu^2\|\sin\Theta(U, U^\star)\|_2^2 + 2\sigma_{(1)}^2\|U - U^\star\|_2).$$

For the term $(T2)$,

$$(T2) = \|(U^\top\Sigma_x U)^{-1} - U^{\star\top}(U^\star + (U - U^\star))(U^\top\Sigma_x U)^{-1}(U^\star + (U - U^\star))^\top U^\star\|_2$$

$$= \| - U^{\star\top}(U - U^\star)(U^\top\Sigma_x U)^{-1} - (U^\top\Sigma_x U)^{-1}(U - U^\star)^\top U^\star$$

$$\quad - U^{\star\top}(U - U^\star)(U^\top\Sigma_x U)^{-1}(U - U^\star)^\top U^\star\|_2$$

$$\leq \frac{1}{\nu^2(1 - \|\sin\Theta(U, U^\star)\|_2^2) + \sigma_{(d)}^2}(2\|U - U^\star\|_2 + \|U - U^\star\|_2^2).$$

From Lemma A.1, $\|\sin\Theta(U, U^\star)\|_2 \leq \|U - U^\star\|_2$. Finally from these results and $\|U - U^\star\|_2^2 \leq 2\|U - U^\star\|_2$,

$$\|\nu^2(U^{\star\top}\Sigma_x U^\star)^{-1} - \nu^2 U^{\star\top}U(U^\top\Sigma_x U)^{-1}U^\top U^\star\|_2$$

$$= O\left(\frac{\nu^2}{\nu^2(1 - \|\sin\Theta(U, U^\star)\|_2^2) + \sigma_{(d)}^2}\frac{\nu^2 + \sigma_{(1)}^2}{\nu^2 + \sigma_{(d)}^2}\|U - U^\star\|_2\right).$$

Since LHS does not depend on the orthogonal transformation $U \leftarrow UO$ where $O \in \mathbb{O}_{r,r}$, we obtain

$$\|\nu^2(U^{\star\top}\Sigma_x U^\star)^{-1} - \nu^2 U^{\star\top}U(U^\top\Sigma_x U)^{-1}U^\top U^\star\|_2$$

$$= O\left(\frac{\nu^2}{\nu^2(1 - \|\sin\Theta(U, U^\star)\|_2^2) + \sigma_{(d)}^2} \frac{\nu^2 + \sigma_{(1)}^2}{\nu^2 + \sigma_{(d)}^2} \inf_{O\in\mathbb{O}_{r,r}} \|UO - U^\star\|_2\right).$$

Combined again with Lemma A.1, we obtain the desired result. $\qquad\square$

**Lemma B.8.** *For any* $U \in \mathbb{O}_{d,r}$,

$$\lambda_{\min}(\nu^2(U^{\star\top}\Sigma_x U^\star)^{-1} - \nu^2 U^{\star\top}U(U^\top\Sigma_x U)^{-1}U^\top U^\star)$$

$$\geq \frac{\nu^2}{\nu^2 + \sigma_{(1)}^2} - \frac{\nu^2}{\sigma_{(d)}^2}(r - \|\sin\Theta(U, U^\star)\|_F^2).$$

*Proof.* Observe

$$\lambda_{\min}(\nu^2(U^{\star\top}\Sigma_x U^\star)^{-1} - \nu^2 U^{\star\top}U(U^\top\Sigma_x U)^{-1}U^\top U^\star)$$
$$\geq \lambda_{\min}((I + (1/\nu^2)U^{\star\top}\Sigma U^\star)^{-1}) - \|U^{\star\top}U(U^\top U^\star U^{\star\top}U + (1/\nu^2)U^\top\Sigma U)^{-1}U^\top U^\star\|_2.$$

Since $U^\top U^\star U^{\star\top}U \succeq 0$, it follows that $(U^\top U^\star U^{\star\top}U + (1/\nu^2)U^\top\Sigma U)^{-1} \preceq \nu^2(U^\top\Sigma U)^{-1}$. Thus

$$\|U^{\star\top}U(U^\top U^\star U^{\star\top}U + (1/\nu^2)U^\top\Sigma U)^{-1}U^\top U^\star\|_2$$
$$\leq \nu^2\lambda_{\max}((U^\top\Sigma U)^{-1})\|U^{\star\top}U\|_2^2$$
$$\leq \frac{\nu^2}{\sigma_{(d)}^2}\|U^{\star\top}U\|_F^2$$
$$= \frac{\nu^2}{\sigma_{(d)}^2}(r - \|\sin\Theta(U, U^\star)\|_F^2),$$

where we used $\lambda_{\max}((U^\top\Sigma U)^{-1}) \leq 1/\lambda_{\min}(U^\top\Sigma U) \leq 1/\sigma_{(d)}^2$ and $\|\sin\Theta(U_1, U_2)\|_F^2 = r - \|U_1^\top U_2\|_F^2$ from Proposition A.1. Combined with Lemma B.6, we obtain

$$\lambda_{\min}(\nu^2(U^{\star\top}\Sigma_x U^\star)^{-1} - \nu^2 U^{\star\top}U(U^\top\Sigma_x U)^{-1}U^\top U^\star)$$

$$\geq \frac{\nu^2}{\nu^2 + \sigma_{(1)}^2} - \frac{\nu^2}{\sigma_{(d)}^2}(r - \|\sin\Theta(U, U^\star)\|_F^2).$$

$\qquad\square$

**Lemma B.9.** *Suppose the conditions in Theorem 3.1 hold. Fix* $c_1 \in (0, 1)$. *There exists a constant* $c_2 > 0$ *such that if* $\sqrt{r\log d/d} \vee \rho^2 \vee d/n < c_2$, *then*

$$\mathbb{E}_{\mathcal{D}}\|\sin\Theta(U_{AE}, U^\star)\|_F^2 \geq c_1 r,$$

*where* $c_1 \in (0, 1)$ *is a universal constant.*

*Proof.* By Cauchy-Schwartz inequality,

$$\mathbb{E}_{\mathcal{D}}\|\sin\Theta(U_{AE}, U^\star)\|_F^2 - r$$
$$\geq (\mathbb{E}_{\mathcal{D}}\|\sin\Theta(U_{AE}, U^\star)\|_F)^2 - r$$
$$= (\mathbb{E}_{\mathcal{D}}\|\sin\Theta(U_{AE}, U^\star)\|_F - \sqrt{r})(\mathbb{E}_{\mathcal{D}}\|\sin\Theta(U_{AE}, U^\star)\|_F + \sqrt{r}).$$

From Theorem 3.1, there exists a constant $c_3 > 0$ such that

$$\mathbb{E}_{\mathcal{D}}\|\sin\Theta(U^\star, U_{AE})\|_F \geq \sqrt{r} - c_3\frac{r}{\sqrt{d}}\sqrt{\log d} - c_3\sqrt{r}\left(\rho^2 + \sqrt{\frac{d}{n}} + \rho\sqrt{\frac{d}{n}}\right).$$

Therefore combined with a trivial bound $\|\sin\Theta(U_{AE}, U^\star)\|_F \leq \sqrt{r}$,

$$\mathbb{E}_{\mathcal{D}}\|\sin\Theta(U_{AE}, U^\star)\|_F^2 - r \geq -rc_3\left(\frac{r^{1/2}}{\sqrt{d}}\sqrt{\log d} + \rho^2 + \sqrt{\frac{d}{n}} + \rho\sqrt{\frac{d}{n}}\right)$$

$$\geq -rc_3\left(2\frac{r^{1/2}}{\sqrt{d}}\sqrt{\log d} \vee 6\rho^2 \vee 6\sqrt{\frac{d}{n}}\right),$$

where we used $\rho\sqrt{d/n} \leq \rho^2 \vee d/n \leq \rho^2 \vee \sqrt{d/n}$ since $d < n$. Thus we can take $c_2 = 6(1-c_1)/c_3$. This concludes the proof. □

**Lemma B.10.** *For any $U \in \mathbb{O}_{d,r}$,*

$$\mathbb{E}_{\mathcal{D}}[\inf_{w\in\mathbb{R}^r}\mathcal{R}_c(\delta_{U,w}) - \inf_{w\in\mathbb{R}^r}\mathcal{R}_c(\delta_{U^\star,w})]$$
$$\leq ((\kappa(1+\rho^2))^3 + \kappa\rho^2(1+\rho^{-2})^2 + (\kappa\rho^2 \vee 1)^{-1})\mathbb{E}_{\mathcal{D}}[\|\sin\Theta(U,U^\star)\|_2].$$

*Proof.* Recall that we are considering the class of linear classifiers $\{\delta_{U,w} : w \in \mathbb{R}^r\}$, where $\delta_{U,w}(\check{x}) = \mathbb{I}\{F(\check{x}^\top Uw) > 1/2\}$. For notational simplicity, write $\beta := Uw$ and $\beta^\star := U^\star w^\star$.

$$\mathcal{R}_c(\delta_{U,w}) = \mathbb{P}_{\mathcal{E}}(\delta_{U,w}(\check{x}) \neq \check{y}) = \mathbb{P}_{\mathcal{E}}(\check{y} = 0, F(\check{x}^\top\beta) > 1/2) + \mathbb{P}_{\mathcal{E}}(\check{y} = 1, F(\check{x}^\top\beta) \leq 1/2).$$

Since $F(0) = 1/2$ and $F$ is monotone increasing, the false positive probability becomes

$$\mathbb{P}_{\mathcal{E}}(\check{y} = 0, F(\check{x}^\top\beta) > 1/2) = \mathbb{P}_{\mathcal{E}}(\check{y} = 0, \check{x}^\top\beta > 0)$$
$$= \mathbb{E}_{\mathcal{E}}[\mathbb{E}_{\mathcal{E}}[\mathbb{I}\{\check{y} = 0\}|\check{x}, \check{z}]\mathbb{I}\{\check{x}^\top\beta > 0\}]$$
$$= \mathbb{E}_{\mathcal{E}}[(1 - F(\nu^{-1}\check{z}^\top U^{\star\top}\beta^\star))\mathbb{I}\{\check{x}^\top\beta > 0\}].$$

Write $\omega := \check{x}^\top\beta$ and $\omega^\star := \nu^{-1}\check{z}^\top U^{\star\top}\beta^\star$. From assumption, $(\omega^\star, \omega)$ jointly follows a normal distribution with mean 0. Write $v^{\star 2} := \mathrm{Var}(\omega^\star) = w^{\star\top}w^\star$, $v^2 := \mathrm{Var}(\omega) = \beta^\top\Sigma_x\beta$, where $\Sigma_x := \nu^2 U^\star U^{\star\top} + \Sigma$. Let $\tau := \mathrm{Cor}(\omega^\star, \omega) = \nu w^{\star\top}U^{\star\top}\beta/(v^\star v)$. By a formula for conditional normal distribution, we have $\omega|\omega^\star \sim N(\tau v\omega^\star/v^\star, v^2(1-\tau^2))$. This gives

$$\mathbb{P}_{\mathcal{E}}(\check{y} = 0, F(\check{x}^\top\beta) > 1/2)$$
$$= \mathbb{E}_{\mathcal{E}}[(1 - F(\omega^\star))\mathbb{I}\{\omega > 0\}]$$
$$= \mathbb{E}_{\mathcal{E}}[(1 - F(\omega^\star))\mathbb{E}_{\mathcal{E}}[\mathbb{I}\{\omega > 0\}|\omega^\star]]$$
$$= \mathbb{E}_{\mathcal{E}}[(1 - F(\omega^\star))\mathbb{P}_{\mathcal{E}}(\omega > 0|\omega^\star)]$$
$$= \mathbb{E}_{\mathcal{E}}\left[(1 - F(\omega^\star))\mathbb{P}_{\mathcal{E}}\left(\frac{\omega - \tau v\omega^\star/v^\star}{v(1-\tau^2)^{1/2}} > -\frac{\tau v\omega^\star/v^\star}{v(1-\tau^2)^{1/2}}\Big|\omega^\star\right)\right]$$
$$= \mathbb{E}_{\mathcal{E}}[(1 - F(\omega^\star))\Phi(\alpha\omega^\star/v^\star)]$$
$$= \mathbb{E}_{\mathcal{E}}[(1 - F(\omega^\star))\Phi(\alpha\omega^\star/v^\star)\mathbb{I}\{\omega^\star > 0\}] + \mathbb{E}_{\mathcal{E}}[(1 - F(\omega^\star))\Phi(\alpha\omega^\star/v^\star)\mathbb{I}\{\omega^\star < 0\}],$$

where $\Phi$ is cumulative distribution function of $N(0,1)$ and $\alpha := \tau/(1-\tau^2)^{1/2}$. We define $\Psi_F$ as $\Psi_F(s^2) := 2E_{u\sim N(0,s^2)}[F(u)\mathbb{I}\{u > 0\}]$. When $F(u) = 1/(1+e^{-u})$, $\Psi_F(s^2)$ is called the logistic-normal integral, whose analytical form is not known (Pirjol (2013)). Since a random variable $\omega^\star$ is symmetric about mean 0 and $F(u) = 1 - F(-u)$,

$$\mathbb{E}_{\mathcal{E}}[(1 - F(\omega^\star))\Phi(\alpha\omega^\star/v^\star)\mathbb{I}\{\omega^\star < 0\}] = \mathbb{E}_{\mathcal{E}}[(1 - F(-\omega^\star))(1 - \Phi(\alpha\omega^\star/v^\star))\mathbb{I}\{\omega^\star > 0\}]$$
$$= \mathbb{E}_{\mathcal{E}}[F(\omega^\star)(1 - \Phi(\alpha\omega^\star/v^\star))\mathbb{I}\{\omega^\star > 0\}].$$

Hence

$$\mathbb{P}_{\mathcal{E}}(\check{y} = 0, F(\check{x}^\top\beta) > 1/2)$$
$$= \mathbb{E}_{\mathcal{E}}[(\Phi(\alpha\omega^\star/v^\star) + F(\omega^\star) - 2F(\omega^\star)\Phi(\alpha\omega^\star/v^\star))\mathbb{I}\{\omega^\star > 0\}]$$
$$= \frac{1}{2}\Psi_F(v^{\star 2}) - \mathbb{E}_{\mathcal{E}}[(2F(\omega^\star) - 1)\Phi(\alpha\omega^\star/v^\star)\mathbb{I}\{\omega^\star > 0\}].$$

Note that the true negative probability is exactly the same as the false positive probability under our settings:

$$
\begin{aligned}
\mathbb{P}_{\mathcal{E}}(\check{y} = 1, F(\check{x}^\top \beta) \leq 1/2) &= \mathbb{E}_{\mathcal{E}}[F(\check{x}^\top \beta^\star)\mathbb{I}\{\check{x}^\top \beta \leq 0\}] \\
&= \mathbb{E}_{\mathcal{E}}[F(-\check{x}^\top \beta^\star)\mathbb{I}\{\check{x}^\top \beta \geq 0\}] \\
&= \mathbb{E}_{\mathcal{E}}[(1 - F(\check{x}^\top \beta^\star))\mathbb{I}\{\check{x}^\top \beta \geq 0\}] \\
&= \mathbb{P}_{\mathcal{E}}(\check{y} = 0, F(\check{x}^\top \beta) > 1/2).
\end{aligned}
$$

Therefore

$$
\mathcal{R}_c(\delta_{U,w}) = \Psi_F(v^{\star 2}) - 2\mathbb{E}_{\mathcal{E}}[(2F(\omega^\star) - 1)\Phi(\alpha\omega^\star/v^\star)\mathbb{I}\{\omega^\star > 0\}].
$$

Let

$$
\tau_{\max,U} := \sup_{w \in \mathbb{R}^r} \nu w^{\star\top} U^{\star\top} U w / (w^{\star\top} w^\star w^\top U^\top \Sigma_x U w)^{1/2},
$$

$$
\tau_{\max,U^\star} := \sup_{w \in \mathbb{R}^r} \nu w^{\star\top} w / (w^{\star\top} w^\star w^\top U^{\star\top} \Sigma_x U^\star w)^{1/2}.
$$

From Cauchy-Schwartz inequality,

$$
\tau_{\max,U}^2 = \frac{\nu^2 w^{\star\top} U^{\star\top} U (U^\top \Sigma_x U)^{-1} U^\top U^\star w^\star}{w^{\star\top} w^\star},
$$

$$
\tau_{\max,U^\star}^2 = \frac{\nu^2 w^{\star\top} (U^{\star\top} \Sigma_x U^\star)^{-1} w^\star}{w^{\star\top} w^\star}.
$$

Define $\alpha_{\max,U} := \tau_{\max,U}/(1 - \tau_{\max,U}^2)^{1/2}$ and $\alpha_{\max,U^\star} := \tau_{\max,U^\star}/(1 - \tau_{\max,U^\star}^2)^{1/2}$. Then, since on the event where $\omega^\star > 0$, $\alpha \mapsto \Phi(\alpha\omega^\star/v^\star)$ is monotone increasing and $2F(w^\star) - 1$ is non-negative, we have

$$
\inf_{w \in \mathbb{R}^r} \mathcal{R}_c(\delta_{U,w}) = \Psi_F(v^{\star 2}) - 2\mathbb{E}_{\mathcal{E}}[(2F(\omega^\star) - 1)\Phi(\alpha_{\max,U}\omega^\star/v^\star)\mathbb{I}\{\omega^\star > 0\}]
$$

$$
\inf_{w \in \mathbb{R}^r} \mathcal{R}_c(\delta_{U^\star,w}) = \Psi_F(v^{\star 2}) - 2\mathbb{E}_{\mathcal{E}}[(2F(\omega^\star) - 1)\Phi(\alpha_{\max,U^\star}\omega^\star/v^\star)\mathbb{I}\{\omega^\star > 0\}].
$$

This yields

$$
\begin{aligned}
&\inf_{w \in \mathbb{R}^r} \mathcal{R}_c(\delta_{U,w}) - \inf_{w \in \mathbb{R}^r} \mathcal{R}_c(\delta_{U^\star,w}) \\
&= 2\mathbb{E}_{\mathcal{E}}[(2F(\omega^\star) - 1)(\Phi(\alpha_{\max,U^\star}\omega^\star/v^\star) - \Phi(\alpha_{\max,U}\omega^\star/v^\star))\mathbb{I}\{\omega^\star > 0\}].
\end{aligned}
$$

Note that for any $a, b \geq 0$,

$$
|\Phi(b) - \Phi(a)| \leq \phi(a \wedge b)|b - a|,
$$

where $\phi$ is a density function of standard normal distribution. Observe

$$
\begin{aligned}
&\inf_{w \in \mathbb{R}^r} \mathcal{R}_c(\delta_{U,w}) - \inf_{w \in \mathbb{R}^r} \mathcal{R}_c(\delta_{U^\star,w}) \\
&\leq 2\mathbb{E}_{\mathcal{E}}[(2F(\omega^\star) - 1)|\Phi(\alpha_{\max,U^\star}\omega^\star/v^\star) - \Phi(\alpha_{\max,U}\omega^\star/v^\star)|\mathbb{I}\{\omega^\star > 0\}] \\
&\lesssim \frac{2}{v^\star} \int_0^\infty (2F(\omega^\star) - 1)|\alpha_{\max,U^\star} - \alpha_{\max,U}|\omega^\star \phi((\alpha_{\max,U^\star} \wedge \alpha_{\max,U})\omega^\star/v^\star)\frac{\phi(\omega^\star/v^\star)}{v^\star} \, \mathrm{d}\omega^\star \\
&\lesssim \frac{|\alpha_{\max,U^\star} - \alpha_{\max,U}|}{v^\star} \int_0^\infty (2F(\omega^\star) - 1)\phi((\alpha_{\max,U^\star} \wedge \alpha_{\max,U})\omega^\star/v^\star) \, \mathrm{d}\omega^\star \\
&= \frac{|\alpha_{\max,U^\star} - \alpha_{\max,U}|}{\alpha_{\max,U^\star} \wedge \alpha_{\max,U}} \int_0^\infty (2F(\omega^\star) - 1)\frac{\exp(-1/(2((\alpha_{\max,U^\star} \wedge \alpha_{\max,U})^{-2}v^{\star 2}))\omega^{\star 2})}{\sqrt{2\pi((\alpha_{\max,U^\star} \wedge \alpha_{\max,U})^{-2}v^{\star 2})}} \, \mathrm{d}\omega^\star \\
&= \frac{|\alpha_{\max,U^\star} - \alpha_{\max,U}|}{\alpha_{\max,U^\star} \wedge \alpha_{\max,U}} (\Psi_F(((\alpha_{\max,U^\star} \wedge \alpha_{\max,U^\star})^{-2}v^{\star 2})) - 1/2),
\end{aligned}
$$

where we used $\sup_{u>0} u\phi(u) < \infty$. Since $(a - b) = (a^2 - b^2)/(a + b) \leq (a^2 - b^2)/(a \wedge b)$ for $a, b > 0$, and $\Psi_F \leq 1$, we obtain

$$
\inf_{w \in \mathbb{R}^r} \mathcal{R}_c(\delta_{U,w}) - \inf_{w \in \mathbb{R}^r} \mathcal{R}_c(\delta_{U^\star,w}) \lesssim \frac{|\alpha_{\max,U^\star}^2 - \alpha_{\max,U}^2|}{\alpha_{\max,U^\star}^2 \wedge \alpha_{\max,U}^2}.
$$

When $\tau_{\max, U^\star} \geq \tau_{\max, U}$, since $\tau \mapsto \tau^2/(1 - \tau^2)$ is increasing in $\tau > 0$,

$$\inf_{w \in \mathbb{R}^r} \mathcal{R}_c(\delta_{U,w}) - \inf_{w \in \mathbb{R}^r} \mathcal{R}_c(\delta_{U^\star,w}) \lesssim \frac{\alpha_{\max,U^\star}^2 - \alpha_{\max,U}^2}{\alpha_{\max,U}^2}$$

$$= \frac{\tau_{\max,U^\star}^2 - \tau_{\max,U}^2}{(1 - \tau_{\max,U^\star}^2)\tau_{\max,U}^2}. \tag{B.38}$$

From Lemma B.5 and B.6, we have

$$\frac{\nu^2}{\nu^2 + \sigma_{(1)}^2}(1 - \|\sin\Theta(U,U^\star)\|_2^2) \leq \tau_{\max,U}^2 \leq \frac{\nu^2}{\nu^2(1 - \|\sin\Theta(U,U^\star)\|_2^2) + \sigma_{(d)}^2},$$

$$\frac{\nu^2}{\nu^2 + \sigma_{(1)}^2} \leq \tau_{\max,U^\star}^2 \leq \frac{\nu^2}{\nu^2 + \sigma_{(d)}^2}. \tag{B.39}$$

Then, equation B.38 becomes

$$\inf_{w \in \mathbb{R}^r} \mathcal{R}_c(\delta_{U,w}) - \inf_{w \in \mathbb{R}^r} \mathcal{R}_c(\delta_{U^\star,w})$$

$$\lesssim \frac{\nu^2 + \sigma_{(d)}^2}{\sigma_{(d)}^2} \frac{\nu^2 + \sigma_{(1)}^2}{\nu^2(1 - \|\sin\Theta(U,U^\star)\|_2^2)}(\tau_{\max,U^\star}^2 - \tau_{\max,U}^2)$$

$$\leq \frac{\nu^2 + \sigma_{(d)}^2}{\sigma_{(d)}^2} \frac{\nu^2 + \sigma_{(1)}^2}{\nu^2(1 - \|\sin\Theta(U,U^\star)\|_2^2)}\|\nu^2(U^{\star\top}\Sigma_x U^\star)^{-1} - \nu^2 U^{\star\top}U(U^\top\Sigma_x U)^{-1}U^\top U^\star\|_2$$

$$\leq \frac{(\kappa\rho^2 + 1)(\rho^{-2} + 1)^2}{(1 + \kappa^{-1}\rho^{-2})(1 - \|\sin\Theta(U,U^\star)\|_2^2)^2}\|\sin\Theta(U,U^\star)\|_2$$

$$= \frac{\kappa\rho^2(\rho^{-2} + 1)^2}{(1 - \|\sin\Theta(U,U^\star)\|_2^2)^2}\|\sin\Theta(U,U^\star)\|_2.$$

where the last inequality follows from Lemma B.7.

On the event where $\|\sin\Theta(U,U^\star)\|_2^2 \leq 1/2$,

$$\inf_{w \in \mathbb{R}^r} \mathcal{R}_c(\delta_{U,w}) - \inf_{w \in \mathbb{R}^r} \mathcal{R}_c(\delta_{U^\star,w}) \lesssim \kappa\rho^2(1 + \rho^{-2})^2\|\sin\Theta(U,U^\star)\|_2.$$

When $\tau_{\max,U^\star} < \tau_{\max,U}$, on the event where $\|\sin\Theta(U,U^\star)\|_2 \leq \kappa^{-1}\rho^{-2}/2$,

$$\inf_{w \in \mathbb{R}^r} \mathcal{R}_c(\delta_{U,w}) - \inf_{w \in \mathbb{R}^r} \mathcal{R}_c(\delta_{U^\star,w})$$

$$\lesssim \frac{\nu^2 + \sigma_{(1)}^2}{\nu^2} \frac{\nu^2(1 - \|\sin\Theta(U,U^\star)\|_2^2) + \sigma_{(d)}^2}{-\nu^2\|\sin\Theta(U,U^\star)\|_2^2 + \sigma_{(d)}^2}(\tau_{\max,U}^2 - \tau_{\max,U^\star}^2)$$

$$\leq \frac{(\nu^2 + \sigma_{(1)}^2)^2}{\nu^2} \frac{1}{-\nu^2\|\sin\Theta(U,U^\star)\|_2^2 + \sigma_{(d)}^2}$$

$$\times \|\nu^2(U^{\star\top}\Sigma_x U^\star)^{-1} - \nu^2 U^{\star\top}U(U^\top\Sigma_x U)^{-1}U^\top U^\star\|_2$$

$$\leq \frac{(1 + \rho^{-2})^3}{(-\|\sin\Theta(U,U^\star)\|_2^2 + \kappa^{-1}\rho^{-2})^3}\|\sin\Theta(U,U^\star)\|_2$$

$$\lesssim (\kappa(1 + \rho^2))^3\|\sin\Theta(U,U^\star)\|_2,$$

where we used Lemma B.7 again.

In summary, on the event where $\|\sin\Theta(U,U^\star)\|_2 \leq \kappa^{-1}\rho^{-2}/2 \wedge 1/2$,

$$\inf_{w \in \mathbb{R}^r} \mathcal{R}_c(\delta_{U,w}) - \inf_{w \in \mathbb{R}^r} \mathcal{R}_c(\delta_{U^\star,w})$$

$$\lesssim ((\kappa(1 + \rho^2))^3 + \kappa\rho^2(1 + \rho^{-2})^2)\|\sin\Theta(U,U^\star)\|_2.$$

On the other hand, on the event where $\|\sin\Theta(U, U^\star)\|_2 > \kappa^{-1}\rho^{-2}/2 \wedge 1/2$, we have a trivial inequality $\inf_{w\in\mathbb{R}^r}\mathcal{R}_c(\delta_{U,w}) - \inf_{w\in\mathbb{R}^r}\mathcal{R}_c(\delta_{U^\star,w}) \leq 1$. This gives

$$\mathbb{E}_\mathcal{D}\big[\inf_{w\in\mathbb{R}^r}\mathcal{R}_c(\delta_{U,w}) - \inf_{w\in\mathbb{R}^r}\mathcal{R}_c(\delta_{U^\star,w})\big]$$
$$\lesssim ((\kappa(1+\rho^2))^3 + \kappa\rho^2(1+\rho^{-2})^2)\mathbb{E}_\mathcal{D}[\|\sin\Theta(U, U^\star)\|_2]$$
$$+ \mathbb{P}_\mathcal{D}(\|\sin\Theta(U, U^\star)\|_2 > \kappa^{-1}\rho^{-2}/2 \wedge 1/2)$$
$$\lesssim ((\kappa(1+\rho^2))^3 + \kappa\rho^2(1+\rho^{-2})^2 + (\kappa\rho^2 \vee 1))\mathbb{E}_\mathcal{D}[\|\sin\Theta(U, U^\star)\|_2],$$

where the last inequality follows from Markov's inequality. $\qquad\square$

**Lemma B.11.** *Suppose $U \in \mathbb{O}_{d,r}$ satisfies $1/(1+\rho^2) - \kappa(r - \|\sin\Theta(U, U^\star)\|_F^2) \geq 0$. Then,*

$$\inf_{w\in\mathbb{R}^r}\mathcal{R}_c(\delta_{U,w}) - \inf_{w\in\mathbb{R}^r}\mathcal{R}_c(\delta_{U^\star,w})$$
$$\gtrsim \frac{(1+\rho^2)^{3/2}}{(1+\kappa\rho^2)^{3/2}}\rho^2\left(\frac{1}{1+\rho^2} - \kappa(r - \|\sin\Theta(U, U^\star)\|_F^2)\right).$$

*Proof.* We firstly bound the term $\tau_{\max,U^\star}^2 - \tau_{\max,U}^2$. From Lemma B.8,

$$\tau_{\max,U^\star}^2 - \tau_{\max,U}^2 \geq \lambda_{\min}(\nu^2(U^{\star\top}\Sigma_x U^\star)^{-1} - \nu^2 U^{\star\top}U(U^\top\Sigma_x U)^{-1}U^\top U^\star)$$
$$\geq \frac{\nu^2}{\nu^2 + \sigma_{(1)}^2} - \frac{\nu^2}{\sigma_{(d)}^2}(r - \|\sin\Theta(U, U^\star)\|_F^2). \tag{B.40}$$

From assumption, RHS of equation B.40 is non-negative. Then using the inequality $a - b = (a^2 - b^2)/(a + b) \geq (a^2 - b^2)/(2a)$ for $a \geq b \geq 0$,

$$\alpha_{\max,U^\star} - \alpha_{\max,U} \gtrsim \frac{1}{\alpha_{\max,U^\star}}(\alpha_{\max,U^\star}^2 - \alpha_{\max,U^\star}^2)$$
$$\geq \frac{(1 - \tau_{\max,U^\star}^2)^{1/2}}{\tau_{\max,U^\star}}\frac{\tau_{\max,U^\star}^2 - \tau_{\max,U}^2}{(1 - \tau_{\max,U^\star}^2)(1 - \tau_{\max,U}^2)}.$$

From equation B.39 and equation B.40,

$$\alpha_{\max,U^\star} - \alpha_{\max,U}$$
$$\gtrsim \left(\frac{\nu^2 + \sigma_{(d)}^2}{\nu^2}\right)^{1/2}\left(\frac{\nu^2 + \sigma_{(1)}^2}{\sigma_{(1)}^2}\right)^{3/2}\left(\frac{\nu^2}{\nu^2 + \sigma_{(1)}^2} - \frac{\nu^2}{\sigma_{(d)}^2}(r - \|\sin\Theta(U, U^\star)\|_F^2)\right)$$
$$= (1 + \kappa^{-1}\rho^{-2})^{1/2}(1 + \rho^2)^{3/2}\left(\frac{\nu^2}{\nu^2 + \sigma_{(1)}^2} - \frac{\nu^2}{\sigma_{(d)}^2}(r - \|\sin\Theta(U, U^\star)\|_F^2)\right). \tag{B.41}$$

From the proof of Lemma B.10,

$$\inf_{w\in\mathbb{R}^r}\mathcal{R}_c(\delta_{U,w}) - \inf_{w\in\mathbb{R}^r}\mathcal{R}_c(\delta_{U^\star,w})$$
$$= 2\mathbb{E}_\mathcal{E}[(2F(\omega^\star) - 1)(\Phi(\alpha_{\max,U^\star}\omega^\star/v^\star) - \Phi(\alpha_{\max,U}\omega^\star/v^\star))\mathbb{I}\{\omega^\star > 0\}].$$

Note that for any $b \geq a \geq 0$, $\Phi(b) - \Phi(a) \geq \phi(b)(b - a)$. Since we assume RHS of equation B.40 is positive, $\alpha_{\max,U^\star} \geq \alpha_{\max,U}$. Thus on the event where $\omega^\star > 0$, $\alpha_{\max,U^\star}\omega^\star/v^\star \geq \alpha_{\max,U}\omega^\star/v^\star$. Observe

$$\inf_{w\in\mathbb{R}^r}\mathcal{R}_c(\delta_{U,w}) - \inf_{w\in\mathbb{R}^r}\mathcal{R}_c(\delta_{U^\star,w})$$
$$\geq 2\mathbb{E}_\mathcal{E}[(2F(\omega^\star) - 1)\phi(\alpha_{\max,U^\star}\omega^\star/v^\star)(\alpha_{\max,U^\star}\omega^\star/v^\star - \alpha_{\max,U}\omega^\star/v^\star)\mathbb{I}\{\omega^\star > 0\}]$$
$$= \frac{2}{v^\star}(\alpha_{\max,U^\star} - \alpha_{\max,U})\int_0^\infty (2F(\omega^\star) - 1)\omega^\star\frac{\phi(\omega^\star/v^\star)}{v^\star}\phi(\alpha_{\max,U^\star}\omega^\star/v^\star)\,\mathrm{d}\omega^\star$$
$$\simeq \frac{\alpha_{\max,U^\star} - \alpha_{\max,U}}{v^\star}\int_0^\infty (2F(\omega^\star) - 1)\omega^\star\exp\left(-(1/2)(1 + \alpha_{\max,U^\star}^2)\omega^{\star 2}/v^{\star 2}\right)\mathrm{d}\omega^\star$$
$$\simeq \frac{\alpha_{\max,U^\star} - \alpha_{\max,U}}{1 + \alpha_{\max,U^\star}^2}\int_0^\infty (2F((1 + \alpha_{\max,U^\star}^2)^{-1/2}v^\star\omega^\star) - 1)\omega^\star\exp\left(-(1/2)\omega^{\star 2}\right)\mathrm{d}\omega^\star,$$

where in the last equality we transformed $w^\star \to (1 + \alpha_{\max,U^\star}^2)^{1/2} w^\star / v^\star$. Since $F(u)$ is differentiable at 0 and $F(0) = 1/2$,

$$F(u) - 1/2 = F'(0)u + o(u).$$

Thus there exists a constant $\epsilon > 0$ only depending on $F$ such that $2(F(u) - 1/2) \geq F'(0)u$ for all $u \in [0, \epsilon]$ since $F'(0) > 0$. This gives

$$\inf_{w \in \mathbb{R}^r} \mathcal{R}_c(\delta_{U,w}) - \inf_{w \in \mathbb{R}^r} \mathcal{R}_c(\delta_{U^\star,w})$$

$$\gtrsim \frac{\alpha_{\max,U^\star} - \alpha_{\max,U}}{1 + \alpha_{\max,U^\star}^2} F'(0)(1 + \alpha_{\max,U^\star}^2)^{-1/2} v^\star$$

$$\times \int_0^{\epsilon(1+\alpha_{\max,U^\star}^2)^{1/2} v^\star} \omega^{\star 2} \exp\left(-(1/2)\omega^{\star 2}\right) d\omega^\star$$

$$\gtrsim \frac{\alpha_{\max,U^\star} - \alpha_{\max,U}}{1 + \alpha_{\max,U^\star}^2} (1 + \alpha_{\max,U^\star}^2)^{-1/2} v^\star \int_0^{\epsilon v^\star} \omega^{\star 2} \exp\left(-(1/2)\omega^{\star 2}\right) d\omega^\star$$

$$\gtrsim \frac{\alpha_{\max,U^\star} - \alpha_{\max,U}}{1 + \alpha_{\max,U^\star}^2} (1 + \alpha_{\max,U^\star}^2)^{-1/2}.$$

The last inequality follows since $v^\star = \|w^\star\| = 1$ by assumption. It is noted that $\alpha_{\max,U^\star}^2 \leq \nu^2/\sigma_{(d)}^2$ from equation B.39. Therefore with equation B.41,

$$\inf_{w \in \mathbb{R}^r} \mathcal{R}_c(\delta_{U,w}) - \inf_{w \in \mathbb{R}^r} \mathcal{R}_c(\delta_{U^\star,w})$$

$$\gtrsim \frac{1}{(1 + \kappa\rho^2)^{3/2}} (1 + \kappa^{-1}\rho^{-2})^{1/2}(1 + \rho^2)^{3/2}\left(\frac{1}{1 + \rho^{-2}} - \kappa\rho^2(r - \|\sin\Theta(U, U^\star)\|_F^2)\right)$$

$$\gtrsim \frac{(1 + \rho^2)^{3/2}}{(1 + \kappa\rho^2)^{3/2}} \rho^2\left(\frac{1}{1 + \rho^2} - \kappa(r - \|\sin\Theta(U, U^\star)\|_F^2)\right).$$

$\square$

**Proposition B.1.** *For any* $U \in \mathbb{O}_{d,r}$,

$$\inf_{w \in \mathbb{R}^r} \mathcal{R}_r(\delta_{U,w}) = \nu^2 w^{\star\top}(I - \nu^2 U^{\star\top} U(\nu^2 U^\top U^\star U^{\star\top} U + U^\top \Sigma U)^{-1} U^\top U^\star)w^\star + \sigma_\epsilon^2.$$

*Proof of Proposition B.1.* Generate random variables $(\check{x}, \check{z}, \check{\xi}, \check{\epsilon})$ following the model equation B.34. We calculate the prediction risk of $\delta_{U,w}$ as:

$$\mathcal{R}_r(\delta_{U,w}) := \mathbb{E}_{\mathcal{E}}(\check{y} - \check{x}^\top U w)^2$$

$$= \text{Var}_{\mathcal{E}}(\nu^{-1}\check{z}^\top w^\star + \check{\epsilon})^2 - 2\text{Cov}_{\mathcal{E}}(\nu^{-1}\check{z}^\top w^\star + \check{\epsilon}, U^\star\check{z} + \check{\xi})Uw$$

$$+ w^\top U^\top \text{Var}_{\mathcal{E}}(U^\star\check{z} + \check{\xi})Uw$$

$$= \|w^\star\|^2 + \sigma_\epsilon^2 - 2\nu w^{\star\top} U^{\star\top} Uw + w^\top(\nu^2 U^\top U^\star U^{\star\top} U + U^\top \Sigma U)w$$

$$= (w - A^{-1}b)^\top A(w - A^{-1}b) - b^\top A^{-1}b + \|w^\star\|^2 + \sigma_\epsilon^2,$$

where $A := \nu^2 U^\top U^\star U^{\star\top} U + U^\top \Sigma U$ and $b := \nu U^\top U^\star w^\star$. From this, we obtain

$$\inf_{w \in \mathbb{R}^r} \mathcal{R}_r(\delta_{U,w}) = w^{\star\top}\left(I - U^{\star\top} U(U^\top U^\star U^{\star\top} U + (1/\nu^2)U^\top \Sigma U)^{-1} U^\top U^\star\right)w^\star + \sigma_\epsilon^2.$$

$\square$

**Lemma B.12.** *For any* $U \in \mathbb{O}_{d,r}$,

$$\mathbb{E}_{\mathcal{D}}\left[\inf_{w \in \mathbb{R}^r} \mathcal{R}_r(\delta_{U,w}) - \inf_{w \in \mathbb{R}^r} \mathcal{R}_r(\delta_{U^\star,w})\right] = O\left((1 + \rho^{-2})\mathbb{E}_{\mathcal{D}}[\|\sin\Theta(U, U^\star)\|_2]\|w^\star\|^2\right).$$

*Proof of Lemma B.12.* From proposition B.1, we have

$$\inf_{w \in \mathbb{R}^r} \mathcal{R}_r(\delta_{U,w}) - \inf_{w \in \mathbb{R}^r} \mathcal{R}_r(\delta_{U^\star,w})$$

$$= w^{\star\top}\left((I + (1/\nu^2)U^{\star\top}\Sigma U^\star)^{-1} - U^{\star\top} U(U^\top U^\star U^{\star\top} U + (1/\nu^2)U^\top \Sigma U)^{-1} U^\top U^\star\right)w^\star.$$

Note that $\inf_{w \in \mathbb{R}^r} \mathcal{R}_r(\delta_{U,w}) - \inf_{w \in \mathbb{R}^r} \mathcal{R}_r(\delta_{U^\star,w}) \equiv \inf_{w \in \mathbb{R}^r} \mathcal{R}_r(\delta_{UO,w}) - \inf_{w \in \mathbb{R}^r} \mathcal{R}_r(\delta_{U^\star,w})$ for any orthogonal matrix $O \in \mathbb{O}_{r,r}$. Take $\tilde{O} \in \mathbb{O}_{r,r}$ such that $\|U\tilde{O} - U^\star\|_2 \leq \sqrt{2}\|\sin\Theta(U, O)\|_2$ without loss of generality, since we can always take a sequence $(\tilde{O}_m)_{m \geq 1}$ such that $\|UO_m - U^\star\|_2 \leq \sqrt{2}\|\sin\Theta(U, O)\|_2 + 1/m$ from Lemma A.1.

Lemma B.7 gives

$$\inf_{w \in \mathbb{R}^r} \mathcal{R}_r(\delta_{U,w}) - \inf_{w \in \mathbb{R}^r} \mathcal{R}_r(\delta_{U^\star,w})$$
$$= O\left(\frac{1}{1 - \|\sin\Theta(U, U^\star)\|_2^2 + \kappa^{-1}\rho^{-2}} \frac{1 + \rho^{-2}}{1 + \kappa^{-1}\rho^{-2}}\|\sin\Theta(U, U^\star)\|_2\|w^\star\|^2\right).$$

On the event where $\|\sin\Theta(U, U^\star)\|_2^2 < 1/2$,

$$\inf_{w \in \mathbb{R}^r} \mathcal{R}_r(\delta_{U,w}) - \inf_{w \in \mathbb{R}^r} \mathcal{R}_r(\delta_{U^\star,w}) = O\left(\frac{1 + \rho^{-2}}{(1 + \kappa^{-1}\rho^{-2})^2}\|\sin\Theta(U, U^\star)\|_2\|w^\star\|^2\right).$$

On the event where $\|\sin\Theta(U, U^\star)\|_2^2 \geq 1/2$, we utlize the trivial upper bound

$$\inf_{w \in \mathbb{R}^r} \mathcal{R}_r(\delta_{U,w}) - \inf_{w \in \mathbb{R}^r} \mathcal{R}_r(\delta_{U^\star,w}) \leq \|(I + \nu^{-2}U^{\star\top}\Sigma U^\star)^{-1}\|_2\|w^\star\|^2 \leq \frac{\nu^2}{\nu^2 + \sigma_{(d)}^2}\|w^\star\|^2.$$

Combining these results, we have

$$\mathbb{E}_\mathcal{D}\big[\inf_{w \in \mathbb{R}^r} \mathcal{R}_r(\delta_{U,w}) - \inf_{w \in \mathbb{R}^r} \mathcal{R}_r(\delta_{U^\star,w})\big]$$
$$\lesssim \frac{1 + \rho^{-2}}{(1 + \kappa^{-1}\rho^{-2})^2}\mathbb{E}_\mathcal{D}[\|\sin\Theta(U, U^\star)\|_2]\|w^\star\|^2$$
$$+ \frac{1}{1 + \kappa^{-1}\rho^{-2}}\|w^\star\|^2\mathbb{P}_\mathcal{D}(\|\sin\Theta(U, U^\star)\|_2 \geq 1/\sqrt{2})$$
$$\lesssim \frac{1 + \rho^{-2}}{(1 + \kappa^{-1}\rho^{-2})^2}\mathbb{E}_\mathcal{D}[\|\sin\Theta(U, U^\star)\|_2]\|w^\star\|^2,$$

where the last inequality follows from Markov's inequality. $\qquad\square$

## C  OMITTED PROOFS FOR SECTION 4

### C.1  PROOFS FOR SECTION 4.1

In this section, we will provide the proof of a generalized version of Theorem 4.1 to cover the imbalanced setting, the statement and the detailed proof can be found in Theorem C.2. In the main body, we assume the unlabeled data and labeled data are both balanced for the sake of clarity and simplicity. Now we allow them to be imbalanced and provide a more general analysis. Suppose we have $n$ unlabeled data $X = [x_1, \cdots, x_n] \in \mathbb{R}^{d \times n}$ and $n_k$ labeled data $X_k = [x_k^1, \cdots, x_k^{n_k}] \in \mathbb{R}^{d \times n_k}$ for class $k$, the contrastive learning task can be formulated as:

$$\min_{W \in \mathbb{R}^{r \times d}} \mathcal{L}(W) := \min_{W \in \mathbb{R}^{r \times d}} \mathcal{L}_{\text{SelfCon}}(W) + \mathcal{L}_{\text{SupCon}}(W). \tag{C.1}$$

In addition, we write a generalized version of supervised contrastive loss function to cover the imbalanced cases:

$$\mathcal{L}_{\text{SupCon}}(W) = -\frac{1}{r+1}\sum_{k=1}^{r+1}\frac{\alpha_k}{n_k}\sum_{i=1}^{n_k}\left[\sum_{j \neq i}\frac{\langle Wx_i^k, Wx_j^k\rangle}{n_k - 1} - \frac{\sum_{j=1}^n\sum_{s \neq k}\langle Wx_i^k, Wx_j^s\rangle}{\sum_{s \neq k} n_s}\right] + \frac{\lambda}{2}\|WW^\top\|_F^2,$$
$$\tag{C.2}$$

where $\alpha_k > 0$ is the weight for supervised loss of class $k$. Again we first provide a theorem to give the optimal solution of contrastive learning problem.

**Theorem C.1.** *The optimal solution of supervised contrastive learning problem (C.1) is given by :*

$$W_{CL} = C\left(\sum_{i=1}^r u_i \sigma_i v_i^\top\right)^\top,$$

*where $C > 0$ is a positive constant, $\sigma_i$ is the $i$-th largest eigenvalue of the following matrix:*

$$\frac{1}{4n}\left(\Delta(XX^\top) - \frac{1}{n-1}X(1_n1_n^\top - I_n)X^\top\right)$$
$$+ \frac{1}{r+1}\sum_{k=1}^{r+1}\frac{\alpha_k}{n_k}\left[\frac{1}{n_k-1}X_k(1_{n_k}1_{n_k}^\top - I_{n_k})X_k^\top - \frac{1}{\sum_{t\neq k}n_t}X_k1_k1_s^\top X_s^\top\right],$$

*$u_i$ is the corresponding eigenvector and $V = [v_1, \cdots, v_n] \in \mathbb{R}^{r\times r}$ can be any orthonormal matrix.*

*Proof.* Under this setting, combine with the result obtained in Corollary B.1, the contrastive loss can be rewritten as:

$$\mathcal{L}(W) = \frac{\lambda}{2}\|WW^\top\|_F^2 - \frac{1}{2n}\operatorname{tr}\left(\left(\frac{1}{2}\Delta(XX^\top) - \frac{1}{2(n-1)}X(1_n1_n^\top - I_n)X^\top\right)W^\top W\right)$$
$$- \frac{1}{r+1}\sum_{k=1}^{r+1}\alpha_k\frac{1}{n_k}\sum_{i=1}^{n_k}\left[\frac{1}{n_k-1}\sum_{j\neq i}\langle Wx_i^k, Wx_j^k\rangle - \frac{1}{\sum_{t\neq k}n_t}\sum_{s\neq k}\sum_{j=1}^{n_s}\langle Wx_i^k, Wx_j^s\rangle\right].$$

Then we deal with the last term independently, note that:

$$\sum_{i=1}^{n_k}\left[\frac{1}{n_k-1}\sum_{j\neq i}\langle Wx_i^k, Wx_j^k\rangle - \frac{1}{\sum_{t\neq k}n_t}\sum_{s\neq k}\sum_{j=1}^{n_s}\langle Wx_i^k, Wx_j^s\rangle\right]$$
$$= \frac{1}{n_k-1}\sum_{i=1}^{n_k}\sum_{j\neq i}\langle Wx_i^k, Wx_j^k\rangle - \frac{1}{\sum_{t\neq k}n_t}\sum_{i=1}^{n_k}\sum_{s\neq k}\sum_{j=1}^{n_s}\langle Wx_i^k, Wx_j^s\rangle$$
$$= \frac{1}{n_k-1}\operatorname{tr}\left(X_k(1_{n_k}1_{n_k}^\top - I_{n_k})X_k^\top W^\top W\right) - \frac{1}{\sum_{t\neq k}n_t}\sum_{s\neq k}\operatorname{tr}\left(X_k1_k1_s^\top X_s^\top W^\top W\right).$$

Thus we have:

$$\mathcal{L}(W) = \frac{\lambda}{2}\|WW^\top\|_F^2 - \frac{1}{4n}\operatorname{tr}\left((\Delta(XX^\top) - \frac{1}{n-1}X(1_n1_n^\top - I_n)X^\top)W^\top W\right)$$
$$- \frac{1}{r+1}\sum_{k=1}^{r+1}\frac{\alpha_k}{n_k}[\frac{1}{n_k-1}\operatorname{tr}\left(X_k(1_{n_k}1_{n_k}^\top - I_{n_k})X_k^\top W^\top W\right)$$
$$- \frac{1}{\sum_{t\neq k}n_t}\sum_{s\neq k}\operatorname{tr}\left(X_k1_k1_s^\top X_s^\top W^\top W\right)].$$

Then by similar argument as in the proof of Theorem B.2, we can conclude that the optimal solution $W_{CL}$ must satisfy the desired conditions. $\quad\square$

With optimal solution obtained in Theorem C.1, we can provide a generalized version of Theorem 4.1 to cover the imbalance cases.

**Theorem C.2** (Generalized version of Theorem 4.1). *If Assumption 3.1-3.3 hold, $n > d \gg r$ and let $W_{CL}$ be any solution that minimizes the supervised contrastive learning problem in Eq.(C.1), and denote its singular value decomposition as $W_{CL} = (U_{CL}\Sigma_{CL}V_{CL}^\top)^\top$, then we have*

$$\mathbb{E}\|\sin\Theta(U_{CL}, U)\|_F \lesssim \frac{\nu^2}{\lambda_r(T)}\left(\frac{r^{3/2}}{d}\log d + \sqrt{\frac{dr}{n}}\right.$$
$$\left. + \frac{1}{r+1}\sum_{k=1}^{r+1}\alpha_k\left[\sum_{s\neq k}\frac{\sqrt{n_sd}}{\sum_{t\neq k}n_t}(\sqrt{\frac{d}{n_k}} + \sqrt{r}) + \sqrt{\frac{dr}{n_k}}\right]\right),$$

*where $T \triangleq \frac{1}{4}\sum_{k=1}^{r+1}p_i\mu_k\mu_k^\top + \frac{1}{r+1}\sum_{k=1}^{r+1}\alpha_k(\mu_k\mu_k^\top - \sum_{s\neq k}\frac{n_s}{\sum_{t\neq k}n_t}\frac{1}{2}(\mu_k\mu_s^\top + \mu_s\mu_k^\top))$.*

*Proof.* For labeled data $X = [x_1, \cdots, x_n]$, we write it to be $X = M + E$, where $M = [\mu_1, \cdots, \mu_n]$ and $E = [\xi_1, \cdots, \xi_n]$ are two matrices consisting of class mean and random noise. To be more specific, if $x_i$ subject to the $k$-th cluster, then $\mu_i = \mu^k$ and $\xi_i \sim \mathcal{N}(0, \Sigma^k)$. Since the data is randomly drawn from each class, $\mu_i$ follows the multinomial distribution over $\mu^1, \cdots, \mu^r$ with probability $p_1, \cdots, p_{r+1}$. Thus $\mu_i$ follows a subgaussian distribution with covariance matrix $N = \sum_{k=1}^{r+1} p_k \mu_k \mu_k^\top$.

As shown in Theorem C.1, the optimal solution of contrastive learning is equivalent to PCA of the following matrix:

$$
\begin{aligned}
\hat{T} \triangleq &\frac{1}{4n}(\Delta(XX^\top) - \frac{1}{n-1}X(1_n 1_n^\top - I_n)X^\top) \\
&+ \frac{1}{r+1}\sum_{k=1}^{r+1}\frac{\alpha_k}{n_k}[\frac{1}{n_k-1}X_k(1_{n_k}1_{n_k}^\top - I_{n_k})X_k^\top \\
&- \frac{1}{\sum_{t\neq k} n_t}\sum_{s\neq k}\frac{1}{2}(X_k 1_k 1_s^\top X_s^\top + X_s 1_s 1_k^\top X_k^\top)].
\end{aligned}
$$

Again we will deal with these terms separately,

1. For the first term, as we have discussed, $X$ can be divided into two matrices $M$ and $E$, each of them consists of subgaussian columns. Again we can obtain the result as in (B.24) (the proof is totally same):

$$
\mathbb{E}\|\frac{1}{n}(\Delta(XX^\top) - \frac{1}{n-1}X(1_n 1_n^\top - I_n)X^\top) - N\|_2 \lesssim \nu^2(\frac{r}{d}\log d + \sqrt{\frac{r}{n}}) + \sigma_{(1)}^2\sqrt{\frac{d}{n}}.
\tag{C.3}
$$

2. For the second term, notice that:

$$
X_k(1_{n_k}1_{n_k}^\top - I_{n_k})X_k^\top = \sum_{i=1}^{n_k}\sum_{j\neq i}(\mu_k + \xi_i^k)(\mu_k + \xi_j^k)^\top
$$

$$
= n_k(n_k - 1)\mu_k\mu_k^\top + (n_k - 1)\mu_k(\sum_{i=1}^{n_k}\xi_i^k)^\top + (n_k - 1)(\sum_{i=1}^{n_k}\xi_i^k)\mu_k^\top + \sum_{i=1}^{n_k}\sum_{j\neq i}\xi_i^k\xi_j^{kT},
\tag{C.4}
$$

and that:

$$
\frac{1}{\sum_{t\neq k} n_t}\sum_{s\neq k}X_k 1_k 1_s^\top X_s^\top = \frac{1}{\sum_{t\neq k} n_t}\sum_{s\neq k}\sum_{i=1}^{n_k}(\mu_k + \xi_i^k)\sum_{j=1}^{n_s}(\mu_s + \xi_j^s)^\top
$$

$$
= \frac{1}{\sum_{t\neq k} n_t}\sum_{s\neq k}[n_k n_s \mu_k\mu_s^\top + n_k\mu_k(\sum_{j=1}^{n_s}\xi_j^s)^\top + n_s\sum_{i=1}^{n_k}\xi_i^k\mu_s^\top + \sum_{i=1}^{n_k}\xi_i^k\sum_{j=1}^{n_s}\xi_j^{sT}].
\tag{C.5}
$$

Since $\xi_i^k \sim \mathcal{N}(0, \Sigma^k)$, we can conclude that:

$$
\mathbb{E}\|\frac{1}{n_k}\sum_{i=1}^{n_k}\xi_i^k\|_2 \leq \sqrt{\mathbb{E}\|\frac{1}{n_k}\sum_{i=1}^{n_k}\xi_i^k\|_2^2} = \sqrt{\frac{d}{n_k}}\sigma_{(1)}.
\tag{C.6}
$$

Moreover, we have

$$
\frac{1}{n_k(n_k-1)}\mathbb{E}\|\sum_{i=1}^{n_k}\sum_{j\neq i}\xi_i^k\xi_j^{kT}\|_2 \leq \frac{1}{n_k(n_k-1)}\mathbb{E}\|E_k E_k^\top\|_2 + \frac{n_k}{n_k-1}\mathbb{E}\|\bar{\xi}^k\bar{\xi}^{k\top}\|_2
\tag{C.7}
$$

$$
\lesssim \frac{d}{n_k}\sigma_{(1)}^2.
$$

Take equation (C.6) and (C.7) back into (C.4) we can conclude:

$$
\mathbb{E}\|\frac{1}{n_k(n_k-1)}X_k(1_{n_k}1_{n_k}^\top - I_{n_k})X_k^\top - \mu_k\mu_k^\top\|_2 \lesssim \sqrt{\frac{d}{n_k}}\sigma_{(1)}\sqrt{r}\nu + \frac{d}{n_k}\sigma_{(1)}^2.
\tag{C.8}
$$

On the other hand, by equation (C.6) we know:

$$\mathbb{E}\|\frac{1}{\sum_{t\neq k} n_t}\sum_{s\neq k}\sum_{j=1}^{n_s}\xi_j^s\|_2 \leq \sum_{s\neq k}\frac{n_s}{\sum_{t\neq k} n_t}\mathbb{E}\|\frac{1}{n_s}\sum_{i=1}^{n_s}\xi_i^s\|_2 \lesssim \sum_{s\neq k}\frac{n_s}{\sum_{t\neq k} n_t}\sqrt{\frac{d}{n_s}}\sigma_{(1)}. \tag{C.9}$$

Notice that:

$$\mathbb{E}\|\frac{1}{\sum_{t\neq k} n_t}\frac{1}{n_k}\sum_{s\neq k}\sum_{i=1}^{n_k}\xi_i^k\sum_{j=1}^{n_s}\xi_j^{sT}\|_2 \leq \mathbb{E}\|\sum_{s\neq k}\frac{n_s}{\sum_{t\neq k} n_t}\bar{\xi}^k\bar{\xi}^{s\top}\|_2$$

$$\leq \sum_{s\neq k}\frac{n_s}{\sum_{t\neq k} n_t}\mathbb{E}\|\bar{\xi}^k\bar{\xi}^{s\top}\|_2 \lesssim \sum_{s\neq k}\frac{n_s}{\sum_{t\neq k} n_t}\frac{d}{\sqrt{n_k n_s}}\sigma_{(1)}^2. \tag{C.10}$$

Thus take equations (C.9) and (C.10) back into equation (C.5) we have:

$$\mathbb{E}\|\frac{1}{n_k}\frac{1}{\sum_{t\neq k} n_t}\sum_{s\neq k}X_k 1_k 1_s^\top X_s^\top - \sum_{s\neq k}\frac{n_s}{\sum_{t\neq k} n_t}\mu_k\mu_s^\top\|_2 \tag{C.11}$$

$$\lesssim \sum_{s\neq k}\frac{\sqrt{n_s d}}{\sum_{t\neq k} n_t}(\sqrt{\frac{d}{n_k}}\sigma_{(1)}^2 + \sigma_{(1)}\sqrt{r}\nu). \tag{C.12}$$

Then combine equations (C.3)(C.8)(C.11) together, we can obtain the following result:

$$\mathbb{E}\|\hat{T} - \frac{1}{4}N - \frac{1}{r+1}\sum_{k=1}^{r+1}\alpha_k(\mu_k\mu_k^\top - \sum_{s\neq k}\frac{n_s}{\sum_{t\neq k} n_t}\frac{1}{2}(\mu_k\mu_s^\top + \mu_s\mu_k^\top))\|_2$$

$$\lesssim \nu^2\left(\frac{r}{d}\log d + \sqrt{\frac{r}{n}}\right) + \sigma_{(1)}^2\sqrt{\frac{d}{n}}$$

$$+ \frac{1}{r+1}\sum_{k=1}^{r+1}\alpha_k\left[\sum_{s\neq k}\frac{\sqrt{n_s d}}{\sum_{t\neq k} n_t}\left(\sqrt{\frac{d}{n_k}}\sigma_{(1)}^2 + \sqrt{r}\sigma_{(1)}\nu\right) + \sqrt{\frac{d}{n_k}}\sigma_{(1)}\sqrt{r}\nu + \frac{d}{n_k}\sigma_{(1)}^2\right].$$

Since we have assumed that $\text{rank}(\sum_{k=1}^{r+1}p_k\mu_k\mu_k^\top) = r$ we can find that the top-r eigenspace of matrix:

$$T = \frac{1}{4}\sum_{k=1}^{r+1}p_i\mu_k\mu_k^\top + \frac{1}{r+1}\sum_{k=1}^{r+1}\alpha_k\left(\mu_k\mu_k^\top - \sum_{s\neq k}\frac{n_s}{\sum_{t\neq k} n_t}\frac{1}{2}(\mu_k\mu_s^\top + \mu_s\mu_k^\top)\right)$$

is spanned by $U^\star$, then apply Lemma D.1 again we have:

$$\mathbb{E}\|\sin\Theta(U_{SCL}, U)\|_F \leq \frac{2\sqrt{r}\mathbb{E}\|\hat{N} - N\|_2}{\lambda_r(N)}$$

$$\lesssim \frac{\sqrt{r}}{\lambda_r(T)}\left[\nu^2\left(\frac{r}{d}\log d + \sqrt{\frac{r}{n}}\right) + \sigma_{(1)}^2\sqrt{\frac{d}{n}}\right.$$

$$\left. + \frac{1}{r+1}\sum_{k=1}^{r+1}\alpha_k\left[\sum_{s\neq k}\frac{\sqrt{n_s d}}{\sum_{t\neq k} n_t}\left(\sqrt{\frac{d}{n_k}}\sigma_{(1)}^2 + \sqrt{r}\sigma_{(1)}\nu\right) + \sqrt{\frac{d}{n_k}}\sqrt{r}\sigma_{(1)}\nu + \frac{d}{n_k}\sigma_{(1)}^2\right]\right]$$

$$\lesssim \frac{\nu^2}{\lambda_r(T)}\left(\frac{r^{3/2}}{d}\log d + \sqrt{\frac{dr}{n}} + \frac{1}{r+1}\sum_{k=1}^{r+1}\alpha_k\left[\sum_{s\neq k}\frac{\sqrt{n_s d}}{\sum_{t\neq k} n_t}\left(\sqrt{\frac{d}{n_k}} + \sqrt{r}\right) + \sqrt{\frac{dr}{n_k}}\right]\right).$$

$$\square$$

Roughly speaking, since $\|\mu_k\| = \sqrt{r}\nu$ and $\sum_{k=1}^{r+1}p_k\mu_k = 0$, approximately we have $\frac{\nu^2}{\lambda_r(N)} \approx \frac{1}{\min_{k\in[r]}[1+\alpha_k]}$. Although we can not obtain the closed-form eigenvalue in general, in a special case,

where $\alpha = \alpha_1 = \cdots = \alpha_{r+1}, m = n_1 = n_2 = \cdots = n_{r+1}$ and $\frac{1}{r+1} = p_1 = p_2 = \cdots = p_{r+1}$, it's easy to find that:

$$\sum_{s \neq k} \frac{1}{2}(\mu_k \mu_s^\top + \mu_s \mu_k^\top) = -\mu_k \mu_k^\top,$$

which further implies that:

$$T = \frac{1}{4}\sum_{k=1}^{r+1} p_k \mu_k \mu_k^\top + \frac{1}{r+1}\sum_{k=1}^{r+1}\alpha(1+\frac{1}{r})\mu_k\mu_k^\top, \quad \lambda_r(T) = [\frac{1}{4} + \alpha(1+\frac{1}{r})]\lambda(N).$$

and we can obtain the result in Theorem 4.1.

## C.2 PROOFS FOR SECTION 4.2

In this section, we will provide the proof of generalized version of Theorem 4.2 and 4.3 to cover the imbalanced setting, the statement and detailed proof can be found in Theorem C.4 and C.5. First we prove a useful lemma to illustrate that supervised loss function only yields estimation along a 1-dimensional space. Consider a single source task, where the data $x = U^\star z + \xi$ is generated by spiked covariance model and the label is generated by

$$y = \langle w^\star, z \rangle$$

suppose we have collect $n$ labeled data from this task, denote the data as $X = [x_1, x_2, \cdots, x_n] \in \mathbb{R}^{d \times n}$ and the label $y = [y_1, y_2, \cdots, y_n] \in \mathbb{R}^n$, then we have the following result.

**Lemma C.1.** *Under the conditions similar to Theorem 3.2, we can find an event $A$ such that $\mathbb{P}(A^C) = O(\sqrt{d/n})$ and:*

$$\mathbb{E}\left[\left\|\frac{1}{(n-1)^2}XHyy^\top HX^\top - \nu^2 U^\star w^\star w^{\star\top} U^{\star\top}\right\|_F \mathbb{I}\{A\}\right] \lesssim \sqrt{\frac{d}{n}}\sigma_{(1)}\nu. \tag{C.13}$$

The proof strategy is to estimate the difference between the two rank-1 matrices via bounding the difference of the corresponding the vector component. We first provide a simple lemma to illustrate the technique:

**Lemma C.2.** *Suppose $\alpha, \beta \in \mathbb{R}^d$ are two vectors, then we have:*

$$\|\alpha\alpha^\top - \beta\beta^\top\|_F \leq \sqrt{2}(\|\alpha\|_2 + \|\beta\|_2)\|\alpha - \beta\|_2.$$

*Proof.* Denote $\alpha = (\alpha_1, \cdots, \alpha_d), \beta = (\beta_1, \cdots, \beta_d)$, then we have:

$$\|\alpha\alpha^\top - \beta\beta^\top\|_F^2 \leq \sum_{i=1}^d \sum_{j=1}^d |\alpha_i\alpha_j - \beta_i\beta_j|^2 \leq 2\sum_{i=1}^d \sum_{j=1}^d |\alpha_i\alpha_j - \alpha_i\beta_j|^2 + |\alpha_i\beta_j - \beta_i\beta_j|^2$$

$$\leq 2\sum_{i=1}^d \sum_{j=1}^d |\alpha_i|^2|\alpha_j - \beta_j|^2 + |\beta_j|^2|\alpha_i - \beta_i|^2 \leq 2(\|\alpha\|_2^2 + \|\beta\|_2^2)\|\alpha - \beta\|_2^2$$

$$\leq 2(\|\alpha\|_2 + \|\beta\|_2)^2\|\alpha - \beta\|_2^2.$$

Take square root on both side we can finish the proof. $\qquad\square$

Now we can prove the Lemma C.1.

*Proof of Lemma C.1.* Clearly, we have:

$$\left\|\frac{1}{(n-1)^2}XHyy^\top HX^\top - \nu^2 U^\star w^\star w^{\star\top} U^{\star\top}\right\|_F$$

$$\leq \frac{n^2}{(n-1)^2}\left\|\frac{1}{n^2}XHyy^\top HX^\top - \nu^2 U^\star w^\star w^{\star\top} U^{\star\top}\right\|_F + \frac{2n+1}{(n-1)^2}\|\nu^2 U^\star w^\star w^{\star\top} U^{\star\top}\|_F$$

$$\lesssim \left\|\frac{1}{n^2}XHyy^\top HX^\top - \nu^2 U^\star w^\star w^{\star\top} U^{\star\top}\right\|_F + \frac{r}{n}\nu^2,$$

thus we can replace the $\frac{1}{(n-1)^2}$ with $\frac{1}{n}$ in equation (C.13) and conclude the proof. Denote $\hat{N} \triangleq \frac{1}{n^2} XHyy^\top HX^\top$, note that both of $\hat{N}$ and $Uw^\star w^{\star\top} U^\top$ are rank-1 matrices. We first bound the difference between $\frac{1}{n}XHy$ and $Uw^\star$:

$$
\begin{aligned}
\|\frac{1}{n}XHy - \nu U^\star w^\star\| = & \|\frac{1}{n\nu}(U^\star Z + E)HZ^\top w^\star - \nu U^\star w^\star\| \\
\leq & \|\frac{1}{n\nu}(U^\star Z + E)HZ^\top - \nu U^\star\|_2 \\
\leq & \frac{1}{\nu}(\|\frac{1}{n}U^\star ZZ^\top - \nu^2 U^\star\|_2 + \frac{1}{n}\|EZ^\top\|_2 + \frac{1}{n}\|U^\star Z\bar{Z}^\top\|_2 + \frac{1}{n}\|E\bar{Z}^\top\|_2).
\end{aligned}
\tag{C.14}
$$

We deal with the four terms in (C.14) separately:

1. For the first term, apply Lemma D.3 we have:

$$
\mathbb{E}\|\frac{1}{n}U^\star ZZ^\top - \nu^2 U^\star\|_2 \leq \mathbb{E}\|\frac{1}{n}ZZ^\top - \nu^2 I_r\|_2 \leq \left(\frac{r}{n} + \sqrt{\frac{r}{n}}\right)\nu^2.
\tag{C.15}
$$

2. For the second term, apply Lemma D.2 twice we have:

$$
\begin{aligned}
\frac{1}{n}\mathbb{E}\|EZ^\top\|_2 = & \frac{1}{n}\mathbb{E}_Z[\mathbb{E}_E[\|EZ^\top\|_2|Z]] \\
\lesssim & \frac{1}{n}\mathbb{E}_Z[\|Z\|_2(\sigma_{\text{sum}} + r^{1/4}\sqrt{\sigma_{\text{sum}}\sigma_{(1)}} + \sqrt{r}\sigma_{(1)})] \\
\lesssim & \frac{1}{n}\mathbb{E}_Z[\|Z\|_2]\sqrt{d}\sigma_{(1)} \\
\lesssim & \frac{1}{n}\sqrt{d}\sigma_{(1)}(r^{1/2}\nu + (nr)^{1/4}\nu + n^{1/2}\nu) \\
\lesssim & \frac{\sqrt{d}}{\sqrt{n}}\sigma_{(1)}\nu.
\end{aligned}
\tag{C.16}
$$

3. For the third term and fourth term, from equation (4) we know:

$$
\mathbb{E}\frac{1}{n}\|U^\star Z\bar{Z}^\top\|_2 + \mathbb{E}\frac{1}{n}\|E\bar{Z}^\top\|_2 \leq \mathbb{E}\|\bar{z}\bar{z}^\top\|_2 + \mathbb{E}\|\bar{\xi}\bar{z}^\top\|_2 \leq \frac{r}{n}\nu^2 + \sqrt{\frac{d}{n}}\nu\sigma_{(1)}.
\tag{C.17}
$$

Combine these three equations (C.15)(C.16)(C.17) together we have:

$$
\mathbb{E}\|\frac{1}{n}XHy - \nu U^\star w^\star\| \lesssim \sqrt{\frac{d}{n}}\sigma_{(1)}.
\tag{C.18}
$$

With equation (C.18), we can now turn to the difference between $\hat{N}$ and $Uw^\star w^{\star\top} U^\top$. By Lemma C.2 we know that:

$$
\|\hat{N} - \nu^2 U^\star w^\star w^{\star\top} U^{\star\top}\|_F \lesssim (\|\frac{1}{n}XHy\| + \|\nu U^\star w^\star\|)\|\frac{1}{n}XHy - \nu U^\star w^\star\|.
$$

Using Markov's inequality, we can conclude from (C.18) that:

$$
\mathbb{P}(\|\frac{1}{n}XHy - \nu U^\star w^\star\| \geq \nu) \leq \frac{\mathbb{E}\|\frac{1}{n}XHy - \nu U^\star w^\star\|}{\nu} \lesssim \sqrt{\frac{d}{n}}.
$$

Then denote $A = \{\omega : \|\frac{1}{n}XHy - \nu^2 U^\star w^\star\|_2 < \nu\}$ we have:

$$
\begin{aligned}
\mathbb{E}\|\hat{N} - \nu^2 U^\star w^\star w^{\star\top} U^{\star\top}\|_F \mathbb{I}\{A\} \lesssim & \mathbb{E}(\|\frac{1}{n}XHy\| + \|\nu U^\star w^\star\|)\|\frac{1}{n}XHy - \nu U^\star w^\star\|\|\mathbb{I}\{A\} \\
\lesssim & \nu\mathbb{E}\|\frac{1}{n}XHy - \nu U^\star w^\star\|_2 \lesssim \sqrt{\frac{d}{n}}\sigma_{(1)}\nu.
\end{aligned}
$$

which finished the proof. $\qquad\square$

In the main body, we assume the number of labeled data and the ratio of loss function is both balanced. Now we will provide a more general result to cover the imbalance occasions. Formally, suppose we have $n$ unlabeled data $X = [x_1, \cdots, x_n] \in \mathbb{R}^{d \times n}$ and $n_i$ labeled data $\mathcal{S}_i$ $X_i = [x_i^1, \cdots, x_i^{n_i}], y_i = [y_i^1, \cdots, y_i^{n_1}], \forall i = 1, \cdots T$ for source task , we learn the linear representation via joint optimization:

$$\min_{W \in \mathbb{R}^{r \times d}} \mathcal{L}(W) := \min_{W \in \mathbb{R}^{r \times d}} \mathcal{L}_{\text{SelfCon}}(W) - \sum_{t=1}^{T} \alpha_i \, \text{HSIC}(\hat{X}^t, y^t; W), \tag{C.19}$$

To investigate its feature recovery ability, we first give the following result.

**Theorem C.3.** *For optimization problem C.19, if we apply augmented pairs generation 2.1 with random masking augmentation 2.2 for unlabeled data, then the optimal solution is given by:*

$$W_{CL} = C \left( \sum_{i=1}^{r} u_i \sigma_i v_i^\top \right)^\top,$$

*where $C > 0$ is a constant, $\sigma_i$ is the $i$-largest eigenvalue of the following matrix:*

$$\frac{1}{4n} \left( \Delta(XX^\top) - \frac{1}{n-1} X(1_n 1_n^\top - I_n) X^\top \right) + \sum_{i=1}^{T} \frac{\alpha_i}{(n_i - 1)^2} X_i H_{n_i} y_i y_i^\top H_{n_i} X_i^\top),$$

*$u_i$ is the corresponding eigenvector, $V = [v_1, \cdots, v_r] \in \mathbb{R}^{r \times r}$ can be any orthogonal matrix and $H_{n_i} = I_{n_i} - \frac{1}{n_i} 1_{n_i} 1_{n_i}^\top$ is the centering matrix.*

*Proof.* Under this setting, combine with the result obtained in B.1, the loss function can be rewritten as:

$$\mathcal{L}(W) = \frac{\lambda}{2} \|WW^\top\|_F^2 - \frac{1}{2n} \text{tr} \left( \left( \frac{1}{2} \Delta(XX^\top) - \frac{1}{2(n-1)} X(1_n 1_n^\top - I_n) X^\top \right) W^\top W \right)$$

$$- \sum_{t=1}^{T} \alpha_i \frac{1}{(n_i - 1)^2} \text{tr} \left( X_i^\top W^\top W X_i H y_i y_i^\top H \right)$$

$$= \frac{\lambda}{2} \left\| WW^\top - \frac{1}{4n\lambda} \left( \Delta(XX^\top) - \frac{1}{n-1} X(1_n 1_n^\top - I_n) X^\top \right) \right.$$

$$\left. - \sum_{i=1}^{T} \frac{\alpha_i}{\lambda(n_i - 1)^2} X_i H_{n_i} y_i y_i^\top H_{n_i} X_i^\top) \right\|_F^2$$

$$- \frac{\lambda}{2} \left\| \frac{1}{4n\lambda} \left( \Delta(XX^\top) - \frac{1}{n-1} X(1_n 1_n^\top - I_n) X^\top \right) \right.$$

$$\left. + \sum_{i=1}^{T} \frac{\alpha_i}{\lambda(n_i - 1)^2} X_i H_{n_i} y_i y_i^\top H_{n_i} X_i^\top \right\|_F^2.$$

Then by similar argument as in the proof of Theorem B.2, we can conclude that the optimal solution $W_{CL}$ must satisfy the desired conditions. $\square$

Then we can give the proofs of Theorem 4.2 and Theorem 4.3 under our generalized setting, one can easily obtain those under balanced setting by simply setting $\alpha = \alpha_1 = \cdots = \alpha_T$ and $m = n_1 = \cdots = n_T$, which is consistent with Theorem 4.2 and Theorem 4.3 in the mainbody.

**Theorem C.4** (Generalized version of Theorem 4.2)**.** *Suppose Assumption 3.1-3.3 hold for spiked covariance model Eq.(2.5) and $n > d \gg r$, if we further assume that $T < r$ and $w_t$'s are orthogonal to each other, and let $W^{CL}$ be any solution that optimizes the problem in Eq.(C.19), and denote its singular value decomposition as $W_{CL} = (U_{CL} \Sigma_{CL} V_{CL}^\top)^\top$, then we have:*

$$\mathbb{E} \| \sin(\Theta(U_{CL}, U^\star)) \|_F \lesssim \left( \frac{\sqrt{r-T}}{\min_{i \in [T]} \{\alpha_i, 1\}} + \frac{\sqrt{T}}{\min_{i \in [T]} \alpha_i} \right) \left( \frac{r}{d} \log d + \sqrt{\frac{d}{n}} \right)$$

$$+ \sum_{i=1}^{T} \left( \sqrt{r-T} \frac{\alpha_i + \min_{i \in [T]} \{\alpha_i, 1\}}{\min_{i \in [T]} \{\alpha_i, 1\}} + \sqrt{T} \frac{\alpha_i + \min_{i \in [T]} \alpha_i}{\min_{i \in [T]} \alpha_i} \right) \sqrt{\frac{d}{n_i}}.$$

*Proof.* As shown in Theorem C.3, optimizing loss function (C.19) is equivalent to find the top-$r$ eigenspace of matrix

$$\frac{1}{4n}\left(\Delta(XX^\top) - \frac{1}{n-1}X(1_n 1_n^\top - I_n)X^\top\right) + \sum_{i=1}^T \frac{\alpha_i}{(n_i-1)^2}X_i H_{n_i}y_i y_i^\top H_{n_i}X_i^\top.$$

Again denote $\hat{M}_2 \triangleq \frac{1}{n}(\Delta(XX^\top) - \frac{1}{n-1}X(1_n 1_n^\top - I_n)X^\top)$ and $\hat{N}_i \triangleq \frac{1}{(n_i-1)^2}X_i H y_i y_i^\top H X_i^\top$. By equation (B.24) we know that:

$$\mathbb{E}\|\hat{M}_2 - M\|_2 \lesssim \nu^2\left(\frac{r}{d}\log d + \sqrt{\frac{r}{n}} + \frac{r}{n}\right) + \sigma_{(1)}^2\left(\sqrt{\frac{d}{n}} + \frac{d}{n}\right) + \sigma_{(1)}\nu\sqrt{\frac{d}{n}}.$$

By Theorem C.1 we know that for each task $\mathcal{S}_i$, we can find an event $A_i$ such that $\mathbb{P}(A_i) = O(\sqrt{\frac{d}{n}})$:

$$\mathbb{E}\|\hat{N}_i - \nu^2 U^\star w_i w_i^\top U^{\star\top}\|_F \mathbb{I}\{A_i\} \lesssim \sqrt{\frac{d}{n_i}}\sigma_{(1)}\nu.$$

The target matrix is $N = \nu^2 U^\star U^{\star\top} + \sum_{i=1}^T \alpha_i \nu^2 U^\star w_i w_i^T U^{\star\top}$, and we can obtain the upper bound for the difference between $N$ and $\hat{N}$:

$$\mathbb{E}\|\hat{N} - N\|_2 \mathbb{I}\{\cap_{i=1}^T A_i\} \leq \frac{1}{4}\mathbb{E}\|\hat{M}_2 - M\|_2 + \sum_{i=1}^T \alpha_i \mathbb{E}\|\hat{N}_i - \nu^2 U w_i w_i^\top U^\top\|_F \mathbb{I}\{A_i\}$$

$$\lesssim \nu^2(\frac{r}{d}\log d + \sqrt{\frac{r}{n}} + \frac{r}{n}) + \sigma_{(1)}^2\left(\sqrt{\frac{d}{n}} + \frac{d}{n}\right) + \sigma_{(1)}\nu\sqrt{\frac{d}{n}} + \sum_{i=1}^T \left[\alpha_i\sqrt{\frac{d}{n_i}}\sigma_{(1)}\nu^2\right]. \tag{C.20}$$

We divide the top-r eigenspace $U_{CL}$ of $W_{CL}W_{CL}^\top$ into two parts: the top-$T$ eigenspace $U_{CL}^{(1)}$ and top-$(T+1)$ to top-$r$ eigenspace $U_{CL}^{(2)}$. Similarly, we also divide the top-$r$ eigenspace $U^\star$ of $N$ into two parts: $U^{\star(1)}$ and $U^{\star(2)}$. Then apply Lemma D.1 we have we can bound the $\sin\Theta$ distance for each parts: on the one hand,

$$\mathbb{E}\|\sin\left(\Theta(U_{CL}^{(1)}, U^{\star(1)})\right)\|_F$$

$$= \mathbb{E}\|\sin\left(\Theta(U_{CL}^{(1)}, U^{\star(1)})\right)\|_F \mathbb{I}\{\cap_{i=1}^T A_i\} + \mathbb{E}\|\sin\left(\Theta(U_{CL}^{(1)}, U^{\star(1)})\right)\|_F \mathbb{I}\{\cup_{i=1}^T A_i^C\}$$

$$\leq \frac{\sqrt{T}\mathbb{E}\|\hat{N} - N\|_2 \mathbb{I}\{\cap_{i=1}^T A_i\}}{\lambda_{(T)}(N) - \lambda_{(T+1)}(N)} + \sqrt{T}\mathbb{P}(\cup_{i=1}^T A_i^C)$$

$$\lesssim \frac{\sqrt{T}}{\min_{i\in[T]}\alpha_i \nu^2}\left(\nu^2\frac{r}{d}\log d + \sigma_{(1)}^2\sqrt{\frac{d}{n}} + \sum_{i=1}^T \alpha_i\sqrt{\frac{d}{n_i}}\sigma_{(1)}\nu\right) + \sqrt{T}\sum_{i=1}^T \sqrt{\frac{d}{n_i}}$$

$$\lesssim \frac{\sqrt{T}}{\min_{i\in[T]}\alpha_i}\left(\frac{r}{d}\log d + \sqrt{\frac{d}{n}}\right) + \sqrt{T}\sum_{i=1}^T \frac{\alpha_i + \min_{i\in[T]}\alpha_i}{\min_{i\in[T]}\alpha_i}\sqrt{\frac{d}{n_i}}.$$

On the other hand,

$$\mathbb{E}\|\sin\left(\Theta(U_{CL}^{(2)}, U^{\star(2)})\right)\|_F$$

$$= \mathbb{E}\|\sin\left(\Theta(U_{CL}^{(2)}, U^{\star(2)})\right)\|_F \mathbb{I}\{\cap_{i=1}^T A_i\} + \mathbb{E}\|\sin\left(\Theta(U_{CL}^{(2)}, U^{\star(2)})\right)\|_F \mathbb{I}\{\cup_{i=1}^T A_i^C\}$$

$$\leq \frac{\sqrt{r-T}\mathbb{E}\|\hat{N} - N\|_2 \mathbb{I}\{\cap_{i=1}^T A_i\}}{\min\{\lambda_{(T)}(N) - \lambda_{(T+1)}(N), \lambda_{(r)}(N)\}} + \sqrt{r-T}\mathbb{P}(\cup_{i=1}^T A_i^C)$$

$$\lesssim \frac{\sqrt{r-T}}{\min_{i\in[T]}\{\alpha_i, 1\}\nu^2}\left(\nu^2\frac{r}{d}\log d + \sigma_{(1)}^2\sqrt{\frac{d}{n}} + \sum_{i=1}^T \alpha_i\sqrt{\frac{d}{n_i}}\sigma_{(1)}\nu\right) + \sqrt{r-T}\sum_{i=1}^T \sqrt{\frac{d}{n_i}}$$

$$\lesssim \frac{\sqrt{r-T}}{\min_{i\in[T]}\{\alpha_i, 1\}}\left(\frac{r}{d}\log d + \sqrt{\frac{d}{n}}\right) + \sqrt{r-T}\sum_{i=1}^T \left(\frac{\alpha_i}{\min_{i\in[T]}\{\alpha_i, 1\}} + 1\right)\sqrt{\frac{d}{n_i}}.$$

Note that:

$$\| \sin(\Theta(U_{CL}, U^\star)) \|_F^2$$
$$= r - \| U_{CL}^\top U^\star \|_F^2$$
$$\le r - \| U_{CL}^{(1)\top} U^{\star(1)} \|_F^2 - \| U_{CL}^{(2)T} U^{\star(2)} \|_F^2$$
$$\le T - \| U_{CL}^{(1)\top} U^{\star(1)} \|_F^2 + (r - T) - \| U_{CL}^{(2)\top} U^{\star(2)} \|_F^2$$
$$\le \| \sin\left(\Theta(U_{CL}^{(1)}, U^{\star(1)})\right) \|_F^2 + \| \sin\left(\Theta(U_{CL}^{(1)}, U^{\star(1)})\right) \|_F^2,$$

and the $\sin \Theta$ distance has trivial upper bounds:

$$\| \sin\left(\Theta(U_{CL}^{(1)}, U^{\star(1)})\right) \|_F^2 \le T, \quad \| \sin\left(\Theta(U_{CL}^{(2)}, U^{\star(2)})\right) \|_F^2 \le r - T$$

Thus we can conclude:

$$\mathbb{E}\| \sin(\Theta(U_{CL}, U^\star)) \|_F$$
$$\le \mathbb{E}\| \sin\left(\Theta(U_{CL}^{(1)}, U^{\star(1)})\right) \|_F + \mathbb{E}\| \sin\left(\Theta(U_{CL}^{(2)}, U^{\star(2)})\right) \|_F$$
$$\lesssim \left( \frac{\sqrt{r-T}}{\min_{i\in[T]}\{\alpha_i, 1\}} \left( \frac{r}{d}\log d + \sqrt{\frac{d}{n}} \right) + \sum_{i=1}^T \sqrt{r-T}\frac{\alpha_i + \min_{i\in[T]}\{\alpha_i, 1\}}{\min_{i\in[T]}\{\alpha_i, 1\}} \sqrt{\frac{d}{n_i}} \right) \wedge \sqrt{r-T}$$
$$+ \left( \frac{\sqrt{T}}{\min_{i\in[T]}\alpha_i} \left( \frac{r}{d}\log d + \sqrt{\frac{d}{n}} \right) + \sum_{i=1}^T \sqrt{T}\frac{\alpha_i + \min_{i\in[T]}\alpha_i}{\min_{i\in[T]}\alpha_i} \sqrt{\frac{d}{n_i}} \right) \wedge \sqrt{T}.$$

$\square$

**Theorem C.5** (Generalized version of Theorem 4.3). *Suppose Assumptions 3.1-3.3 hold for spiked covariance model Eq.(2.5) and $n > d \gg r$, if we further assume that $T \ge r$ and $\sum_{i=1}^T \alpha_i w_i w_i^\top$ is full rank, suppose $W^{CL}$ is the optimal solution of optimization problem eq.(C.19), and denote its singular value decomposition as $W_{CL} = (U_{CL}\Sigma_{CL}V_{CL}^\top)^\top$, then we have:*

$$\mathbb{E}\| \sin(\Theta(U_{CL}, U)) \|_F \lesssim \frac{\sqrt{r}}{1 + \nu^2\lambda_{(r)}(\sum_{i=1}^T \alpha_i w_i w_i^\top)} \left( \frac{r}{d}\log d + \sqrt{\frac{d}{n}} \right)$$
$$+ \sqrt{r}\sum_{i=1}^T \left( \frac{\alpha_i}{1 + \nu^2\lambda_{(r)}(\sum_{i=1}^T \alpha_i w_i w_i^\top)} + 1 \right)\sqrt{\frac{d}{n_i}}.$$

*Proof.* The proof strategy is similar to that of Theorem 4.2, here the difference is that each direction can be accurately estimated by the labeled data and we don't need to separate the eigenspace. Directly applying Lemma D.1 and equation (C.20) we have:

$$\mathbb{E}\| \sin(\Theta(U_{CL}, U^\star)) \|_F$$
$$= \mathbb{E}\| \sin(\Theta(U_{CL}, U^\star)) \|_F \mathbb{I}\{\cap_{i=1}^T A_i\} + \mathbb{E}\| \sin(\Theta(U_{CL}, U^\star)) \|_F \mathbb{I}\{\cup_{i=1}^T A_i^C\}$$
$$\lesssim \frac{\sqrt{r}\mathbb{E}\|\hat{N} - N\|_2 \mathbb{I}\{\cap_{i=1}^T A_i\}}{\lambda_{(r)}(N)} + \sqrt{r}\mathbb{P}(\cup_{i=1}^T A_i^C)$$
$$\lesssim \frac{\sqrt{r}}{\nu^2 + \nu^2\lambda_{(r)}(\sum_{i=1}^T \alpha_i w_i w_i^\top)} \left( \nu^2\frac{r}{d}\log d + \sigma_{(1)}^2\sqrt{\frac{d}{n}} + \sum_{i=1}^T \alpha_i\sqrt{\frac{d}{n_i}}\sigma_{(1)}\nu \right) + \sqrt{r}\sum_{i=1}^T \sqrt{\frac{d}{n_i}}$$
$$\lesssim \frac{\sqrt{r}}{1 + \lambda_{(r)}(\sum_{i=1}^T \alpha_i w_i w_i^\top)} \left( \frac{r}{d}\log d + \sqrt{\frac{d}{n}} \right) + \sqrt{r}\sum_{i=1}^T \left( \frac{\alpha_i}{1 + \lambda_{(r)}(\sum_{i=1}^T \alpha_i w_i w_i^\top)} + 1 \right)\sqrt{\frac{d}{n_i}}.$$

$\square$

# D USEFUL LEMMAS

In this section, we list some of the main techniques that have been used in the proof of the main results.

**Lemma D.1** (Theorem 2 in Yu et al. (2015)). *Let* $\Sigma, \hat{\Sigma} \in \mathbb{R}^{p \times p}$ *be symmetric, with eigenvalues* $\lambda_1 \geq \ldots \geq \lambda_p$ *and* $\hat{\lambda}_1 \geq \ldots \geq \hat{\lambda}_p$ *respectively. Fix* $1 \leq r \leq s \leq p$ *and assume that* $\min(\lambda_{r-1} - \lambda_r, \lambda_s - \lambda_{s+1}) > 0$ *where* $\lambda_0 := \infty$ *and* $\lambda_{p+1} := -\infty$. *Let* $d := s - r + 1$, *and let* $V = (v_r, v_{r+1}, \ldots, v_s) \in \mathbb{R}^{p \times d}$ *and* $\hat{V} = (\hat{v}_r, \hat{v}_{r+1}, \ldots, \hat{v}_s) \in \mathbb{R}^{p \times d}$ *have orthonormal columns satisfying* $\Sigma v_j = \lambda_j v_j$ *and* $\hat{\Sigma}\hat{v}_j = \hat{\lambda}_j \hat{v}_j$ *for* $j = r, r+1, \ldots, s$. *Then*

$$\|\sin\Theta(\hat{V}, V)\|_F \leq \frac{2\min\left(d^{1/2}\|\hat{\Sigma} - \Sigma\|_2, \|\hat{\Sigma} - \Sigma\|_F\right)}{\min(\lambda_{r-1} - \lambda_r, \lambda_s - \lambda_{s+1})}.$$

*Moreover, there exists an orthogonal matrix* $\hat{O} \in \mathbb{R}^{d \times d}$ *such that*

$$\|\hat{V}\hat{O} - V\|_F \leq \frac{2^{3/2}\min\left(d^{1/2}\|\hat{\Sigma} - \Sigma\|_2, \|\hat{\Sigma} - \Sigma\|_F\right)}{\min(\lambda_{r-1} - \lambda_r, \lambda_s - \lambda_{s+1})}.$$

**Lemma D.2** (Lemma 2 in Zhang et al. (2018)). *Assume that* $E \in \mathbb{R}^{p_1 \times p_2}$ *has independent sub-Gaussian entries,* $\text{Var}(E_{ij}) = \sigma_{ij}^2, \sigma_C^2 = \max_j \sum_i \sigma_{ij}^2, \sigma_R^2 = \max_i \sum_j \sigma_{ij}^2, \sigma_{(1)}^2 = \max_{i,j} \sigma_{ij}^2$. *Assume that*

$$\|E_{ij}/\sigma_{ij}\|_{\psi_2} := \max_{q \geq 1} q^{-1/2}\left\{\mathbb{E}\left(|E_{ij}|/\sigma_{ij}\right)^q\right\}^{1/q} \leq \kappa.$$

*Let* $V \in \mathbb{O}_{p_2, r}$ *be a fixed orthogonal matrix. Then*

$$\mathbb{P}\left(\|EV\|_2 \geq 2(\sigma_C + x)\right) \leq 2\exp\left(5r - \min\left\{\frac{x^4}{\kappa^4\sigma_{(1)}^2\sigma_C^2}, \frac{x^2}{\kappa^2\sigma_{(1)}^2}\right\}\right),$$

$$\mathbb{E}\|EV\|_2 \lesssim \sigma_C + \kappa r^{1/4}\left(\sigma_{(1)}\sigma_C\right)^{1/2} + \kappa r^{1/2}\sigma_{(1)}.$$

**Lemma D.3** (Theorem 6 in Cai et al. (2020)). *Suppose* $Z$ *is a* $p_1$-by-$p_2$ *random matrix with independent mean-zero sub-Gaussian entries. If there exist* $\sigma_1, \ldots, \sigma_p \geq 0$ *such that* $\|Z_{ij}/\sigma_i\|_{\psi_2} \leq C_K$ *for constant* $C_K > 0$, *then*

$$\mathbb{E}\left\|ZZ^\top - \mathbb{E}ZZ^\top\right\|_2 \lesssim \sum_i \sigma_i^2 + \sqrt{p_2 \sum_i \sigma_i^2} \cdot \max_i \sigma_i.$$

**Lemma D.4** (The Eckart-Young-Mirsky Theorem (Eckart & Young, 1936)). *Suppose that* $A = U\Sigma V^T$ *is the singular value decomposition of* $A$. *Then the best rank-$k$ approximation of the matrix* $A$ *w.r.t the Frobenius norm,* $\|\cdot\|_F$, *is given by*

$$A_k = \sum_{i=1}^k \sigma_i u_i v_i^T.$$

*that is, for any matrix* $B$ *of rank at most* $k$

$$\|A - A_k\|_F \leq \|A - B\|_F.$$

# E NUMERICAL EXPERIMENTS

## E.1 SUPPORTING EMPIRICAL RESULTS IN RELATED WORKS

In this section, we list some recent empirical evidence that provides sound support to our theory:

**Contrastive learning outperforms generative self-supervised learning.** In Figure 1 of Chen et al. (2020a) and Figure 5 of Liu et al. (2021), it's observed that contrastive learning has superior performance compared to the generative approach. These results are consistent with our theory in Theorem 3.1-3.4, where we show that contrastive learning achieves better performance on both feature recovery and downstream tasks compared with autoencoder, a representative generative self-supervised learning method.

**Supervised contrastive learning improves downstream accuracy.** In Table 2 of Khosla et al. (2020) and the first column in Table 4 of Islam et al. (2021), supervised contrastive learning shows significant improvement with 7%-8% accuracy increase on ImageNet and Mini-ImageNet. This observation is consistent with our finding in Theorem 4.1, where we prove that supervised contrastive learning can achieve a better upper bound in feature recovery compared with self-supervised contrastive learning.

**Label information may hurt transferability in contrastive learning.** In Table 4 of Khosla et al. (2020) and Table 4 of Islam et al. (2021), where the supervised contrastive learning hardly increases the predictive accuracy compared to the self-supervised contrastive learning (the difference of mean accuracy is less than 1%) and can harm significantly on some datasets (e.g. 5.5% lower for SUN 397 in Table 3 of Khosla et al. (2020)). These results indicate that some mechanisms in supervised contrastive learning hurt model transferability since the improvement on source tasks is significant. Moreover, in Table 4 of Islam et al. (2021), it is observed that combining supervised learning and self-supervised contrastive learning together achieves the best transfer learning performance compared with each of them individually. These findings are consistent with our theory in Theorem 4.2, where we show that the error can increase as $\alpha$ (the ratio between supervised learning loss and self-supervised learning loss) grows and the optimal error is achieved when choosing a moderate $\alpha$, i.e., combining self-supervised learning and supervised learning together.

### E.2 SIMULATION WITH SYNTHETIC DATA

To verify our theory, we conducted numerical experiments on the spiked covariance model (2.5) under a linear representation setting. As we have explicitly formulated the loss function and derive its equivalent form in the main body and appendix, we simply minimize the corresponding loss by gradient descent to find the optimal linear representation $W$. For self-supervised contrastive learning with random masking augmentation, we independently draw the augmentation function by Definition 2.2 and apply them to all of the samples in each iteration. To ensure the convergence, we set the maximum number of iteration for it (typically 10000 or 50000 depends on dimension $d$).

We report two criteria to evaluate the quality of the representation, downstream error and sine distance. To obtain the sine distance for a learned representation $W$, we perform singular value decomposition to get $W = (U\Sigma V^\top)^\top$ and then compute $\|\sin\Theta(U, U^\star)\|_F$. To obtain the downstream task performance, in the comparison between autoencoder and contrastive learning, we first draw $n$ labeled data from spiked covariance model (2.5) with labels generated as in Section 3.2, then we train the model by using the data without labels to obtain the linear representation $W$, and learn a linear predictor $w$ using the data with labels and compute the regression error. In the transfer learning setting, we draw some labeled data on the source tasks and additional unlabeled data. The number of labeled data is set $m = 1000$ and the number of unlabeled data is set $n = 1000$. Then train with them to obtain the linear representation $W$, and draw labeled data from a new source task to learn a linear predictor $w$ to compute the regression error. In particular, we subtract the optimal regression error obtained by the best representation $U^{\star\top}$ for each regression error and report the difference, or more precisely, the excess risk as downstream performance.

The results are reported in Fig. 1, 2 and Table 1, 2. As predicted by Theorem 3.1 and 3.2, the feature recovery error and downstream task risk of contrastive learning decreases as $d$ increases (Fig. 1: **Left**) and as $n$ increases (Fig. 1: **Center**) while that of autoencoder is insensible to the changes in $d$ and $n$. The performance of transfer learning exhibits a *U-shape* type curve when the number of tasks is insufficient, which implies that the supervised training may hurt the transferability and we need to choose an appropriate ratio $\alpha$ to obtain the best performance. When tasks are abundant enough, the performance of transfer learning becomes better as we increase the weight of supervised loss.

| $\log_e(\alpha)$ | -5 | -4 | -3 | -2 | -1 | 0 | 1 | 2 | 3 | 4 | 5 |
|---|---|---|---|---|---|---|---|---|---|---|---|
| $T = 8, r = 10$ | 0.0242 | 0.0231 | 0.0199 | 0.0141 | 0.0122 | 0.0125 | 0.0184 | 0.0345 | 0.0499 | 0.0535 | 0.0587 |
| $T = 20, r = 10$ | 0.0223 | 0.0163 | 0.0156 | 0.0096 | 0.0079 | 0.0055 | 0.0064 | 0.0064 | 0.0067 | 0.0070 | 0.0079 |

Table 1: Downstream performance in transfer learning against the penalty parameter $\alpha$. $T$ is the number of source tasks.

| $\log_e(\alpha)$ | -5 | -4 | -3 | -2 | -1 | 0 | 1 | 2 | 3 | 4 | 5 |
|---|---|---|---|---|---|---|---|---|---|---|---|
| $T = 8, r = 10$ | 2.0373 | 2.0371 | 2.0228 | 1.9908 | 2.0021 | 2.0055 | 2.0010 | 2.0362 | 2.0699 | 2.0705 | 2.0813 |
| $T = 20, r = 10$ | 2.0352 | 2.0292 | 2.0030 | 1.9871 | 1.9740 | 1.9690 | 1.9766 | 1.9702 | 1.9790 | 1.9714 | 1.9672 |

Table 2: Feature recovery performance in transfer learning against the penalty parameter $\alpha$. $T$ is the number of source tasks.

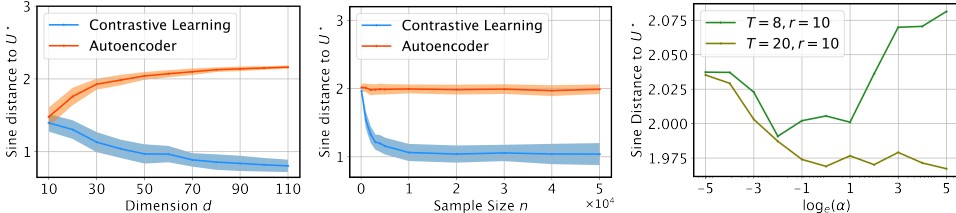

Figure 2: **Left:** Comparison of learned feature between contrastive learning and autoencoders against the dimension $d$. The sample size $n$ is set as $n = 20000$. **Center:** Comparison of feature recovery performance between contrastive learning and autoencoders against the dimension $n$. The dimension $d$ is set as $d = 40$. **Right:** Feature recovery performance in transfer learning against penalty parameter $\alpha$ in log scale. $T$ is the number of source tasks. We set the number of labeled data and unlabeled data as $m = 1000$ and $n = 1000$ respectively.

