# OpenReview forum: "The Power of Contrast for Feature Learning: A Theoretical Analysis"
_ICLR.cc/2022/Conference — ICLR 2022 Submitted_

### Official Review · Reviewer_R3zT · 2021-11-02

**Correctness:** 3
**Technical Novelty And Significance:** 3
**Empirical Novelty And Significance:** 3
**Recommendation:** 5
**Confidence:** 4

**Main Review:**

Overall, I appreciate the plentiful theoretical analysis in this paper. However, some concerns are preventing its acceptance.

1). Theorem 3.1 and Theorem 3.2 provide the sin-distance error bounds, which measure the divergence between the optimal eigenvector U* and the solution U.  And it is good to see that the upper bound of CL is lower than the lower bound of AE (with the increase of d and n). However, I do not think such an inequality relation can really interpret the good generalization performance of CL. Firstly, this error bound is just based on training data which cannot be generalized to the unseen test data. Secondly, the recovered variable is merely U, but the optimization variable W contains U, S, and V.

2). The generalization error bound in Theorem 3.3 is similar to existing work [R1, R2]. The authors can further elaborate on the difference or merit of their error bound compared with those existing works.

3). For the supervised CL case (Theorem 4.2), I have the same concern as my first point. The error bound merely focuses on the training data, which cannot be generalized to unseen test data. Thus it does not really explain the effectiveness of the existing supervised CL model. Actually, we know that the supervised CL can also be regarded as a (supervised) similarity metric learning problem that has been widely studied in both theoretical and experimental aspects.

Given the above my concerns, I would like to vote for a weak reject. I will increase my score if I misunderstand something.

[R1]. Saunshi N, Plevrakis O, Arora S, et al. A theoretical analysis of contrastive unsupervised representation learning, ICML'19.
[R2]. Chen S, Niu G, Gong C, et al. Large-Margin Contrastive Learning with Distance Polarization Regularizer, ICML'21.

**Summary Of The Paper:**

This paper presents theoretical results to explain the effectiveness of contrastive learning (CL). For the self-supervised model, the authors prove new sin-distance-based error bounds for autoencoder (AE) and CL. For the fully-supervised model, the generalization error bound is derived. Numerical experiments validate the theoretical finds.

**Summary Of The Review:**

See the above.

---

> ### Author Response · Authors · 2021-11-19
> **Response to Reviewer R3zT**
>
> We are really grateful for your reviews and suggestions. There may be some misunderstandings in your comments that we clarify below.
>
> - First, we want to emphasize that $U^\star$ is the ground truth in our data generation model rather than the optimal solution on the training data, so the metric $\sin\Theta(U, U^{\star})$ measures how the learned linear transformation $W$ can recover the core features in the true underlying data generation model. Consequently, the results in Theorems 3.1, 3.2 and 4.2 are not ''based on training data only'', they in fact give error bounds about how could these algorithms learn the true model and generalize to unseen test data. Additionally, in Theorems 3.3 and 3.4, we have already explicitly given the error bounds of generalization performance on *unseen test data* in downstream tasks.
>
>   In this paragraph, we add more explanation about the metric $\sin\Theta(U, U^{\star})$, and hope this can clarify our results do show the generalization performance of CL as you asked for.  As discussed in the first two paragraphs on the top of page 4, we aim to learn a good projection matrix $W \in \mathbb{R}^{r\times d}$ onto a lower-dimensional subspace from observation $x$.
>   Since the information of $W$ is invariant under the transformation $W \leftarrow O W$ for any invertible matrix $O \in \mathbb{R}^{r\times r}$, the essential information of $W$ is contained in the right eigenvectors of $W$ which determines how $W$ compress $x\in\mathbb{R}^d$ into a low-dimensional representation $Wx \in\mathbb{R}^r$. For example, it makes no difference for the goodness of representation if we permute two rows or change their ratio in $W$ since it will yield the same performance when linear evaluation on any downstream tasks. Thus we quantify the goodness of the representation $W$ by the sine distance $\|\sin\Theta(U, U^\star)\|_F$, where $U$ is the top-$r$ right eigenspace of $W$.  We do not need to care about $S$ and $V$ (Recall that $U$,$S$ and $V$ is defined by singular value decomposition of $W=(USV)^T$, where $W\in\mathbb{R}^{r\times d},U\in\mathbb{O}^{d\times r},V\in \mathbb{O}^{r\times r}$ and $S\in\mathbb{R}^{r\times r}$ is a diagonal matrix with non-negative entries), which do not affect the goodness of representation and performance on downstream tasks.
>
> - The setting and the goal of this paper are quite different from previous works that you referenced. In [1] and [2], they aim to show why self-supervised contrastive learning can help learn representations that are useful in downstream tasks. Their main results (Theorem 4.1 and 4.5 in [1], Theorem 4 in [2])  characterize the gap between self-supervised loss and downstream loss: $|L_{SelfSup}-L_{Downstream}|\leq ...$. Instead, the result in our paper is based on the characterization of optimal solutions, which yields a direct bound on downstream performance. More specifically, in our paper, we directly obtain the downstream task risk bound based on the learned representation of contrastive learning and autoencoders. These results build the connection between the downstream performance and the representation learning of the contrastive learning. Importantly, we establish the results showing that why contrastive learning **outperforms** other representation methods such as autoencoders and further show that label information can gain accuracy in downstream tasks while possibly hurting transferability, while [1] and [2] did not directly compare with autoencoders and investigate the supervised setting. We believe that these are significant contributions of our results compared with the previous works [1] and [2].
>
> We hope that our response has addressed your concerns. We would very much appreciate it if you could reconsider your score based on the response. Please let us know if you have any further questions and we are happy to clarify. Thank you!
>
> ---
> [1]. Saunshi N, Plevrakis O, Arora S, et al. A theoretical analysis of contrastive unsupervised representation learning, ICML 19.
>
> [2]. Chen S, Niu G, Gong C, et al. Large-Margin Contrastive Learning with Distance Polarization Regularizer, ICML 21.

---

> ### Author Response · Authors · 2021-11-26
> **Dear R3zT: we'd love to see if you have any further questions after our response**
>
> Dear reviewer R3zT
>
> Thank you again for your time and feedback! We hope our detailed responses addressed all of your questions. Please let us know if you have any more comments and we are very happy to follow up. If you don't more questions, we hope you'd consider raising your score.
>
> Thank you very much!

---

### Official Review · Reviewer_wN4s · 2021-11-04

**Correctness:** 4
**Technical Novelty And Significance:** 4
**Empirical Novelty And Significance:** 2
**Recommendation:** 6
**Confidence:** 4

**Main Review:**

**Strengths**

The theoretical results explaining various interesting phenomena about contrastive learning are novel to the best of my knowledge. Contrastive learning and its variants have enjoyed a lot of success recently, and theoretical understanding of these methods is appreciated. The simplicity of the data model used to understand these phenomena is good in some ways; more on its weaknesses later. The paper is well written and easy to follow for most part. The notations, assumptions, theorem statements and discussions that ensued are clearly presented. Simulations verify many of the theoretical findings, e.g. existence of optimal $\alpha$ (relative strength of supervised learning and contrastive learning) when not enough source tasks available in transfer learning. The results seem mostly believable and a quick read of some parts of the Appendix did not raise any red flags; in fact the Appendix presentation also seemed clean.


**Weaknesses**

While the simplicity of the setting is not a huge weakness, it certainly raises the question of relevance of the analysis to practice. Evidence for a lot of empirical phenomena studied in this paper is provided by citing Khosla et al. and Islam et al. at various points in the paper. It would help to make these connections more explicit and present findings from prior work more clearly. One way to justify the setting would be to experimentally demonstrate relevance of some of the theoretical components on some real benchmark datasets. This would also increase the impact and visibility of the theoretical results.

It would also help to provide some more intuition for some of the results, for e.g. what is the reason for the extra bias term in contrastive learning compared to supervised contrastive learning? It seems like this arises from not learning the diagonal entries of $U^* {U^*}^{\top}$ due to the augmentation, but it's not clear why this is not a big issue for supervised learning. Similarly it will help to include some more intuition for why transfer learning fails and how self-supervised contrastive learning can save from that.




*Other comments/questions*

- Abstract should clearly specify that theoretical results are shown for simplified linear representation setting for a particular data model. The current phrasing makes the results seem very general

- It would be nice to present Theorem 3.1 and 3.2 for the same (or comparable) values of $\rho$ (signal to noise ratio), or perhaps make the dependency more explicit. Although both work for $\rho = \Theta(1)$, the dependence is hidden.

- The choice of augmentation (random coordinate subsets) is such that it aligns with independence structure of noise in data ($\Sigma$ being diagonal). This seems important for the superiority of contrastive learning over autoencoders. Could this have any connection to its superiority in practice?

- (Minor) The augmentation generation process from Definition 2.1 is different from what is done in practice, where the two views are typically independent of each other given an input $x$. In this paper the two views are dependent since they use complementary subset of coordinates. Perhaps the analysis could be extended to this independence case, e.g. two views can correspond to 2 random subsets of coordinates.

- Is there any specific reason for choosing HSIC apart from ease of analysis?

**Summary Of The Paper:**

This paper performs a theoretical study of various empirical phenomena regarding contrastive learning in a simplified setting, with a linear representation function, spiked covariance data model and linearized version of contrastive loss. In such a setting, the following results are shown theoretically (a) contrastive learning with a particular data augmentation can learn much better feature than an autoencoder by virtue of learning the underlying low rank signal as features, (b) supervised contrastive learning can do better than contrastive learning by getting rid some bias that data augmentation introduces, (c) for transfer learning, a combination of unlabeled contrastive loss and supervised learning can do better than either of them individually. Simulation experiments are used to verify many of these findings.

**Summary Of The Review:**

Overall the theoretical results are quite interesting, for a very relevant problem of contrastive learning. The main negative point is about the relevance of the results to more practical settings.

---

> ### Author Response · Authors · 2021-11-19
> **Response to Reviewer wN4s (Part 1/2)**
>
> We would like to thank the reviewer for your positive comments and insightful suggestions. In the following, we address your concerns point by point.
>
> - Regarding relevance to practice, we have added discussions in the paper to highlight how our theory agrees well with previous empirical observations:
>
>   1. We would like to refer to Figure 1 in [1] and Figure 5 in [2] for the comparison between the generative and contrastive self-supervised learning, where the contrastive approach has superior performance compared to the generative approach. These results are consistent with our theory in Theorem 3.1-3.4, where we showed that contrastive learning achieves better performance compared with autoencoder, a representative generative self-supervised learning method.
>   2. For the comparison between supervised contrastive learning and self-supervised learning on downstream tasks, please see Table 2 in [3] and the first column of Table 4 in [4], supervised contrastive learning, again, has significant improvement with 7\%-8\% accuracy increase on ImageNet and Mini-ImageNet. This observation is consistent with our finding in Theorem 4.1, where we proved that supervised contrastive learning can achieve a better upper bound.
>   3. For the transfer learning performance, please see Table 4 in [3] and Table 4 in [4], where the supervised contrastive learning hardly increases the predictive accuracy compared to the self-supervised contrastive learning (the difference of mean accuracy is less than 1\%) and can harm significantly on some datasets (e.g. 5.5\% lower for SUN 397 in Table 3 of [3]).  These results indicate that some mechanisms in supervised contrastive learning hurt model transferability since the improvement on source tasks is significant. Moreover, in Table 4 of [4], it was observed that combining supervised learning and self-supervised contrastive learning together achieve the best transfer learning performance compared with each of them individually. These findings are consistent with our theory in Theorem 4.2, where we showed that the error can increase as $\alpha$ (the ratio between supervised learning loss and self-supervised learning loss) grows and the optimal error is achieved when choosing a moderate $\alpha$, i.e., combining self-supervised learning and supervised learning together.
>
>   These empirical observations provide sound support for our theory, and we have added this discussion to the revised paper (Section E.1).
>
> - Regarding the proof intuition about supervised contrastive learning, the idea is that for self-supervised learning, where the label information is not available, we can only extract features by introducing augmentations and have to compromise to the possible domain shift bias. However, when label information is available, we can select multiple positive samples for each anchor based on the label information. This approach allows us to extract features from data with less bias compared to augmented pairs generation in the self-supervised setting. Here we do not use data augmentation for the supervised contrastive learning for the clarity of presentation, as it can help us disentangle how the data augmentation and label information help to extract features in contrastive learning respectively.
>
> - Regarding the proof intuition about transfer learning, when tasks are not diverse enough, supervised training will only focus on the features that are helpful to predict labels of source tasks and ignore other features. For example, we have unlabeled images which contain cats or dogs and the background can be sandland or forest. If the source task focuses on classifying the background, supervised learning will not learn features associated with cats and dogs, while self-supervised learning can learn these features since they are helpful to discriminate different images. As a result, though supervised learning can help to classify sandland and forest, it can hurt performance on the classification of dogs and cats and we should incorporate self-supervised contrastive learning to learn these features.
>
> ---
>   [1]Chen T, Kornblith S, Norouzi M, et al. A simple framework for contrastive learning of visual representations. ICML 2020.
>
>   [2]Liu X, Zhang F, Hou Z, et al. Self-supervised learning: Generative or contrastive. IEEE Transactions on Knowledge and Data Engineering, 2021.
>
>   [3]Khosla P, Teterwak P, Wang C, et al. Supervised contrastive learning. NeurIPS 2020.
>
>   [4]Islam A, Chen C F, Panda R, et al. A Broad Study on the Transferability of Visual Representations with Contrastive Learning. ICCV 2021.

---

> > ### Author Response · Authors · 2021-11-19
> > **Response to Reviewer wN4s (Part 2/2)**
> >
> > - Thank you for the suggestions regarding the abstract, we will revise accordingly and explicitly point out the model we analyze.
> >
> > - Regarding the question about the value of $\rho$, it in fact takes the same value for both Theorem 3.1 and 3.2. For an explicit dependence, please see the last two equations in the proof of Theorem B.1 (page 19) and the last two equations in the proof of Theorem B.3 (page 23), where we prove Theorem 3.1 and 3.2 for any $\rho$ and only plugin $\rho=\Omega(1)$ at the last step for the simplicity of presentation.
> >
> > - In this paper, we focus on random masking augmentation, which has also been used in other works on the theoretical understanding of contrastive learning, eg. [1].  The random masking augmentation is an analog of the random cropping augmentation used in practice. As shown in [2], cropping augmentation achieves overwhelming performance on linear evaluation (ImageNet top-1 accuracy) compared with other augmentation methods, please see figure 5 in [2] for details.
> >
> > - HSIC is a widely used measurement in various machine learning problems, such as feature selection [3], feature extraction [4], clustering [5], and supervised PCA [6]. It can be viewed as an unregularized version of the mean square error for the downstream task, and a detailed discussion can be found in section A.3.1. Since we have added the regularization term in the self-supervised learning loss, here we use HSIC to derive an explicit solution that allows us to provide a solid theoretical foundation to justify the recent (somewhat surprising) empirical observations that including labels information in contrastive learning may hurt the transferability [7,8].
> >
> > We hope our response can address your concerns and that you will consider increasing the rating. We would be happy to further elaborate/respond to any other questions that you may have during the discussion period!
> >
> > ---
> >
> >   [1] Wen, Z., \& Li, Y. (2021). Toward Understanding the Feature Learning Process of Self-supervised Contrastive Learning. ICML 2021.
> >
> >   [2]Chen T, Kornblith S, Norouzi M, et al. A simple framework for contrastive learning of visual representations. ICML 2020.
> >
> >   [3] Le Song, Alex Smola, Arthur Gretton, Karsten M Borgwardt, and Justin Bedo. Supervised feature selection via dependence estimation. ICML 2007.
> >
> >   [4] Le Song, Arthur Gretton, Karsten Borgwardt, and Alex Smola. Colored maximum variance unfolding. NeurIPS 2007.
> >
> >   [5] Le Song, Alex Smola, Arthur Gretton, and Karsten M Borgwardt. A dependence maximization view of clustering. ICML 2007.
> >
> >   [6] Elnaz Barshan, Ali Ghodsi, Zohreh Azimifar, and Mansoor Zolghadri Jahromi. Supervised principal component analysis: Visualization, classification and regression on subspaces and submanifolds. Pattern Recognition, 44(7):1357–1371, 2011.
> >
> >   [7]Khosla P, Teterwak P, Wang C, et al. Supervised contrastive learning. NeurIPS 2020.
> >
> >   [8]Islam A, Chen C F, Panda R, et al. A Broad Study on the Transferability of Visual Representations with Contrastive Learning. ICCV 2021.

---

### Official Review · Reviewer_TTCS · 2021-11-08

**Correctness:** 2
**Technical Novelty And Significance:** 3
**Empirical Novelty And Significance:** 2
**Recommendation:** 6
**Confidence:** 3

**Main Review:**

The main result of this paper sounds interesting, despite the analysis conducted on linear models.
The result shows that in some cases (large noise regime), the autoencoder cannot recover the core features and thus performs badly on downstream tasks, but the contrastive learned model has the ability to recover the core features due to the random augmentation.
Therefore, we can see the strong power of contrastive learning, at least in the large noise regime.
The analysis of supervised contrastive learning is also novel and convincing.

I have some detailed questions about the comparison between autoencoder and contrastive learning (Thm 3.2 and 3.3).
- Does $c'$ in Thm 3.4 depend on $r$ or $r_c$? If so, can we say that $c'$ is worse than the bias term (first term) in Thm 3.3? If not, why do we need to assume that $r\le r_c$ for all $r$? Sorry for that the proofs in the appendix are not clear for me to read.
- Do we use data augmentations for autoencoder training in the analysis? If so, the comparison between autoencoder and contrastive learning is fair. If not, contrastive learning seems to involve more human knowledge for the training.

Moreover, I wonder how much insight of this paper can be extended to non-linear models and other augmentations? Do you have any empirical studies to verify how many theoretical observations still hold for those cases?

Minor comment: The appendix is not well organized. It is hard to see a clear one-to-one match between theorems in the main text and the proofs in the appendix.

**Summary Of The Paper:**

This paper studies the generative power of contrastive learning from a theoretical perspective. To enable the analysis, this paper considers the linear model with random masking augmentation. Training data are assumed to be generated by the spiked covariance model. The main result is that contrastive learning can recovery the core features since the random masking augmentation can mitigate the influence of random noise on the diagonal entries in the covariance matrix, while the autoencoder is unable to recover the core features due to the large noise. Then this paper shows that downstream excess risk can be upper bounded by contrastive learning but cannot be less than a universal constant for an autoencoder. Beyond the unsupervised setting, this paper also studies the role of labeled data in supervised contrastive learning and gives the upper bound of sine distance between the supervised contrastive learned model and the true model.


**Summary Of The Review:**

Overall, I think this paper is good and interesting. My main concern is 1) whether the comparison between autoencoder and contrastive learning is fair, and 2) how much insight can be extended to the more realistic scenarios.

---

> ### Author Response · Authors · 2021-11-19
> **Response to Reviewer TTCS**
>
> We are really grateful for your comments and suggestions. In the following, we address your concerns point by point.
>
> - The constant $c'$ appearing in Theorem 3.4 is a constant term independent of $d$ and $n$. Thus, when $d$ is sufficiently large compared to $r$, the upper bound of downstream task performance via contrastive learning (in Theorem 3.3) is smaller than the lower bound of downstream task performance via autoencoders (in Theorem 3.4). The assumption of $r \leq r_c$ in Theorem 3.4 is assumed for clarity of presentation. Using the same techniques in the proof of Theorem 3.4, one can obtain a constant lower bound for autoencoders with slightly stronger assumptions, e.g., $\rho^2 = O(1/\log d)$ or $n \gg dr$, without assuming $r \leq r_c$. Our theory can be adapted to both of these assumptions. We will clarify it in the main text. We also slightly changed the structure of the appendix and moved the proofs of Theorem B.2 and B.3 (corresponding to Theorem 3.3 and 3.4) just below the theorem statement.
>
> - Thank you for pointing this out. In our current paper, we do not use data augmentation for autoencoders. In the following, we will argue that such a comparison is still fair.
>
>   First, in practice, augmentation is an essential step in contrastive learning to generate positive samples, which helps contrastive learning learn meaningful representations. By contrasting the two augmented views, contrastive learning can effectively extract core representations.
>   In contrast, for the standard autoencoder, data augmentation is not applied. [1,2]
>
>   Second, applying the same data augmentation considered for contrastive learning (Definition 2.2) to autoencoder makes no difference to the results in Theorems 3.1 and 3.4. More specifically, a brief proof sketch is as below and we have added details of this in Supplementary Section B  of the revised paper (Theorem B.4 \& B.5):
>
>   When random masking data augmentation is applied, the optimization problem of autoencoders can be formulated as equation B.25. Then by similar proof as Theorem B.1 and Corollary B.1, we can obtain that the optimal solution $W_{AE}$ is given by:
>
>   $$
>   W_{AE}=C\Big(U_{AE}\Sigma_{AE}V_{AE}^\top\Big)^\top,
>   $$
>
>   where $C>0$ is a positive constant, $U_{AE}$ is top-$r$ singular vectors of matrix $M:=D(XX^\top)+\frac{1}{2}\Delta(XX^\top)$ (recall that $D(M)$ is defined as the matrix $M$ with off-diagonal entries set to be zero and $\Delta$ is defined as $M-D(M)$), $\Sigma_{AE}$ is a diagonal matrix consisting of eigenvalues of $M$ and $V_{AE}=[v_1,\cdots,v_n]\in\mathbb{R}^{r\times r}$ can be any orthonormal matrix. Now by the same proof strategy of Theorem 3.1 and 3.4, we obtain the same equation as in Eq. (3.3), which implies the constant lower bound result does not change when the augmentation is applied to autoencoder.
>
> - Regarding the extension to other augmentations, our theory provides a general framework to analyze contrastive learning in the linear representation setting, which can be potentially used for other data augmentations and data generation models. Although we focus on random masking to show how contrastive learning learns features in this paper, we also provided Theorem B.1 (in the appendix) for general augmentations. It will be exciting to combine our work with recent theory [3,4] on general data augmentation. Regarding extension to non-linear settings, the major issue is to determine the solution given by different algorithms. We think the techniques in [5] can be adopted to generalize our results to one hidden layer neural network. These are very interesting directions for further exploration and we will continue to work on them.
>
> - Regarding the appendix, thanks for pointing out the problems of the organization. We have added restatements of the theorem to the appendix in the sequential versions.
>
> ---
> [1]Goodfellow I, Bengio Y, Courville A. Deep learning[M]. MIT Press, 2016.
>
> [2]Fan J, Ma C, Zhong Y. A selective overview of deep learning. Statistical Science, 2021, 36(2), 264-290.
>
> [3] Chen S, Dobriban E, Lee J H. A group-theoretic framework for data augmentation. Journal of Machine Learning Research, 2020, 21(245): 1-71.
>
> [4] Dao T, Gu A, Ratner A, et al. A kernel theory of modern data augmentation. ICML 2019
>
> [5] Wen, Z., \& Li, Y. (2021). Toward Understanding the Feature Learning Process of Self-supervised Contrastive Learning. ICML 2021.

---

> > ### Comment · Reviewer_TTCS · 2021-11-26
> > **Further Comments**
> >
> > Thanks for the explanation.
> >
> > I am not convinced by the second point of the usage of data augmentation.
> >
> > > "When random masking data augmentation is applied, the optimization problem of autoencoders can be formulated as equation B.25".
> >
> > I think directly using data augmentation for increasing sample size in autoencoder is not a meaningful way and is still unfair for comparison.
> >
> > In particular, in contrastive learning, the pair of positive samples are complementary (i.e., $t_1(x)+t_2(x)=x$). By comparing them, the model can learn a lot of useful information from the data augmentation.
> > However, directly using data augmentation for increasing sample size in autoencoder wastes a lot of useful information hidden in the data augmentation.
> > A meaningful usage might be to recover $x$ instead of $t_1(x)$ from $t_1(x)$ or to predict $t_2(x)$ by $t_1(x)$.
> >
> > ---
> >
> > Furthermore, I thought the value of this paper is showing that the contrastive approach is theoretically better than the autoencoder (even some assumptions are made: linear, large noise, etc.). Therefore,  it suggests people pay more attention to exploring contrastive learning rather than autoencoder in self-supervised learning. However, this work can only claim that the contrastive approach **with** data augmentation is better than the autoencoder **without** data augmentation. This will weaken the value of this paper. In fact, a recent work [6] shows the great power of autoencoder with data augmentation.
> >
> > In summary, I think this is a borderline paper.
> >
> > [6] He, Kaiming, Xinlei Chen, Saining Xie, Yanghao Li, Piotr Dollár, and Ross Girshick. "Masked Autoencoders Are Scalable Vision Learners." arXiv preprint arXiv:2111.06377 (2021). https://arxiv.org/abs/2111.06377

---

> > > ### Author Response · Authors · 2021-11-26
> > > **Thank you for your additional feedback**
> > >
> > > Thank you very much for your comment. We agree with you that adding masking to autoencoders, like in the recent reference you cited, is a very interesting approach to improve autoencoders. However, we still believe that it is valuable to theoretically analyze the performance of contrastive learning with standard autoencoders because: 1) standard autoencoders (i.e. without augmentation) is still commonly used in practice and 2) several empirical papers directly compared contrastive learning with autoencoders (eg. Table 1 in [1], Table VIII in [2], and Table 3 in [3]) and that motivated our work to explain these theoretical findings. Our work provides the **first** theoretical framework for comparing contrastive learning with autoencoders, enlarging our theoretical toolkit. This fills a useful gap in literature and we believe it can be a springboard for analyzing more advanced versions of masked autoencoders in future works like you suggested. Our paper, combined with the reference you suggested, indeed illustrate that data augmentation (in the contrastive fashion) is essential for autoencoders (from theoretical and empirical perspectives respectively), which is an important insight to our knowledge. Our paper, again, provides **new** theoretical evidence to support this important insight. We will add this discussion and the reference you suggested to our paper's Discussion section in the revised version.
> > >
> > > We also want to note that another baseline method that is often used to compare with contrastive learning is the GANs framework [4,5], and our result (Theorem 3.1) can also be used to justify why contrastive learning is better than the standard GANs, since [6] showed that the global solution for standard GANs recovers the empirical PCA solution as the generative model.
> > >
> > > We also want to highlight that in addition to the results we mentioned above, our paper also provides the **first** theoretical justification on how the label information in contrastive learning plays a role in downstream tasks, and theoretically justify the previously observed empirical phenomenon that supervised contrastive learning would sometimes hurt the transfer learning performance.  We believe these are also significant contributions in our work.
> > >
> > > ---
> > > [1]Kipf T, van der Pol E, Welling M. Contrastive Learning of Structured World Models. ICML 2019.
> > >
> > > [2]Cao Z, Li X, Zhao L. Unsupervised Feature Learning by Autoencoder and Prototypical Contrastive Learning for Hyperspectral Classification. arXiv preprint arXiv:2009.00953, 2020.
> > >
> > > [3]You Y, Chen T, Sui Y, et al. Graph contrastive learning with augmentations. NeurIPS 2020.
> > >
> > > [4]Chen T, Kornblith S, Norouzi M, et al. A simple framework for contrastive learning of visual representations. ICML 2020.
> > >
> > >  [5]Liu X, Zhang F, Hou Z, et al. Self-supervised learning: Generative or contrastive. IEEE Transactions on Knowledge and Data Engineering, 2021.
> > >
> > > [6]Feizi S, Farnia F, Ginart T, et al. Understanding gans in the LQG setting: Formulation, generalization and stability. IEEE Journal on Selected Areas in Information Theory, 2020, 1(1): 304-311.

---

### Author Response · Authors · 2021-11-19
**An updated version for the paper**

We thank all the reviewers for their helpful feedback. We have uploaded a revised paper based on their suggestions, and the major revisions are summarized below:

- In the appendix, we made the organization more clear and also added a restatement of each Theorem appearing in the main text and a paragraph summarizing the organization of proof in each section.
- We added Theorem B.4 and B.5 in the appendix to show that autoencoders with an additional random masking augmentation make no difference to our original results for standard autoencoders, which indicates that our comparison between the contrastive learning and autoencoders is fair.
- We added Section E.1 to list details of empirical results in related works, which provide sound support to our theory.
- We fixed some minor typos in our paper.

---

### Decision · Program_Chairs · 2022-01-20

**Decision:**

Reject

**Comment:**

While the reviewers appreciated the theoretical analysis performed in this work, some concerns were raised during discussion, such as how relevant the obtained results are wrt current contrastive learning practices, and whether the comparison against a simple auto-encoder (basically PCA) is fair or insightful. The authors' response did not satisfactorily address these concerns. Overall, the current work appears to be preliminary, and important questions were left out: (a) how realistic the assumptions are (linear, spike covariance, Bernoulli random augmentation)? (b) performing better than PCA in a specifically designed setting may not be as impressive as it appears; what can we say about the optimality of CL against any algorithm? (c). validation on existing benchmark and CL algorithms would be welcome. The authors are encouraged to revise the current draft by incorporating the reviewers' comments and submit again. In its current form, we believe this work is not ready yet.